# Streamflow drought: implication of drought definitions and its application for drought forecasting

Samuel J. Sutanto[1] and Henny A. J. Van Lanen[1]

[1]Hydrology and Quantitative Water Management Group, Environmental Sciences Department, Wageningen University and Research, Droevendaalsesteeg 3a, 6708PB, Wageningen, the Netherlands

**Correspondence:** Samuel Sutanto (samuel.sutanto@wur.nl)

**Abstract.** Streamflow drought forecasting is a key element of contemporary Drought Early Warning Systems (DEWS). The term streamflow drought forecasting (not streamflow forecasting), however, has created confusion within the scientific hydro-meteorological community, as well as in operational weather and water management services. Streamflow drought forecasting requires an additional step, which is the application of a drought identification method to the forecasted streamflow time series. The way, streamflow drought is identified, is the main reason for this misperception. The purpose of this study, therefore, is to provide a comprehensive overview of the differences between different drought identification approaches to identify droughts in the European rivers, including an analysis of both historical drought and implications for forecasting. Streamflow data were obtained from the LISFLOOD hydrological model forced with gridded meteorological observations (known as LISFLOOD-Simulation Forced with Observed, SFO). The same model fed with seasonal meteorological forecasts of the European Centre for Medium-range Weather Forecasts system 5 (ECMWF SEAS 5) was used to obtain the forecasted streamflow. Streamflow droughts were analyzed using the daily and monthly Variable Threshold methods (VTD and VTM, respectively), daily and monthly Fixed Threshold methods (FTD and FTM, respectively), and the Standardized Streamflow Index (SSI). Our results clearly show that streamflow droughts derived from different approaches deviate from each other in their characteristics, which also vary in different climate regions across Europe. The daily threshold methods (FTD and VTD) identify 25-50% more drought events than the monthly threshold methods (FTM and VTM) and accordingly the average drought duration is longer for the monthly than for the daily threshold methods. The FTD and FTM, in general, identify drought occurrences earlier in the year than the VTD and VTM. In addition, the droughts obtained with VTM and FTM approaches also have higher drought deficit volumes (about 25-30%) than the VTD and FTD approaches. Overall, the characteristics of SSI-1 drought are close to what is being identified by the VTM. The different outcome obtained with the drought identification methods illustrated with the historical analysis is also found in drought forecasting, as documented for the 2003 drought across Europe and for the Rhine River specifically. In the end, there is no unique hydrological drought definition (identification method) that fits all purposes, hence developers of DEWS and end-users should clearly agree in the co-design phase upon a sharp definition on which type of streamflow drought is required to be forecasted for a specific application.

## 1 Introduction

Drought is a creeping natural disaster that has major socio-economic and environmental impacts across the world (e.g., Tallaksen and Van Lanen , 2004; Wilhite et al. , 2007; Ding et al., 2011; Van Dijk et al. , 2013; Stahl et al., 2016; Haile et al. , 2019). The IPCC  (2014) reports with very high confidence that impacts of drought on society are already considerable. Drought hazard and their impacts are projected to increase in numerous regions under a future warmer climate (e.g., Feyen and Dankers , 2009; Forzieri et al., 2014; Prudhomme et al., 2014; Wanders et al. , 2015; Samaniego et al., 2018; Cammalleri et al., 2020). Gu et al. (2020) analyzed how drought influences regional gross domestic product (GDP) under different representative concentration pathways (RCPs) and shared socioeconomic pathways (SSPs) at the global scale. The fraction of drought-affected GDP relative to the country's GDP would equal 100% in over about 75 countries under $1.5°C$ warming, which is projected to increase to over 90 countries under $2.0°C$ warming. There is an urgent necessity for society to respond to these signs. National Drought Policy Plans (NDPPs) should be implemented that convert the usually reactive drought crisis management into a pro-active risk management (Sivakumar et al., 2014; WMO and GWP , 2014; Poljanšek et al., 2017). One of the elements to be included in the NDPP is a Drought Early Warning System (DEWS) that in addition to real-time monitoring contains operational drought forecasting with appropriate lead times, i.e. multi-months or seasonal.

The term drought forecasting has been used in an indefinite way, which has created misconceptions, miss-citations, and confusion in the scientific hydro-meteorological community (authors, readers, editors, and reviewers), as well as among policy makers, and in operational weather and water management services. An explicit definition of what is being forecasted is crucial to avoid any misunderstanding on the usability of drought forecast products for different purposes. Firstly, meteorological drought forecast systems have been developed (e.g., Mishra and Desai , 2005; Belayneh et al., 2014; Dutra et al., 2014), which frequently use the Standardized Precipitation Index, SPI (McKee et al., 1993), or the Standardized Precipitation Evaporation Index, SPEI (Vicente-Serrano et al., 2010). These standardized drought indices aggregate precipitation (SPI), and precipitation minus potential evaporation (SPEI) over at least one month and have lead times of one to several months. It should be noted that conventional weather forecast systems, which predict low or no precipitation and above normal temperature, as part of their regular suite of forecast products, should not be classified as a drought forecasting system, because of their rather short lead time (sub-daily to 10-15 days). Secondly, hydrological drought forecasts are provided (e.g., Pozzi et al., 2013; Sutanto et al., 2020a), which involve groundwater, river flow, soil moisture, and runoff. Hydrological drought deviates from meteorological drought (e.g., Changnon , 1987; Peters et al., 2003; Mishra and Singh, 2010; van Loon and van Lanen , 2012; Barker et al., 2016; Sutanto et al., 2020b), which means that the latter cannot straightforwardly be used to predict drought in groundwater or river flow. Because of all these differences, an explicit delineation of what is being forecasted is a prerequisite. Here, our study focuses on streamflow drought forecasting, as part of hydrological drought forecasting, which is defined as the forecasting of below-normal streamflow (Hisdal et al. , 2004; Peters et al., 2006; Fleig et al., 2006; Feyen and Dankers , 2009; Sarailidis et al., 2019).

Forecasting of streamflow drought follows different approaches on how the hydrological drought is defined (Hisdal et al. , 2004; van Loon , 2015), which is also essential to consider when using forecast products. Yuan et al.  (2017) use the so-

called standardized approach. They forecasted the Standardized Streamflow Index (SSI), which measures monthly normalized
anomalies in streamflow and, if negative, then SSI signifies a dry anomaly. Others applied the threshold approach to predict
drought in river flow from the forecasted flow time series. This implies that the river is in drought when it is below a predefined
flow. Marx et al. (2018) and Wanders et al. (2019) use a fixed threshold meaning that it does not vary throughout the year.
Usually, a percentile of the flow duration curve is taken using all flow data to identify the fixed threshold. On the contrary,
Fundel et al. (2013), Sutanto et al. (2020a), and van Hateren et al. (2019) have used the variable threshold approach to identify
drought events with their hydrological drought forecasting system. In this approach, the threshold varies over the year and
accounts for seasonality, which means that forecasted drought can occur in every season. The threshold is derived from, for
instance, the daily, monthly, or seasonal flow duration curve.

In the context of this study, it is also important to note, that hydrological drought forecasting is different from just streamflow
forecasting (e.g., Day , 1985; Clark and Hay , 2004; Schaake et al., 2007; Bell et al., 2017; Mendoza et al., 2017; Arnal et al.,
2018; Duan et al., 2019), although the latter provides key input data to derive hydrological drought. For hydrological drought
forecasting, an additional step has to be taken, that is, derivation of drought events from the forecasted flow time series, e.g. the
flow time series is converted into a time series of drought events. In summary, the different approaches that are being used to
identify and communicate drought in rivers call for an explicit description of what is being meant. Clearly, different users have
diverse needs and these can be accommodated by forecasts of drought indices obtained by different identification approaches,
such as standardized drought indices (SPI, SPEI, SSI) and threshold drought indices (variable threshold and fixed threshold),
as provided in the DEWS.

The purpose of this study, therefore, is firstly to provide a clear overview of the differences between streamflow drought using
different definitions (i.e. identification methods) and temporal resolutions, i.e. daily and monthly. This is done through a historic
analysis using streamflow data from 1990-2018. Differences are illustrated for entire Europe to investigate spatial aspects and
some major rivers across different climate regions to study temporal aspects. The historical analysis is innovative because it
covers the entire pan-European river network with all its hydrological regimes instead of a single country (Heudorfer and Stahl
, 2017; Vidal et al. , 2010) or a river basin (Sarailidis et al., 2019) and involves both threshold and standardized identification
approaches (drought indices), including different temporal resolutions. Secondly, in this study, the implications of applying
different drought identification approaches for forecasting hydrological drought are elaborated using the extreme 2003 drought
in Europe as an example, which demonstrates that none of the hydrological drought forecast approaches fits all needs.

The paper is organized as follows: the datasets with observed and ensemble forecasts of streamflow, used in this study, are
described in Section 2.1, followed by a description of the methodology to derive the drought indices, i.e. drought identification
approaches (Section 2.2), an explanation of presented characteristics, such as the number of drought occurrences (frequency),
timing, duration, and deficit volume (Section 2.3). The results are presented and discussed in Section 3. We divided the results
and discussion section into two parts, that are, drought characteristics analysis using different identification approaches for 1)
historical data, and 2) forecasted data of the 2003 drought. Detailed analysis of drought characteristics for both historic and
forecasts is provided for the selected river basins. Finally, we conclude the findings in Section 4.

## 2 Data and Methods

### 2.1 Data

A state-of-the-art hydrological model, LISFLOOD, was used to simulate the streamflow of rivers across Europe from 1990 to 2018, which was derived from the routed runoff of 5 x 5 km grid cells (van der Knijff et al., 2010; Burek et al., 2013a). We decided to use simulated river flow rather than the observed flow, because sufficiently-long time series of observed flow for a common period covering the whole of the European river network do not exist. The LISFLOOD model was fed by gridded meteorological observations (e.g. precipitation, temperature, relative humidity, wind speed) to obtain daily proxies for observed

streamflow data, known as LISFLOOD-Simulation Forced with Observed (SFO) (hereafter referred to observed streamflow for simplicity). The gridded meteorological observation data were collected from ground observations (>5000 synoptic stations), obtained from the Joint Research Centre (JRC) meteorological database, the Global Telecommunication System of the WMO, and high-resolution data received from the National Member States institutions (Pappenberger et al., 2011). The time series of observed streamflow data for each cell at the river network (river grid cell) across Europe were used to derive the streamflow

drought following different approaches. In this study, however, we only selected major European rivers, indicated by river cells that have average discharge above 10 m$^3$ s$^{-1}$. Potential evapotranspiration was calculated through the offline LISVAP pre-processor based on the Penman-Monteith equation (van der Knijff, 2008; Burek et al., 2013b). A kinematic wave approach was used for routing the water movement on the river network.

The model was calibrated using time series of observed streamflow from over 700 calibration stations across Europe. The

hydrological skill of the LISFLOOD model expressed by the Kling-Gupta Efficiency (KGE) shows that 42% of all calibration stations score a KGE higher than 0.75, 33% of all stations score a KGE between 0.5 and 0.75, and 25% of all stations score a KGE below 0.5 (Arnal et al., 2019). Although the model was originally developed for operational flood forecasts in the EU under the European Flood Awareness System (EFAS) platform (Thielen et al., 2009; Pappenberger et al., 2011; Cloke et al., 2013), the LISFLOOD model has been tested for drought identification, forecasting and projections (Feyen and Dankers ,

2009; Sepulcre-Canto et al., 2012; Trambauer et al., 2013; Forzieri et al., 2014; Sutanto et al., 2019, 2020a, b; van Hateren et al., 2019). It appears from these studies that the model also performs rather well for drought studies. The model used in this study is the latest version of LISFLOOD that has been implemented in the operational EFAS since 2019 (version 3).

Besides the SFO data, we also used re-forecasted (known as hindcast) time series of streamflow data for the year 2003, as an example of drought forecasts. The European Centre for Medium-range Weather Forecasts System 5 (ECMWF S5)

seasonal forecast was used as forcing for the LISFLOOD hydrological model to forecast streamflow at the pan-European scale (Stockdale et al., 2018). The seasonal forecasts are available as daily re-forecast data for each month from day 1 to day 215 ( 7 months lead-time) for 25 ensemble members (see Sutanto et al., 2020a, for detailed information). In this study, we selected the re-forecast data from 2003, because a severe drought across extended areas in Europe was observed (Fink et al., 2006; Ionita et al. , 2017; Laaha et al., 2017).

## 2.2 Streamflow drought identification

In this study, we employed two well-known drought identification methods, i.e. the threshold drought approach and the standardized drought approach (van Loon , 2015).

### 2.2.1 The variable and fixed threshold methods

Using both the variable and fixed threshold-based approaches, drought was derived from time series of observed streamflow data from 1990 to 2018 and re-forecasted data of 2003 to calculate the water deficit in the streamflow. The threshold approach originates from the theory of runs and is developed based on a pre-defined threshold level (Yevjevich , 1967; Zelenhasic and Salvai , 1987; Hisdal et al. , 2004). The threshold approach uses an event-based sampling of the flow time series to convert this into a time series of drought events. The drought event starts when the hydrological variable falls below the threshold value and ends when it equals or rises above the threshold value. In this study, we applied two different types of drought threshold approaches, which are the Variable Threshold and the Fixed Threshold on both daily (VTD and FTD) and monthly streamflow data (VTM and FTM). The latter was done to allow a comparison of the VTM and FTM approaches with the Standardized Streamflow Index (SSI, Nalbantis and Tsakiris , 2009; Vicente-Serrano et al., 2012), which uses a monthly temporal resolution. However, the use of threshold approaches on monthly streamflow data to identify monthly drought is not common practice (e.g., Fleig et al., 2006; Peters et al., 2006; Hannaford et al. , 2011; Prudhomme et al., 2014; van Loon , 2015; Marx et al., 2018; Wanders et al. , 2019). To the author's knowledge, only a few studies used monthly data (e.g., Tallaksen et al. , 2009; van Loon et al. , 2019) to derive drought using the threshold method, and this was done only for scientific purposes.

The fixed threshold approach uses a pre-defined threshold, which is constant over the year and unique for each river grid cell. The pre-defined variable threshold varies for each day/month and for each river grid cell. The variable threshold method gains more popularity because this method considers seasonality in streamflow (Hannaford et al. , 2011; Prudhomme et al., 2011; van Loon , 2015). For the variable and fixed thresholds, we calculated the threshold values using 29 years of monthly streamflow data that were obtained by averaging daily flow data into monthly. Thresholds in this study were derived from the 80[th] percentile of the streamflow (Q80, flow duration curve), which are the flows that are equaled or exceeded 80 percent of the time. Moreover, the Q80 threshold lays within the range of the 70[th]-90[th] percentile that is commonly used in drought studies (Tallaksen et al. , 1997; Hisdal et al. , 2004; Fleig et al., 2006; Wong et al. , 2011). We would like to note that the use of Q80 is not suitable for arid regions where many zero values in the observed streamflow are observed. The use of a higher threshold level or another method such as the consecutive dry period method is recommended (van Huijgevoort et al., 2012). For the VTM method, the calculated 12 monthly thresholds could straightforwardly be used in the drought analysis. For the VTD method, the calculated monthly thresholds were firstly assigned as the threshold levels for each day of the respective months. This resulted in a jump between two consecutive months, which showed unrealistic drought behavior. Therefore, as a second step, a 30-day centered moving average (30DMA) smoothing technique was applied to the monthly thresholds, eventually leading to daily thresholds (365 and 366 thresholds for no leap and leap years, respectively) (van Loon et al. , 2012; van Lanen et al., 2013; Beyene et al., 2014). Beyene et al. (2014) describe this method as the moving average of monthly quantile

(M_MA). For the FTM and FTD method, we used the same threshold, which is constant throughout the year by definition. In the drought analysis, the same threshold values are applied every year from 1990 to 2018.

The method that we applied to calculate the VTD used monthly streamflow data instead of daily data corresponds with the literature (e.g., van Loon et al. , 2012; van Lanen et al., 2013; Beyene et al., 2014; van Huijgevoort et al., 2014). Use of daily data to calculate the threshold or other methods, such as the 30 days moving window quantile (30D) or fast Fourier transform of daily quantile (D_FF), as introduced by Beyene et al. (2014) is also possible. However, there is no approach that is perfect to identify drought for the whole Europe. For example, the 30D approach shows good performance in detecting drought in

snow-dominated regions compared to others, whereas the D_FF approach performs lower in several catchments in Europe (Beyene et al., 2014). Therefore, in this study, we only focus on the use of the VTD method that has been widely applied in many drought studies.

    The centered 30DMA method was also employed in the historical daily streamflow data to reduce the number of minor droughts (pooling procedure) (Fleig et al., 2006; van Loon and van Lanen , 2012; Sarailidis et al., 2019). Appendix A provides

details how the 30DMA method has been implemented.

### 2.2.2   The standardized streamflow index

The Standardized Streamflow Index (SSI, Nalbantis and Tsakiris , 2009; Vicente-Serrano et al., 2012) was also used to identify drought in the river. The SSI expresses the streamflow as a non-exceedance probability and was calculated using the same theoretical background as the Standardized Precipitation Index (SPI, McKee et al., 1993). The SSI calculation for any river grid

cell was based on the monthly streamflow record that is fitted to a probability distribution, that is gamma in this study, which is then transformed into the standard normal distribution so that the expected median SSI for the site and desired period is zero. The alpha and beta parameters of the gamma probability density function are estimated for each grid cell and for each month of the year using the method of moments. One should note that the gamma distribution for SSI might not be the best choice in all cases. Some previous studies suggest to use several other approaches, such as Plotting Positions, Tweedie, Generalized

Extreme Value, and Generalized Logistic (Vicente-Serrano et al., 2012; Svensson et al., 2017; Tijdeman et al., 2020). Some of these methods were claimed to be more accurate to derive the SSI in particular cases than the gamma distribution. However, we decided to use the gamma distribution as general distribution for the whole Europe since it can be used for hydrological forecasting of both high and low flows (Slater and Villarini , 2018). Moreover, none single probability distribution would fit all streamflow time-series across Europe (Vicente-Serrano et al., 2012), in particular, it does not fit all monthly streamflow data in

all river grid cells (n= 29,000). For example, sample properties of streamflow in January might differ from those in August in each of the river grid cells (Tijdeman et al., 2020). In summary, we obtained a gamma distribution parameter set for each river grid cell and month (in total $>$ 348,000 sets).

    A 1-month accumulation period was used in this study (SSI-1 drought). Longer accumulation periods, e.g. SSI with 6-month accumulation period (SSI-6), as it was used in Trambauer et al. (2015) and Barker et al. (2016), were not selected in our study,

since streamflow already comprises some catchment memory aspects (delayed flow from groundwater). Nevertheless, we need to realize that anomalies in the accumulated flow over a longer period (e.g. SSI-6) have relevance for some purposes, such as

the management of surface water reservoirs. Negative SSI values indicate a drought event, which means that the streamflow in a certain month is lower than the median streamflow of that month. Four SSI classes are commonly distinguished, which are: 1) mild drought: $0 > SSI \geq -1$, 2) moderate drought: $-1 > SSI \geq -1.5$, 3) severe drought: $-1.5 > SSI \geq -2$, and 4) extreme drought: SSI$<$-2 (Nalbantis and Tsakiris , 2009). In this study, however, we assumed the drought event to start when the SSI-1 falls below -0.84. The use of a limit value -0.84 for SSI warrants a fair comparison between the threshold approaches (Q80) and SSI-1 (SSI$<$-0.84) (Tijdeman et al., 2020). The above-mentioned gamma distribution parameter sets for SSI-1 and the limit value of -0.84 were used to identify drought characteristics (Section 2.3) in the historic period (1990-2018) in each of the river grid cells of the pan-European river network.

To forecast a possible SSI-1 drought for a lead-time (LT) of x-month (x = 1, 2, . . . ,7 months), we also used the above-mentioned gamma distribution parameters sets and limit value of -0.84. The SSI-1 times series were derived from the forecasted streamflow using these parameter sets (Sutanto et al., 2020a). For example, to forecast SSI-1 using forecasted streamflow initiated on January 2003 with a lead-time of 7-month, we calculate the SSI-1 for January (LT=1), February (LT=2), up to July (LT=7) using the parameter sets from January, February, up to July, respectively. Same parameter sets were applied to each ensemble member to calculate 25 ensembles of SSI-1.

## 2.3 Drought characteristics

Drought analysis using the threshold methods and the standardized approach shares several common major drought properties or characteristics, which are the number of drought occurrences/frequency (N), drought initiation time or timing (T), and drought duration (D). Another drought characteristics, namely drought deficit volume (DV), can be obtained only by using the threshold methods. The standardized approaches cannot be used to calculate the deficit volume, because it only provides information on the drought severity class (Section 2.2.2) and not about the amount of water that is not available during a drought event ($m^3$ in our case for streamflow). In this study, the number of drought occurrences, timing, duration, and deficit volume will be calculated using the threshold methods (VTD, FTD, VTM, and FTM). For the SSI-1 the same characteristics will be determined, except the deficit volume.

The number of drought occurrences (N) shows how many drought events occurred: 1) from October 1990 to September 2018 (hydrologic years), and 2) from the starting date of the forecast up to 215 days (7 months) ahead. The timing/onset (T) for drought was determined based on the starting month of each drought event (1: Jan, 2: Feb,. . . , 12: Dec) in the time series either in the 28 years (historic analysis) or in the median of the ensembles of 215 days (7 months). If there is more than one drought event in the time series, which is common for the historic data, then we select the timing based on the starting month with the highest frequency. If there is more than two starting months with the same frequency, then we calculate the median value from the selected timings. For example, the month August is selected as drought timing, if months March, August, and October have the same frequency. If there are two starting months detected with the same frequency, then we chose the first timing. Drought duration, expressed in day for VTD and FTD and month for VTM and FTM, is the number of day/month when the streamflow or SSI is continuously below the threshold or limit value, respectively. If there is more than one drought event, then we average the duration of the events. The drought deficit volume (only threshold methods) is calculated by first,

converting the unit into $m^3 \ d^{-1}$ by multiplying the streamflow with 86,400 and second, summing up the difference between streamflow and the threshold level per day/month over the drought event, expressed in $m^3$. For total drought deficit volume, we simply sum up deficit volumes from all drought events (either historic period or forecast period) and we divide it by the number of events to obtain the average drought deficit. In case of an ongoing drought in the forecast, e.g. a drought that already started

prior to the forecast initiation, we determine the drought characteristics from the first day of the forecast. We do not consider what happened before. In case a drought still has not ended by the end of the forecast period (at day 215 or month 7), we break the drought event by the end of the forecast, meaning that we do not take into account the characteristics of the drought event beyond the forecast period. In addition, we also provide a maximum number of ensemble members indicating drought (Ne) for each forecast initiation as a percentage. For example, for forecasts initiated in July 2003 with LT=7 month (from July 2003 to

January 2004), we calculate Ne for every LT (1, 2, . . . ,7 months), and provide information only for the maximum number of Ne.

## 2.4  Köppen-Geiger climate classification

The Köppen-Geiger climate classification has been built based on observed global temperature and precipitation data used in Peel et al. (2007). There are four main climate types found in Europe, which are cold (D), arid (B), temperate (C), and polar

(E). Each climate type can be classified into several sub-climate types. In our study area, the dominant sub-climate types are Bsk, Csa, Cfa, Cfb, Csb, Dfc, Dfb, Dsa, Dfa, Dsc, and ET (Fig. 1). Six sub-climate types are considered in the streamflow drought analysis across entire Europe, that is, the tundra climate (ET), the, warm-summer, humid continental climate (Dfb), the subarctic climate (Dfc), the temperate, oceanic climate (Cfb), the cold, semi-arid climate (Bsk), and the hot summer Mediterranean climate (Csa). The latter two types are clustered in the Mediterranean climate (Med). This means five climate

regions that cover over 90% of the European area. In addition to the analysis of streamflow drought in all grid cells of the pan-European river network, four different rivers located in the major climate regimes of Europe were selected. The locations of the selected rivers are as follow: 1) Rhine River near Cologne, Germany, located at 50.9°N and 6.9°E (Cfb), 2) Danube River near Budapest, Hungary, located at 46.9°N and 18.9°E (Dfb), 3) Vuoksi River close to the Finnish-Russian border, located at 61.1°N and 28.8°E (Dfc), and 4) Ebro River near Asco, Spain, located at 41.2°N and 0.6°E (Bsk). The four rivers are indicated

by red dots in Figure 1. For detailed information about climate classifications used in the study, see the Köppen-Geiger climate classification presented in Peel et al. (2007).

## 3  Results and discussion

We present the differences of streamflow droughts identified using different drought identification approaches in two parts. The first part provides results and discusses the historical analysis that consists of the investigation of differences between drought

analyzed using different approaches (i.e. drought definitions). The analysis was performed in terms of drought characteristics both in over 29,000 river grid cells at the pan European scale, and in four selected river basins in more detail (Section 3.1). The

second part elaborates the implication of streamflow drought forecasting using different definitions at the pan European scale and in one of the selected river basins (Section 3.2).

## 3.1 Historic analysis

### 3.1.1 Streamflow drought characteristics across Europe

One of the most profound differences among streamflow droughts using different identification approaches is the occurrence of these events. In a river grid cell, streamflow drought may be absent, occur once, or even more than once in a hydrological year throughout the period 1990-2018.

The largest deviation between drought occurrences obtained with the five different identification approaches is due to the temporal resolution. In entire Europe, the variable threshold using daily data (VTD) detects almost 50% more drought events than when applying monthly data (VTM), i.e. 49.6 and 26.6 events, respectively (Table 1 and 2). The spatial distribution also shows this clearly (Fig. 2a and 2c). The deviation between the daily and monthly resolution for the whole of Europe is smaller (about 25%) when fixed threshold approaches are applied (FTD: 39.6 and FTM: 28.6 events), see also Fig. 2b and 2d. The data also show that when a daily resolution is used, the VTD method identifies about 25% more events than the FTD method (Table 1, Fig. 2a, and 2b), whereas deviations are small at the monthly scale (VTM versus FTM, Table 2, Fig. 2c, and 2d). At the pan-European scale, there are no substantial differences between drought occurrences (<15%) derived with the methods using monthly data (VTM, FTM and SSI-1, Table 2, Fig. 2c, 2d, and 2e).

The maps (Fig. 2a and 2b) show that in about 20% (VTD approach) and 5% (FTD approach) of the pan-European river grid cells, streamflow drought on average occurs at least twice a year (>60 events). However, in parts of Sweden and Finland, and southeast Europe, such as Romania, Serbia, and Bulgaria, the occurrence of VTD and FTD droughts does not exceed 30 events during the study period. The highest number of droughts is identified in the temperate oceanic climate (Cfb), whereas the lowest is found in the Mediterranean climate region (Med), irrespective of the identification approach. Clearly, a number of drought occurrences vary amongst identification methods, for example, the range for the Cfb and Med climates is 30.4-57.8 and 22.6-41.0 events, respectively (Table 1 and 2).

Minor drought events are assumed to be the main reason for the high occurrence of VTD and FTD droughts in the major European rivers (>60 events), compared to VTM and FTM droughts (Fig. 2a, 2b, 2c, and 2d). To prove our hypothesis, we plotted the percentage of VTD drought events that have duration of shorter than 30 days (Fig. B1). Here, it can be seen that many rivers in the west and east Europe (Cfb and Dfb climates), as well as, the mountainous regions in Norway (Dfc and ET), experience lots of minor drought events (>60% of total number, and even more, up to almost 100% in a few rivers indicated by red color). This means that if we would exclude the number of VTD events shorter than 1 month in these regions, the number of drought occurrences is lower than obtained with the VTM approach. Mediterranean and Dfc climate regions (Sweden and Finland), in general, show a smaller number of minor drought events (~30% of total), meaning that drought events in these regions (Fig. 2a) are caused by droughts that have a long duration. This will be discussed later (Fig. B2).

To investigate the timing of streamflow drought, we present the month when drought mostly starts in each grid cell of European rivers (Fig. 3). The timing was determined for each drought event in the period October 1990 to September 2018. Figure 3 indicates that, as expected, there is a strong relation between streamflow drought timing in the rivers and the Köppen-Geiger climate regions across Europe (compare Fig. 1 and 3). This also differs among drought identification methods. In general, the fixed threshold methods (FTD and FTM) detect earlier drought (Table 1 and 2) than the variable threshold methods (VTD and VTM), except in many rivers located in the humid continental climate (Dfb). Rivers located in cold climate regions (Dfb and Dfc), such as northern and eastern Europe, and the Alps, experience streamflow drought events in late winter and early spring (March-April) when the daily variable threshold method (VTD) is applied (Fig. 3a, Table 1), and later when monthly data (VTM) are used (May-July, Fig. 3c, Table 2). In addition to below normal precipitation and above normal evaporation (classical rainfall deficit drought), drought in cold regions also depends on the length of the frost period and the timing of snow incidents, accumulation, and melting (cold snow season drought) (van Lanen et al., 2004; Pfister et al., 2006; van Loon and van Lanen , 2012). A warm snow season drought may also occur during spring or summer, associated with no snow occurrence during winter or earlier snowmelt than normal (van Lanen et al., 2004; van Loon et al. , 2010). This causes an early peak in streamflow, resulting in lower streamflow in late spring and summer. In the warmer climates (Cfb and Med) droughts start later (mostly July-October) than in the colder regions. However, there is a difference between variable and fixed threshold approaches, i.e. FTD and FTM droughts largely begin earlier (July-August, Fig. 3b and 3d) than the VTD and VTM droughts (September-October, Fig. 3a and 3b, Table 1 and 2). The start of SSI-1 drought in most climates is closest to VTM droughts (Fig. 3e, 3c, and 3d, Table 2).

The average duration of the droughts (Fig. B2) is negatively correlated with the number of drought occurrences (Fig. 2). We have seen that applying methods using daily data result in more drought occurrences than those that use monthly data. Hence, the average drought duration of events is connected with the temporal resolution of the methods. We have seen that droughts obtained with methods fed by daily data (Fig. 2a and 2b) are shorter than those applying monthly data (Fig. 2c and 2d). For instance, for the whole pan-European river network, VTD droughts are about 40% shorter than VTM droughts (44.6 days and 2.4 months/73 days, respectively, Table 1 and 2). For the fixed threshold drought, the following average drought duration was found: FTD 56.0 days and FTM 2.5 months/74 days, implying that the FTD droughts are about 25% shorter than FTM events. We also observed that rivers in the Cfb climate have the highest number of droughts and those in the Mediterranean climate region have the lowest number of droughts, implying that the average drought duration in the Cfb climate is shorter (36.4 and 47.2 days, Table 1, and 1.9 months/57 days, and 2.2 months/66 days, Table 2) than in the Mediterranean region (56.3 and 68.4 days, Table 1, and 2.9 months/87 days, and 2.7 months/81 days, Table 2), see also Fig. B2a, B2b, B2cs and B2d. The average drought duration estimated with the SSI-1 approach is close to both the VTM and FTM methods (Fig. B2c, B2d, and B2e). Differences in average drought duration amongst methods using monthly data for the whole of Europe are around 10% (Table 2).

The average drought deficit volume that has been detected by the different drought identification methods is to some extent linked to the temporal resolution of the methods. For example, for the whole of Europe, we found higher average drought deficits with the approaches using monthly data (VTM: 118.5M $m^3$ and FTM: 104.7M $m^3$) than those fed by daily data (VTD:

79.4 M m$^3$ and FTD: 78.8M m$^3$), indicating about 25-30% higher drought deficit volumes (Table 1 and 2). Plotting average drought deficit volume across European rivers (Fig. B3), in general, shows higher deficit volumes for the bigger rivers in central and north Europe (except coastal areas), which is partly caused by not standardizing the deficit volumes. Hence, the analysis of the drought deficit volume using different identification approaches is more meaningful, if we summarize the results for each climate region (Table 1 and 2) or for selected river grid cells (Section 3.1.2). The highest deficit volume is found in the humid continental climate (Dfb) and the lowest in the Mediterranean climate, irrespective of the identification method (Table 1 and 2), although the deficit volumes differ per method. The difference in average river basin sizes located in different climate regions also contributes to disparities in deficit volume.

The pan-European analysis of the river network (Table 1 and 2, Fig. 2, 3, B2, and B3) evidently demonstrates that drought characteristics (occurrence, timing, average duration, average deficit volume) determined by commonly applied identification methods (variable threshold versus fixed threshold, daily versus monthly resolution, threshold versus standardized approach) are different. The differences are also dependent on the climate region.

### 3.1.2 Drought occurrences in selected rivers and periods

For a more detailed analysis of the differences of streamflow droughts derived from different approaches, as illustrated above for the whole pan-European network, we investigated four rivers situated in main climates across Europe (Fig. 1) for particular periods. The pan-European analysis focused on the spatial aspects of the differences between the drought identification methods, whereas the detailed analysis of the four selected rivers emphasizes the temporal aspects. Figure 4 and 5 show for some selected years a detailed analysis of drought in the rivers. The observed streamflow (30DMA hydrograph) of the period 2000-2004 from the Rhine River in combination with the daily threshold methods (VTD and FTD) clearly show that streamflow drought mainly occurred from summer 2003 to January 2004 (Fig. 4a). The year 2003 is one of the most notable drought years in Europe (Fink et al., 2006; Ionita et al. , 2017; Laaha et al., 2017). During wet years, e.g. from 2000 to 2002, there were no streamflow drought events identified (both VTD and FTD). The difference in drought occurrence in the Rhine River in the selected 5-year period between the daily methods is small, for example, there are a few minor droughts detected (early 2003 summer, December 2004) with the VTD, whereas these were not found with the FTD. The deficit volume of the drought event in summer 2003 was clearly larger for the FTD than of the VTD. In the winter of 2003-2004, the opposite happened (Fig. 4a). The different identification approaches using monthly data (VTM, FTM, and SSI-1) also detected the 2003 drought as the major event in the 2000-2004 time series (Fig. 5a), which terminated in October due to some precipitation. Some minor drought events were identified in autumn and winter 2003 with all three methods, although the timing was different. For instance, the VTM and SSI-1 droughts were later than the FTM drought (Fig. 5a).

A difference between drought identification approaches using daily and monthly drought methods is clearly seen in the Danube River (Fig. 4b and 5b). Many minor drought events were recognized using daily data, that is, in winters from 2000 to 2002, spring 2003 and 2004 for FTD, and in spring 2003, spring 2004, summer 2004, and winter 2004/2005 for VTD. In contrast, minor drought events in winter 2001/2002 and in spring 2003 did not occur if we applied drought identification approaches using monthly data (FTM and VTM, respectively, Fig. 5b). Figure 4b and 5b demonstrate that during rather wet

years (the year 2000-2002), no VTD, VTM, and SSI-1 droughts were observed. The VT and SSI-1 approaches take into account seasonality in their analyses. Similar to the Rhine River, in the Danube, a major drought event in 2003 was identified using all
360 approaches (Figure 4b and 5b).

In the Vuoksi River, which is located in the cold climate region (Dfc, Fig. 1), all drought identification approaches show more or less similar drought occurrences (Fig. 4c and 5c). The fixed threshold approaches, both at the daily and monthly scale (FTD and FTM), detect slightly more events than those that consider seasonality (VTD, VTM, and SSI-1). Two multi-year drought events were detected in 1999-2000 and 2002-2003 with all drought approaches. The main reason for this is that there
is only a small difference between daily and monthly streamflow. The presence of water bodies, such as lakes, causes daily streamflow not to be highly variable in short term. This attenuates and damps the streamflow response to the driving force, i.e. precipitation, including snowmelt, and is thus driven by longer-term previous hydrological conditions (Pechlivanidis et al., 2020).

In the first decade of the 21$^{st}$ Century, climate variability in the Mediterranean regions caused different wet/dry periods
compared to the rest of Europe. In contrast to the severe 2003 drought in central and west Europe, the most severe droughts in, e.g. Catalonia (Spain), were observed from 2005 to 2008 (Martin-Ortega et al., 2012; March et al., 2013), which is illustrated by the streamflow of the Ebro River (Fig. 4d and 5d). Pronounced FTD droughts occurred every year in the period 2005-2009, whereas only minor VTD drought occurred in the last year (Fig, 4d). Using monthly instead of daily streamflow data reveals a similar pattern (Fig. 5d), i.e. no VTM droughts from summer 2008 to 2009, while these happened in all summers according
to the FTM method. The droughts in 2005-2007 also illustrate differences in timing between VT and FT methods, both at the daily and monthly scale, i.e. limited coinciding periods (orange-shaded in Fig. 4d and 5d). As expected, the SSI-1 droughts follow the pattern of VTM droughts because both metrics consider seasonality. A multi-year drought event from summer 2007 to spring 2008 was identified with all approaches, although duration is different (e.g. FTD and FTM lasted longer than VTD and VTM, as well as SSI-1 drought). Another major drought event in the Ebro was observed in 2005 (Vicente-Serrano et al.,
2012). In contrast to the 2007-2008 drought in this year, the Ebro River experienced considerably longer VTD, VTM, and SSI-1 droughts than FTD and FTM droughts.

Above we explained differences in drought characteristics derived from different identification methods for the four selected rivers for a 5-year period. A summary of the outcome from all five drought identification methods for the four selected rivers and all hydrological years (1991-2018) is presented in the Supplementary Material (Supplementary Table S1 and Table S2).

### 3.1.3 Summary of differences between drought identification approaches

The more detailed drought analysis of the four selected rivers in the previous section and the broader analysis of the pan-European river network (Section 3.1.1, Table 1 and 2) show that the FTD approach identifies a lower number of drought occurrences than the VTD. On the other hand, when monthly approaches are used to detect drought, the FTM approach results in slightly more droughts than the VTM. Clearly, relative differences are smaller than for the daily resolution. Sarailidis et
al. (2019) found for the Yermasoyia catchment (Cyprus) a smaller number of drought occurrences both at the monthly and daily resolution when applying the fixed threshold instead of the variable threshold, which is in line with our daily results

(FTD versus VTD). Overall, early droughts were identified using the fixed threshold methods, irrespective of the temporal resolution (FTD and FTM). Rivers located in the Dfb climate, however, have later FTD and FTM droughts than the VTD and VTM droughts. The FTD identifies longer droughts than the VTD, whereas the differences in average duration when using the monthly resolution (FTM and VTM) are small. Our findings on drought duration at the daily time scale are also found by Heudorfer and Stahl (2017) in a study dealing with four case catchments in Germany. In the pan-European analysis (Table 1 and 2), we found that the drought deficit volume obtained with the variable threshold methods (VTD and VTM) is slightly higher than with the fixed threshold methods (FTD and FTM). This is confirmed by a study done by Sung and Chung (2014) for the Seomjin River basin in Korea. Not all four selected rivers follow this pattern, for instance, in the Rhine and Danube, application of the fixed threshold methods results in higher deficit volumes than the variable threshold methods (Supplementary Table S1 and S2), which is also found by Sarailidis et al. (2019) in the Yermasoyia catchment. Obviously, individual rivers may deviate from the general pattern. Our generic finding that the streamflow drought characteristics (frequency, duration, timing) derived using different identification methods differ is in line with the observations made by Vidal et al. (2010). Their study in France also concluded that different identification methods (only standardized-based indices at multiple time scales) generate different drought characteristics.

## 3.2 Implication of different drought identification approaches to forecast streamflow drought

So far, this paper has focused on a historical drought analysis using different identification approaches, which creates a base for the implications of these findings for the forecasting of streamflow drought. First, we illustrate the implications at the pan-European scale with focus at the spatial aspects followed by a more detailed temporal analysis for the Rhine River, as one of the four selected rivers above. The 2003 drought is used as an example.

### 3.2.1 Forecasting streamflow drought characteristics across Europe

Consequences of using different drought identification approaches to forecast streamflow drought characteristics across Europe are described in this section. The forecast initiated in the first of July 2003 (median of 25 ensemble members) for 7 months ahead (up to January 2004, see Section 2.3 for the calculation of drought characteristics using forecast data) is used for illustration. We show the forecasted drought duration and timing here (Fig. 6 and 7, respectively), while the forecasted frequency of drought occurrences and drought deficit volumes are provided in appendix C.

Figure 6 shows the forecasted average drought duration in Europe using the forecast initiated in July 2003 for a 7-month lead time (July 2003-Jan 2004). Longer drought duration is forecasted in many European rivers using the fixed threshold approaches (FTD and FTM, Fig. 6b and 6d) than the variable threshold approaches (VTD and VTM, Fig. 6a and 6b), up to 60 days/2 months, which for the daily resolution was expected based on the historic analysis (up to ~20%, Table 1). The SSI-1 approach forecasts similar drought duration to VTM (Fig. 6e and 6c). Using the VT and SSI-1 approaches, drought was forecasted to last on average 40 days or ~1 month in the period July 2003-Jan 2004 in many European rivers. The fixed threshold approaches predict an average duration of 120 days or ~4 months. Rivers located in Eastern European countries, such as in Belarus, Ukraine, and Romania were predicted to have a long drought duration in the above-mentioned period according

to all approaches (up to 200 days, 7 months, Fig. 6). This region (the eastern part of Europe) is identified as an area suffering from severe hydrological drought hazards, where the frequency of drought is small compared to other European regions, but with the drawback that droughts last long (Sutanto and Van Lanen , 2020). The forecasted average drought durations in the Cfb, Dfc, and Mediterranean climates using the fixed threshold approaches are around 100 days for FTD and 3-4 months for FTM (Fig. 6b and 6d, respectively), which is almost threefold longer than obtained with the variable threshold approaches.

Figure C1 presents the forecasted number of drought occurrences from July 2003 to January 2004 (LT=7) derived from different drought approaches. In general, the VTD, VTM, and SSI-1 approaches forecast that at least one drought event would occur in lots of river grid cells in Europe (∼80%, Fig. C1a, C1c, and C1e), which is lower than the number of droughts forecasted with the FT approaches (∼90%, Fig. C1b and C1d). These differences in the number of drought occurrences between identification approaches are not uniformly distributed over Europe. For instance, in the Cfb and Dfb climates, the opposite is found, that is, the variable threshold methods forecast higher drought occurrence than the fixed threshold methods. The differences in the number of drought occurrences between the identification approaches highlight the importance of considering whether seasonality should be taken into account (the variable threshold and SSI-1 droughts) in the forecasting, or not (the fixed threshold droughts).

The forecasted drought start in the period July 2003-Jan 2004 (month that the first drought appears, see Section 2.3 for the determination of the start month/timing) using the VTD, VTM, and SSI-1 is, in general, later than of the FTD and FTM approaches, except in the cold regions, such as Dfc and ET (Fig. 7). In the Cfb, Dfb, and Mediterranean climates, the VTD, VTM, and SSI-1 approaches predict the drought timing in September to December (Fig. 7a, 7c, and 7e), while the start of the forecasted FTD and FTM droughts is earlier, i.e. July to September (Fig. 7b and 7d). It is vice versa for the Dfc region (Sweden and Finland), where forecasted VTD, VTM, and SSI-1 droughts are earlier (July) than FTD and FTM droughts (December and January). Higher drought deficit volume than 170M m$^3$ is predicted for the period July 2003-Jan 2004 in many European rivers using the FTD and FTM approaches than those predicted with the VTM ones (Fig. C2). An exception is seen for some big rivers flowing through Hungary, Ukraine, Romania, and Bulgaria. Both variable and fixed threshold droughts have a high deficit volume predicted there, because of the long drought durations (Fig. 6).

### 3.2.2 Forecasted drought characteristics for the Rhine River

In the previous section we have dealt with streamflow forecasting for the pan-European river network that mainly focusses on spatial aspects. Here, we concentrate more on the temporal aspects, and use the Rhine River as an example. Figure 8 illustrates the observed and forecasted 25 ensemble streamflows (grey shaded area) in the Rhine River (location 1, Fig. 1) initiated in April and July 2003 for 7 months ahead and the forecasted median ensemble streamflow (purple line) using all drought identification methods. In addition, the forecasted droughts in streamflow are given (shaded areas below thresholds). We choose the 7-month forecast initiated in April 2003 (Fig. 8a, 8c, and 8e) covering spring, summer, and autumn to explore if the forecasts obtained with different identification methods are able to predict drought that occurred in summer 2003. July 2003 was chosen because streamflow drought based on observations was starting in this month (see Fig. 4a). VTD and FTD drought forecasts done in April using the median ensemble identify a minor drought that occurred in April (orange area) and from August to October

only for FTD (red areas), i.e. the purple line is below the blue (VTD) and the red line (FTD) (Fig. 8a). The forecast done in
April (Fig. 8a) also shows that some dry ensemble members predict two long-lasting droughts, both for FTD and VTD, that
is, from mid April to early June, and from August to the end of the forecast record (October). On the other hand, some other
ensemble members do not predict any drought at all in the April-October forecast record. VTD and FTD drought forecasts
done in July using the median ensemble identify minor drought events that would occur in July and November (orange areas),
whereas a major FTD drought would happen from the end of July to the end of October (Fig. 8b). In general, the FTD method
forecasts more drought events in 2003 than the VTD (Fig. 8a and 8b).

In contrast to the daily threshold approaches, drought forecasts done in April 2003 using the monthly drought identification
approaches, VTM, FTM, and SSI-1, do not forecast a drought event that would occur in summer (Fig. 8c and 8e, see Fig. 5a for
observed drought). A minor drought event is predicted with the FTM method by the end of the forecasts, which is September
(red shaded area). Monthly drought forecasts done in July 2003 predict a FTM drought from August to the end of September
(Fig. 8d). This indicates that all the forecast approaches miss the ongoing drought event in July, as it was observed (Fig. 5a).
The VTM approach, on the other hand, does not predict any drought event, whereas the SSI-1 forecasts a minor drought event
in the beginning of July, but no other droughts later in 2003 are forecasted (Fig. 8f).

In general, drought events (i.e. occurrence) can relatively be well forecasted using the median of ensemble members, but
this holds to a lesser extent to other drought characteristics, such as severity, duration, and deficit volume. Additional metrics
than the median, such as $25^{th}$ and $10^{th}$ percentiles taken from the ensembles, must also be considered for drought forecasting,
as done by Sutanto et al. (2020a). Figure 8 clearly demonstrates that the observed streamflow is placed in between the lowest
ensemble member and the ensemble median during a severe drought event, as the 2003 drought. Irrespective of the skill, the
forecasts of the drought for the Rhine River show that predicted drought characteristics very much depend on the identification
method, including the temporal resolution (daily versus monthly).

For a better overview of forecasted drought characteristics in the Rhine River than in Figure 8, we summarize all 7-month
forecast results done from January 2003 to December 2003 in Table 3 for daily drought approaches (VTD and FTD) and in
Table 4 for monthly drought approaches (VTM, FTM, and SSI-1) using the median of the ensemble members. This implies that
not only the April and July forecasts are considered, as done in Figure 8, but also forecasts done in all the other months of 2003,
meaning that the December forecast covers the first six months of 2004. The forecasts initiated in January, February, and March
using daily data (Table 3) did not predict any drought event in 2003 (except for some ensemble members, Ne>0). Droughts
were predicted not earlier than forecasts issued in April. In April, three minor VTD droughts and nine minor FTD droughts were
predicted to occur with average drought duration of 2.7 and 6.1 days, respectively. The timing shows that droughts will start
in April (VTD) and September (FTD). For drought events forecasted in April using the VTD method, the maximum number
of ensemble members (Ne) foreseeing these three minor drought events in the period April-October is 88% (22 members out
of 25 fall below the threshold). The FTD method shows the same number of members (22 members, 88%). The number of
ensemble members in drought (Ne in %) can be used as a measure for drought forecast uncertainty or the forecast confidence
level. The higher the percentage, the more likely the drought will occur (higher confidence level). In our case, VTD and FTD
droughts were predicted to occur in the Rhine River with a high confidence level (Ne=100%) starting from July until at least

the end of the year 2003. The forecast issued in July 2003 predicts the highest number of VTD drought events up to 14 events with an average duration of 2.8 days. The FTD method shows a lower drought frequency (9 events) but with a longer average duration (12.7 days). The longest average drought duration was predicted by the VTD forecast initiated in October (24 days) and in June for FTD (28.7 days). For the drought deficit volume, the highest water deficit was predicted using forecast initiated in October and December for VTD (388.9M m$^3$ and 920.5M m$^3$, respectively) and in August and December for FTD (507.8M m$^3$ and 1,272M m$^3$, respectively).

The monthly drought approaches, on the other hand, show different results from the daily approaches for most of the characteristics (compare Table 3 and 4). The FTM method predicts one drought event in each of the forecasts initiated from April to December with medium to high confidence (Ne>50%). The VTM method foresees one or two events from the May forecast onwards with a medium to high confidence level (Ne>50%). The SSI-1 method starts predicting droughts two or three months later than the threshold approaches. The longest predicted VTM drought (two months) and most severe (total deficit volume: 882.1M m$^3$) was done by the forecast initiated in September. The longest FTM drought was predicted (up to 4 months) with the forecast initiated in August. This FTM drought also has the highest drought deficit volume (2,841M m$^3$). The analysis of the forecasts from January to December for the Rhine River (Table 3 and 4) clearly shows that forecasted drought characteristics depend on the identification method.

In this study, we highlight the occurrence of minor droughts derived with the daily threshold methods (VTD and FTD) as a reason for the high drought frequency (Fig. 2a, Fig. 2b, Fig. 4, Fig. 8, Fig. B1, Fig. C1, Table 1, and Table 3). A high number of VTD minor droughts with short duration and small deficit volume may disturb drought analysis. Tallaksen et al. (1997) and Fleig et al. (2006) suggest several pooling procedures to reduce the number of minor droughts, such as applying the inter-event time method (IT-method), the moving average procedure (used in this study), and the sequent peak algorithm (SPA). They state that minor droughts are automatically filtered out when the moving average procedure is applied. In our study, however, this only happens to a certain extent. As expected, when we would not have applied the 30DMA, the number of drought occurrences would have been higher, i.e. in this case by a factor of three (Fig. 2a and B4).

In addition to these pooling techniques, the exclusion of drought events with duration shorter than a given number of days is recommended (Jakubowski and Radczuk , 2004; van Loon et al. , 2012). For example, van Loon et al. (2012) excluded droughts that have duration less than three days, van Loon and van Lanen (2012) excluded droughts that have duration fewer than 15 days, and some studies excluded droughts that have duration less than five days (Hisdal et al. , 2004; Birkel , 2005; Fleig et al., 2006). In the end, the choice to exclude drought events shorter than a particular number of days to avoid minor droughts in the drought analysis, is a matter of subjectivity. We showed in our analysis that if we would have excluded drought <30 days (Fig. B1), the number of drought occurrences in most of the European rivers would decrease by 60% (Section 3.1.1) In this study, although we excluded many minor drought events by applying moving average procedures, which is the 30DMA for drought analyses (Appendix A), minor drought events are still there (Fig. B1). In this study, we did not apply pooling procedures, as mentioned above, besides the 30DMA.

Our results (historical and reforecast) reveal that each drought identification approach has strengths and weaknesses. The approaches using a daily resolution (VTD and FTD) identify more minor drought events than the rest (VTM, FTM, and SSI),

which may be not relevant for all end-users, including those studying drought characteristics (Tallaksen et al. , 1997; Fleig et al., 2006). In close cooperation with end-users right from the start, the developer of a DEWS, which includes daily approaches, can rather easily exclude drought events that have a short duration, e.g. by applying a pooling technique, as discussed above (including keeping out events < n days). Approaches based on the variable threshold (VTD and VTM) and standardized methods (e.g. SSI-1) have the advantage that these consider seasonality compared to the fixed threshold approaches (FTD and FTM) (Fleig et al., 2006; van Loon and van Lanen , 2012). VTD, VTM and SSI-1 allow to detect droughts that might cause impacts in the normally high flow season. The use of monthly drought threshold methods, such as VTM and FTM are an option if daily data is not available. About 40% of the global flow gauging stations has only monthly data. In Asia the number is even higher (72%) and these stations have 85% longer time series of monthly observation data than those with daily data (GRDC, https://www.bafg.de/SharedDocs/Bilder/Bilder_GRDC/summary_stat.png?__blob=poster). In addition, the approaches using monthly resolution enable better identification of major drought events that last from a month to (sub)season. An occurrence of a short-lived precipitation event may split one major VTD or FTD drought event into two smaller events. A clear example is seen in the Rhine River during major 2003-2004 drought. Unlike the FTM drought that occurred continuously from summer to winter 2003-2004, VTD and FTD drought events were shortly interrupted in November (Figure 4a and 5a). Furthermore, one should note that the drought deficit volume derived from monthly threshold methods only approaches the actual water deficit, because it is calculated from the difference between monthly streamflow and the threshold (Section 2.3). In reality, the number of days with streamflow below the threshold may be less than a month. The use of daily threshold approaches, such as VTD and FTD, is highly recommended to accurately calculate drought deficit volume. As mentioned in previous sections, the end-user should be aware that standardized drought approaches, e.g. SSI, cannot be used to determine the drought deficit volume. If the deficit volume is not required and end-users decide to use the standardized approach, they should investigate if the gamma probability distribution best describes their observations. Under some conditions for smaller and environmentally more homogeneous regions, other distributions (e.g. Generalized Extreme Value) might perform better (Section 2.2.2).

In the end, we recommend that a DEWS should provide information on upcoming drought events using diverse identification approaches to target different end-users, as illustrated in this study. The information should include the different forecasts themselves, but also background on how these are obtained (e.g. factsheets) and how these can be used. Some users from a specific possibly impacted sector (e.g. forestry) will only use a small part of the forecast products, whereas other users with a broader duty will use the full suite of products. For example, water supply or hydroelectric companies are advised to use forecasts using daily threshold methods (VTD and FTD) because these can identify the drought deficit volume on a daily basis. Forecast products based on the variable threshold approach can be of interest for them for the reason that they also deal with below normal conditions in the wet/high flow season. Environmental agencies or navigation authorities might be more interested in products based on the daily threshold (FTD) since they have to control maintaining ecological minimum flow or warning water-borne transport of low river stages, respectively. The use of forecast products derived from monthly approaches (VTM, FTM, and SSI) may be more relevant for the forestry sector, operational hydrometeorological services, agencies dealing with water resources management and planning, and policy makers since they focus more on drought forecasts at multi-monthly or seasonal time horizons. The deep-rooted plants in forests are not highly impacted by short drought duration because they

can extract water from deeper layers (Miguez-Macho and Fan, 2012; Richard et al., 2013). Moreover, the aggregation of daily
streamflow data into monthly enhances forecast skill, which is important for drought mitigation planners and policy makers,
as it is found for streamflow forecasts (e.g. see Wetterhall and Di Giuseppe (2018) for daily resolution and Arnal et al. (2018)
for monthly resolution).

## 4 Conclusions

Streamflow drought forecasting may use different identification approaches to detect drought events, i.e. threshold and stan-
dardized approaches. This study presents a drought analysis using simulated historical streamflow data from the pan-European
rivers network. It consists of almost 30,000 river grid cells and is located in different climates across Europe. We applied
commonly identification approaches, which are the daily Variable Threshold (VTD), the daily Fixed Threshold (FTD), and
the Standardized Streamflow Index (SSI-1) that uses aggregated streamflow over a month. In addition, we also provide results
derived from monthly threshold approaches (VTM and FTM) for a fair comparison with the SSI-1 drought. These approaches
generate several drought characteristics, namely drought occurrence, duration, timing, and deficit volume (latter not for SSI-1).
The largest difference amongst the drought identification approaches comes from the temporal resolution. When using the
same drought identification approach (variable or fixed threshold methods), but using different data aggregation levels (daily
versus monthly), the daily methods evidently generate more drought occurrences. The daily variable threshold method (VTD)
detects almost twice as many drought events as the monthly method (VTM). The FTD also identifies more drought events than
the FTM, but deviation is smaller (about 25%). Minor droughts shorter than 1 month are the main reason for the higher number
of drought occurrences identified by the daily threshold methods (VTD and FTD). The number of drought occurrences derived
from the VTD approach is higher than obtained with the FTD, whereas the differences amongst methods using monthly data
(VTM, FTM and SSI-1) is rather small (<15%).

Identification of streamflow droughts using different methods also affects timing, i.e. the month in which the drought starts.
Differences are also controlled by climate regions. In general, the fixed threshold methods (FTD and FTM) detect earlier
drought than the variable threshold methods (VTD and VTM), except many rivers in the humid continental climate (Dfb).
Rivers located in cold climate regions (ET, Dfb and Dfc) experience streamflow drought events in late winter and early spring
(March-April) when the daily variable threshold method (VTD) is applied, and later when monthly data (VTM) are used (May-
July). When using the fixed threshold methods not such clear pattern in the start of the drought in the cold climates was found
(FTD: February-June, and FTM: February-July). Drought in the Mediterranean climate mostly starts late, in late summer or
autumn (August-October), irrespective of the identification method. The start of SSI-1 droughts is closest to VTM droughts
because both methods use a monthly resolution and consider seasonality. Average drought duration for the threshold methods
is more controlled by the number of occurrences (i.e. negatively correlated). This implies that the drought duration obtained
with the daily threshold methods (VTD and FTD) is shorter than derived from the monthly methods (VTD and VTM), and
that the FTD droughts last longer than VTD droughts. In addition, the methods using daily data produce drought events with
lower drought deficit volumes (25-30%) than the methods fed with monthly data. It is important to note that the findings for

the whole pan-Europe river network are generic. Individual rivers, as illustrated with some selected rivers, may deviate from the general pattern.

The different drought identification approaches were also applied to streamflow forecasting with the 2003 drought as an example, which yielded similar conclusions to the historical analysis. The forecasted average drought duration across Europe done in July 2003 clearly differs between the daily and monthly approaches, in particular the VTM and SSI-1 predict lower average duration for the upcoming 7 months. The seasonal forecasts issued each month in 2003 for the Rhine River supports the substantial differences in forecasted drought characteristics amongst methods using daily or monthly data and between variable and fixed threshold methods. The differences in the number of drought occurrences, average duration, timing, and deficit volumes between the variable threshold droughts (VTD, VTM, and SSI-1) and the fixed threshold droughts (FTD and FTM) highlight the importance of whether end-users of drought forecasts should take seasonality into account or not. Moreover, the temporal resolution of drought identification, that is, the use of daily or monthly data, is critical to consider. When the drought deficit volume is required, then the standardized approach (SSI) cannot be selected. The choice of the drought identification method when forecasting streamflow drought, in the end, lies in the end-users specific requirements and decisions, and there is no one drought identification approach that fits all needs. For this particular reason, the European DEWSs, such as the European Drought Observatory (EDO, Cammalleri et al., 2017, 2021) and the Anywhere DEWS (ADEWS, Sutanto et al., 2020a) forecast both standardized-based and threshold-based drought indices. EDO forecasts a combined drought indicator that consists of the SPI, a Soil Moisture Index (SMI, soil moisture anomaly), and the fAPAR (vegetation anomaly), as well as the low flow index (LFI) using the VTD approach. The ADEWS is more comprehensive than EDO and forecasts both standardized drought indices, such as SPI, SPEI, Standardized Runoff Index (SRI), Standardized Groundwater Index (SGI), and the VTD droughts in precipitation, soil moisture, river, and groundwater. The fixed threshold methods (FTD and FTM) are not used in none of these two DEWSs.

Our study, both the historical analysis and the forecasting, clearly shows that streamflow droughts obtained from different drought identification approaches (variable threshold (daily versus monthly), fixed threshold (daily versus monthly), and standardized index) differ in terms of their drought characteristics. Often scientists have analyzed and provided streamflow drought forecasts without clearly defining the identification method. This created misconceptions, miss-citations, and confusion among the academic community (authors, reviewers, editors), operational weather and water services, as well as end-users, which consider drought forecast products and associated terminology as interchangeably. Our study recommends scientists, developers of Drought Early Warning Systems, and end-users to clearly agree among themselves, preferable in a co-design phase, upon a sharp definition of which type of streamflow drought is required to be forecasted to mitigate the impacts of drought. Obviously, Drought Early Warning Systems also can include more than one drought identification method, as illustrated by Sutanto et al. (2020a). Then the end-user can decide in the end, which forecast product is most adequate based upon the provided description of the identification method and product.

*Data availability.* The streamflow EFAS data are accessible under a COPERNICUS open data license (https://doi.org/10.24381/cds.e3458969). In this study, we used EFAS system version 3. The SSI-1 analyzed using the SFO data and re-forecasts are available online in the 4TU Centre for Research Data with doi:10.4121/13056071.v1.

## Appendix A:  The 30DMA method

In this study, we applied the 30DMA method to the observed streamflow data as one of the pooling procedures to reduce the minor drought. This means that we averaged the first 30 days of the SFO data (from 1 to 30 January 1990) to calculate the streamflow on 16 January 1990. For 31 January 1990, we averaged the SFO data from 16 January 1990 to 14 February 1990 and so on until 15 December 2018. Missing 30DMA streamflow data from 1 to 14 January 1990 and from 16 to 31 December 2018 were not relevant since we have started drought analyses from the hydrologic year 1991 (from October 1990 to September 1991) to the hydrologic year 2018 (from October 2017 to September 2018). We applied the same hydrologic year for all European rivers. The reason for choosing the same hydrologic year (in our case: 28 years) is to ensure consistency in the analysis at the European level.

We also applied the centered 30DMA to the forecast data. To handle the forecast streamflow data at the start of the 215-day forecasts, we averaged 15 days of preceding observed data (SFO) with 15 days of the forecast to predict a possible drought event on the first day. For the second forecast day, we averaged 14 days of preceding observed with 16 days of forecast and so on. For example, the 30DMA forecasted streamflow on 1 August 2003 was obtained from moving averaging the SFO data from 17 July to 31 July 2003 with the forecasted streamflow from 1 August to 15 August 2003 (to predict a possible drought on 1 August 2003, lead time one day). Hence, the first 15 forecasted streamflow data from the 215-day time series included some observed flow that increases drought forecast skill for the first 15 days, which will affect possible forecasted drought events at the start of the forecast record using the VTD and FTD. The fusion method was applied to each of the 25 forecast ensemble members. The 30DMA method had not been applied to the monthly streamflow data for both historic period and forecasts. Thus, there is no influence of the SFO data on the monthly drought forecast analysis using the VTM and FTM.

## Appendix B:  Drought characteristics obtained from historical data

Appendix B includes drought characteristics obtained using observed (SFO) streamflow data from 1990 to 2018. In Figure B1, we present the number of minor drought occurrences that have duration less than 30 days. Streamflow drought was identified using the VTD approach. Figure B2 and B3 show the average duration of streamflow drought and average drought deficit volume in the European rivers identified with different drought identification approaches, namely the VTD, FTD, VTM, FTM, and SSI-1. Figure B4 illustrates the number of drought occurrences in European rivers from October 1990 to September 2018 (28 years) identified using the VTD approach without smoothing, i.e. applying the 30DMA method.

## Appendix C: Forecasting drought occurrence and deficit volume

Appendix C describes forecasted drought characteristics in major European rivers obtained from the forecast initiated in July 2003 for 7 months ahead (up to January 2004). Drought characteristics were derived using different drought identification approaches, namely the VTD, FTD, VTM, FTM, and SSI-1. Figure C1 and C2 show forecasted drought occurrences and forecasted average drought deficit volume, respectively.

*Author contributions.* S.J.S and H.A.J.V.L conceived and implemented the research. Data analyses, model output analyses, and all figures have been performed by S.J.S. S.J.S and H.A.J.V.L wrote the initial version of the paper and equally contributed to interpreting the results, discussion, and improving the paper.

*Competing interests.* The authors declare no competing financial and/or non-financial interests in relation to the work described.

*Acknowledgements.* The research is supported by the ANYWHERE project (Grant Agreement No.: 700099), which is funded within EU's Horizon 2020 research and innovation program www.anywhere-h2020.eu. The streamflow data came from the EFAS computational center, which is part of the Copernicus Emergency Management Service (EMS) and Early Warning Systems (EWS) funded by framework contract number 198702 of the European Commission. We thank Fredrik Wetterhall (ECMWF) for providing the EFAS data and two anonymous reviewers that helped to substantially improve the paper. This research is part of the Wageningen Institute for Environment and Climate Research (WIMEK-SENSE) and it supports the work of UNESCO EURO FRIEND-Water and the IAHS Panta Rhei program of Drought in the Anthropocene.

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

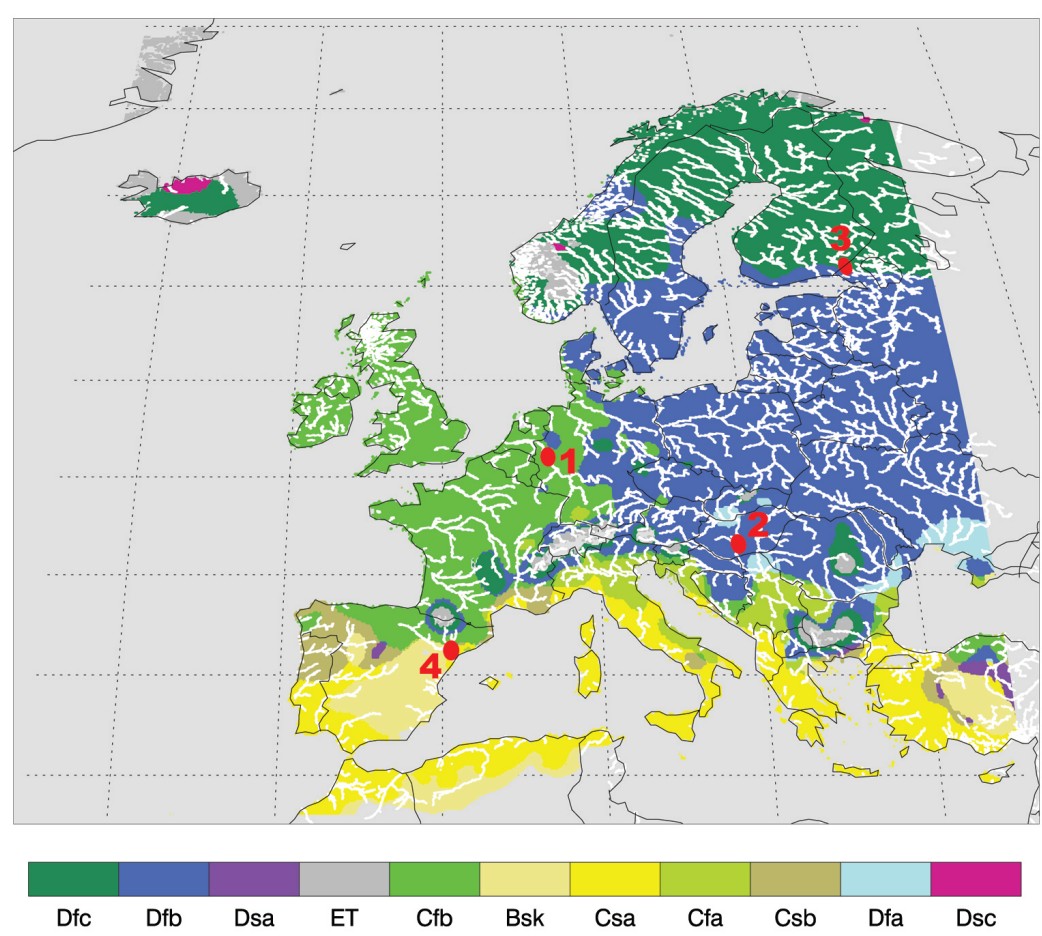

**Figure 1.** Köppen-Geiger map of Europe and locations of selected river basins for detailed hydrological drought analyses in different climate regimes, as shown by red dots. Location selected rivers: 1. Rhine, 2. Danube, 3. Vuoksi, and 4. Ebro. Readers are referred to Peel et al. (2007) for an explanation of Köppen-Geiger climate classification codes (e.g., Dfc, Dfb, Cfb, and so on).

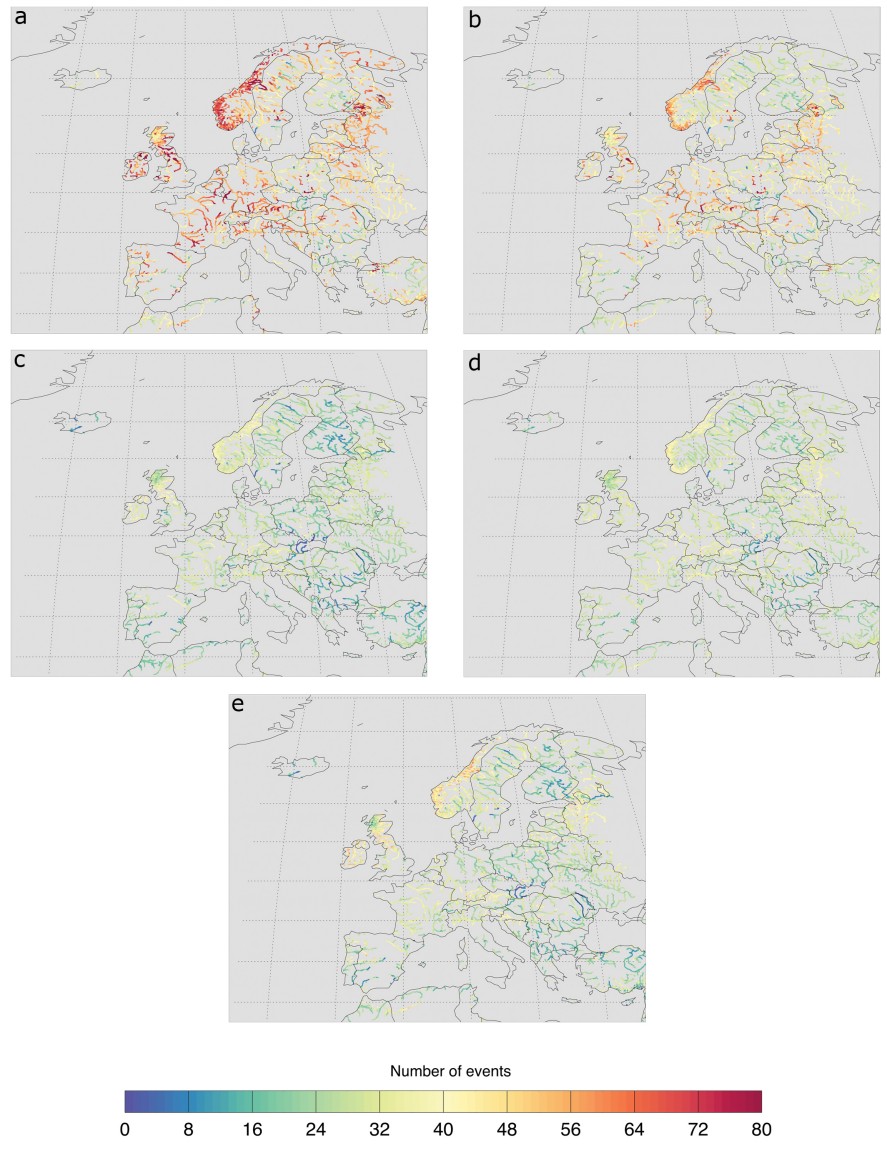

**Figure 2.** Number of drought occurrences in European rivers from October 1990 to September 2018 (28 years) identified using different drought identification methods: a) the variable threshold method with daily streamflow data (VTD drought), b) the fixed threshold method with daily streamflow data (FTD drought), c) the variable threshold method with monthly streamflow data (VTM drought), d) the fixed threshold method with monthly streamflow data (FTM drought), and e) the Standardized Streamflow Index with accumulation time 1 month (SSI-1 drought).

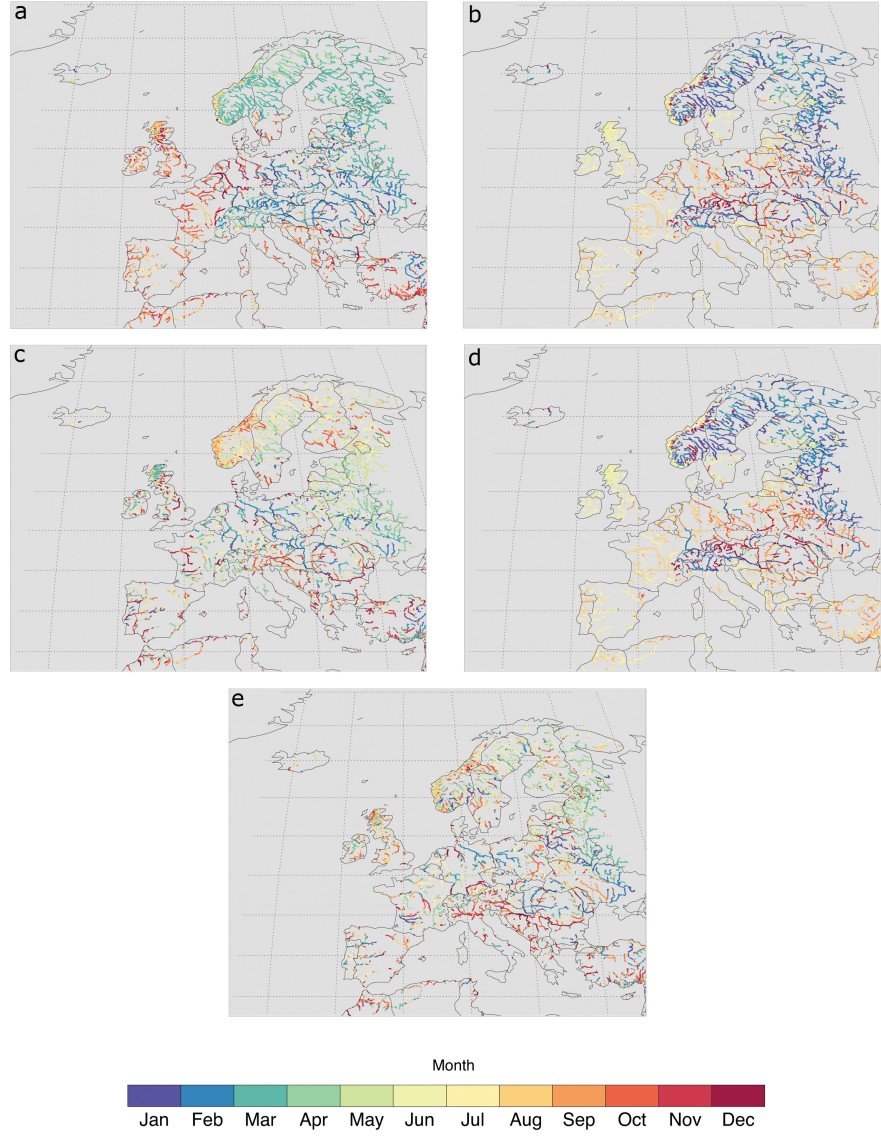

**Figure 3.** Drought timing (onset) in the European rivers from October 1990 to September 2018 identified using different drought identification methods: a) the VTD drought, b) the FTD drought, c) the VTM drought, d) the FTM drought, and e) the SSI-1 drought. For an explanation of the acronyms, see Fig. 2.

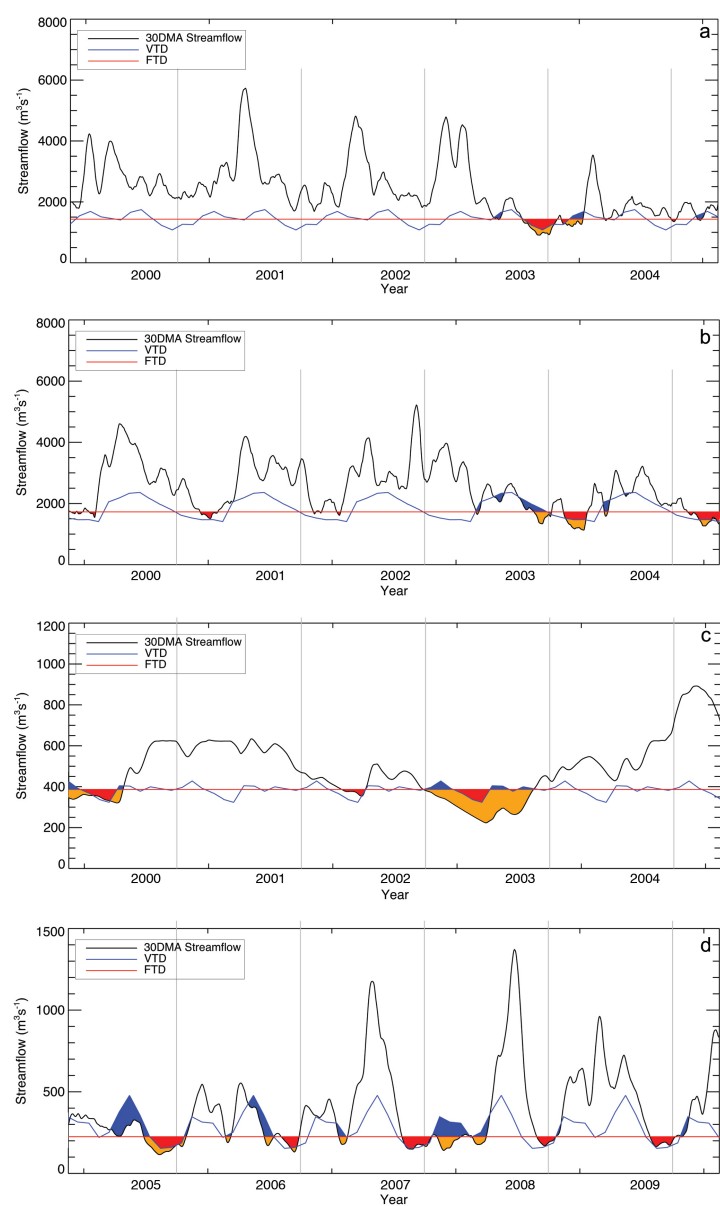

**Figure 4.** Streamflow droughts analyzed using daily streamflow data and different drought identification methods for the: a) Rhine River from 2000 to 2004 (location 1), b) Danube River from 2000 to 2004 (location 2), c) Vuoksi River from 2000 to 2004 (location 3), and d) Ebro River from 2005 to 2009 (location 4). Streamflow drought events are indicated as blue areas below the threshold for the VTD drought and red areas for the FTD drought. Orange areas indicate both VTD and FTD drought occurrences. Light grey lines show the start of hydrologic years (October). Locations are specified in Fig. 1.

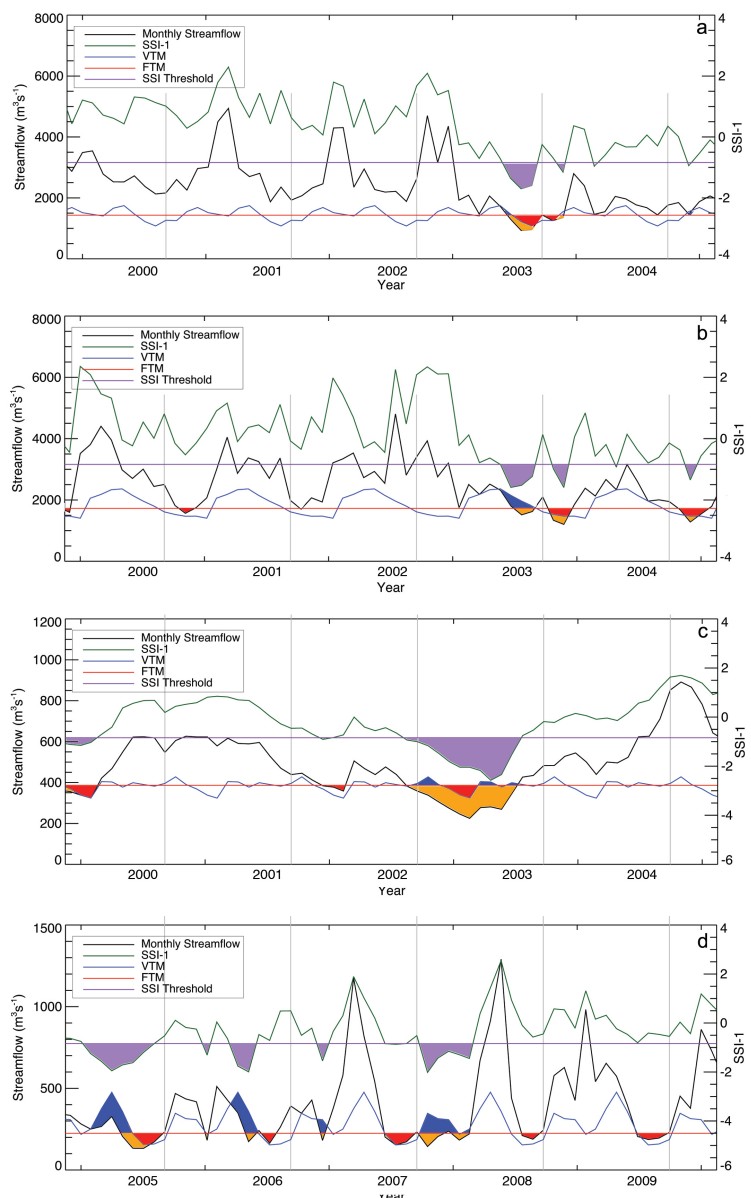

**Figure 5.** Streamflow droughts analyzed using monthly streamflow data derived from daily streamflow and different drought identification methods for the: a) Rhine River from 2000 to 2004 (location 1), b) Danube River from 2000 to 2004 (location 2), c) Vuoksi River from 2000 to 2004 (location 3), and d) Ebro River from 2005 to 2009 (location 4). Streamflow drought events are indicated as blue areas below the VTM drought, red areas for the FTM drought, and purple areas for the SSI-1 drought. Orange areas indicate both VTM and FTM drought occurrences. Light grey lines show the start of hydrologic years (October). Locations are specified in Fig. 1.

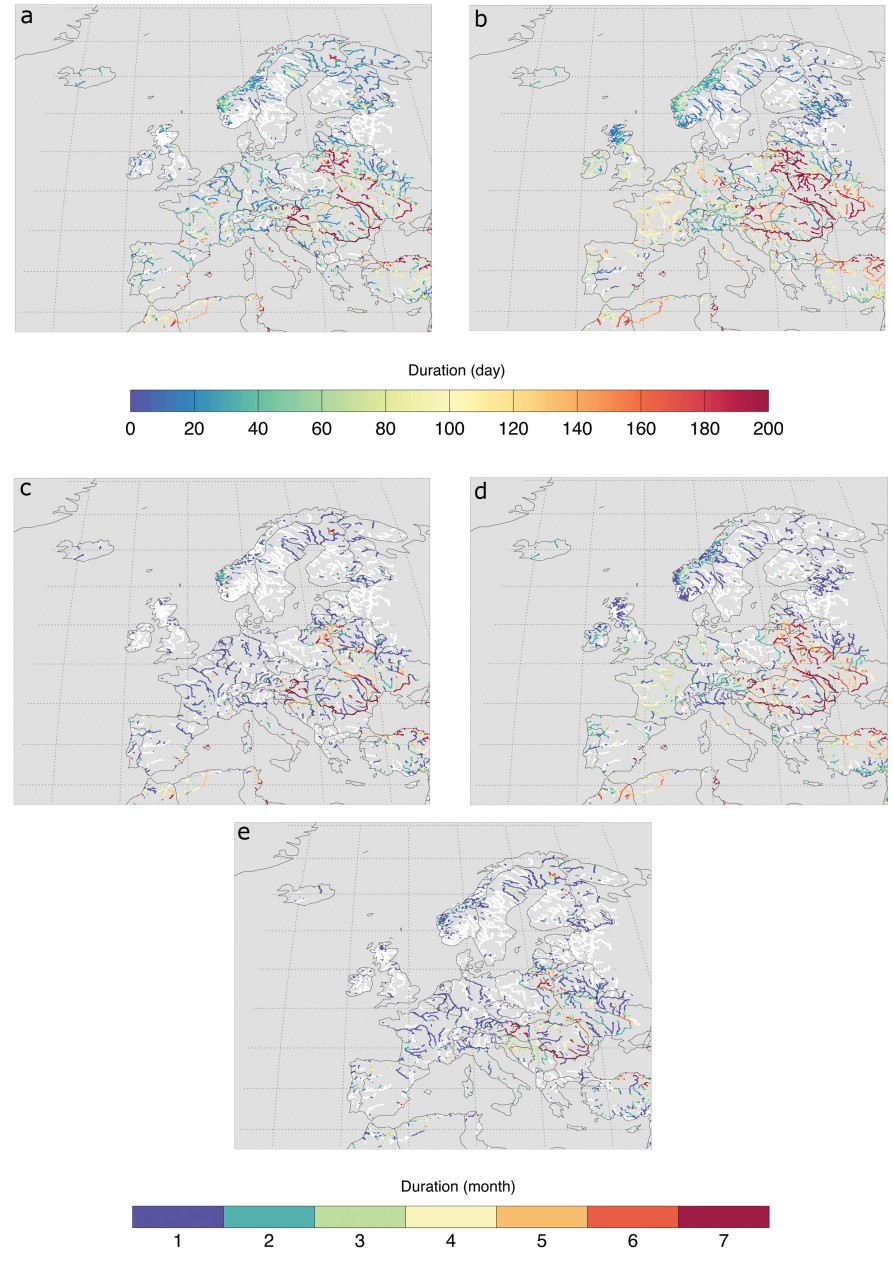

**Figure 6.** Forecasted average duration of drought events (median of 25 ensemble members) in the European rivers using different drought identification methods and the forecast initiated on 1$^{st}$ July 2003 with a lead time 7-month for: a) the VTD drought, b) the FTD drought, c) the VTM drought, d) the FTM drought, and e) the SSI-1 drought. White river color indicates that no drought was forecasted.

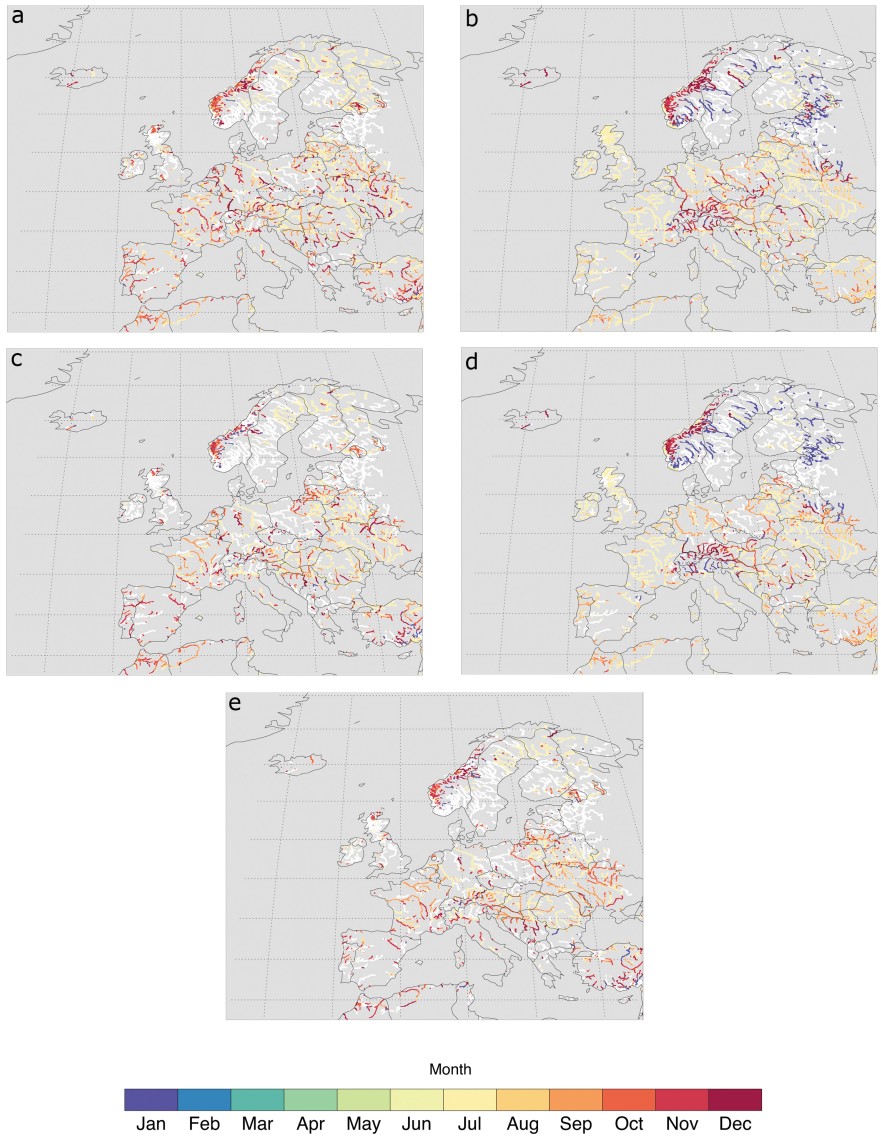

**Figure 7.** Forecasted drought timing (onset) in the European rivers using different drought identification methods and the forecast initiated on 1$^{st}$ July 2003 with a lead time 7-month for: a) the VTD drought, b) the FTD drought, c) the VTM drought, d) the FTM drought, and e) the SSI-1 drought. White river color indicates that no drought was forecasted. Please note that in this case drought cannot start in Feb-June.

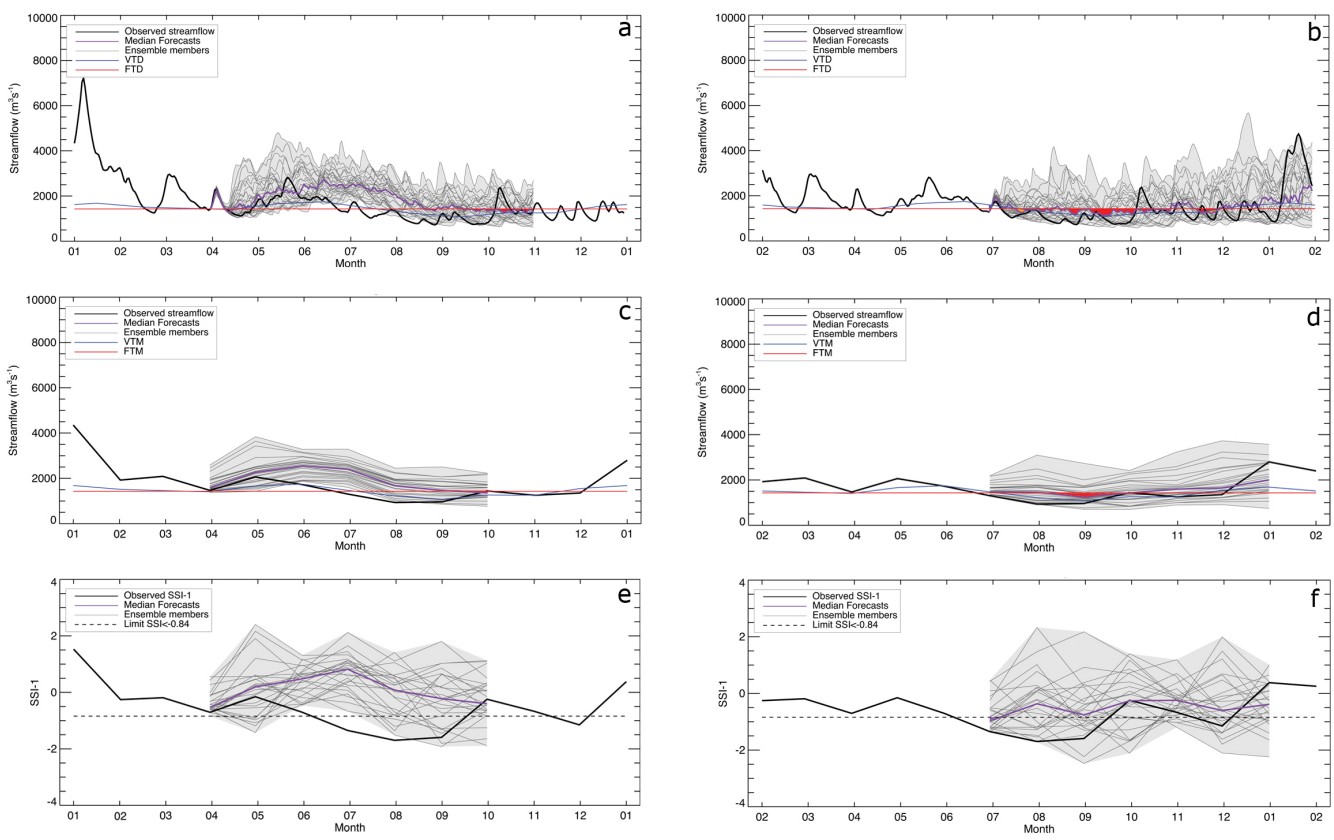

**Figure 8.** Observed and forecasted streamflow (median and 25 ensemble members) of the Rhine River: a) daily streamflow drought (VTD and FTD) initiated on 1[st] April 2003 for 7 months ahead, c) monthly streamflow drought (VTM and FTM) initiated on 1[st] April 2003 for 7 months ahead, and e) forecasted SSI-1 drought initiated on 1[st] April 2003 for 7 months ahead. b), d), and f) same as a, c, and e, but for forecasts initiated on 1[st] July 2003. Droughts are indicated by blue shaded area for VTD and VTM, red shaded area for FTD and FTM, and purple shaded area for SSI-1. Orange areas indicate both VTD and FTD drought.

**Table 1.** Streamflow drought characteristics derived from daily streamflow data using the VTD and the FTD methods obtained from the hydrologic years 1991 to 2018 for the five climate regions in Europe and the entire Europe. N stands for number of drought occurrences, T stands for timing (start month), D stands for duration (day), and DV stands for deficit volume in million m$^3$. D and DV are average drought characteristics and T is median drought timing for all river grid cells located in a climate region. Please note that Mediterranean region only consists of Bsk and Csa. Drought characteristics for Europe are obtained from the weighted average drought characteristics from each climate region considering the relative area in Europe (last column). We did not determine the drought timing for Europe because it does not make sense to calculate the average or median timing from climate regions.

| No | Region | Drought characteristics | | | | | | | | Area % of Europe |
| --- | --- | --- | --- | --- | --- | --- | --- | --- | --- | --- |
| | | VTD | | | | FTD | | | | |
| | | N | T | D | DV | N | T | D | DV | |
| 1 | ET | 55.4 | 4 | 44 | 49.3M | 39.7 | 3 | 55.7 | 40.5M | 2.6 |
| 2 | Dfb | 48.3 | 3 | 43.8 | 96.2M | 42.4 | 6 | 51.9 | 103.4M | 40.6 |
| 3 | Dfc | 49.2 | 3 | 46.7 | 71.1M | 34.5 | 2 | 63.7 | 59.2M | 24 |
| 4 | Cfb | 57.8 | 10 | 36.4 | 76.6M | 45.1 | 7 | 47.2 | 77.8M | 14.9 |
| 5 | Med | 41 | 10 | 56.3 | 39.3M | 31.6 | 8 | 68.4 | 33.6M | 17.8 |
| 6 | Europe | 49.6 | - | 44.6 | 79.4M | 39.6 | - | 56 | 78.8M | 91.3 |

**Table 2.** Streamflow drought characteristics derived from monthly streamflow data using the VTM, the FTM, and the SSI-1 drought identification method obtained from the hydrologic years 1991 to 2018 for the five climate regions in Europe and the entire Europe. See Table 1 for drought characteristic abbreviations. The unit for drought duration is month

| No | Region | Drought characteristics | | | | | | | | | | |
|----|--------|------|---|-----|--------|------|---|-----|--------|------|---|-----|
| | | VTM | | | | FTM | | | | SSI-1 | | |
| | | N | T | D | DV | N | T | D | DV | N | T | D |
| 1 | ET | 28.9 | 7 | 2.5 | 116.1M | 29.2 | 6 | 2.5 | 56.6M | 35.2 | 7 | 2.2 |
| 2 | Dfb | 26.5 | 5 | 2.5 | 149.2M | 29.1 | 7 | 2.5 | 139.9M | 29.5 | 5 | 2.2 |
| 3 | Dfc | 25.6 | 6 | 2.5 | 82.5M | 27.8 | 2 | 2.5 | 67.5M | 30 | 5 | 2.4 |
| 4 | Cfb | 30.7 | 5 | 1.9 | 129.2M | 30.4 | 7 | 2.2 | 115.8M | 34.8 | 7 | 1.9 |
| 5 | Med | 22.6 | 9 | 2.9 | 59.6M | 25.7 | 8 | 2.7 | 40.8M | 25.5 | 8 | 2.4 |
| 6 | Europe | 26.6 | - | 2.4 | 118.5M | 28.6 | - | 2.5 | 104.7M | 30.3 | - | 2.2 |

**Table 3.** Forecasted streamflow drought characteristics derived from daily streamflow data using the VTD and FTD approaches for the Rhine River initiated from 1st January 2003 to 1st December 2003 for 7 months ahead (215 days). Drought characteristics were derived using median of the ensemble. N stands for number of occurrence, Ne stands for maximum number of ensemble members falling below the drought threshold (%), T stands for timing (month), D stands for average duration (day), and DV stands for total deficit volume in million $m^3$

| Forecast initiation month | Drought characteristics | | | | | | | | | |
|---|---|---|---|---|---|---|---|---|---|---|
| | VTD | | | | | FTD | | | | |
| | N | Ne | T | D | DV | N | Ne | T | D | DV |
| 1 | 0 | 20 | 0 | 0 | 0 | 0 | 20 | 0 | 0 | 0 |
| 2 | 0 | 20 | 0 | 0 | 0 | 0 | 28 | 0 | 0 | 0 |
| 3 | 0 | 20 | 0 | 0 | 0 | 0 | 48 | 0 | 0 | 0 |
| 4 | 3 | 88 | 4 | 2.7 | 12.1M | 9 | 88 | 9 | 6.1 | 36.4M |
| 5 | 5 | 56 | 10 | 5 | 24.5M | 3 | 76 | 10 | 24 | 362.8M |
| 6 | 8 | 64 | 10 | 5.1 | 43.7M | 4 | 80 | 8 | 28.7 | 441.3M |
| 7 | 14 | 100 | 12 | 2.8 | 17.4M | 9 | 100 | 8 | 12.7 | 124.7M |
| 8 | 11 | 100 | 11 | 12.3 | 166.8M | 7 | 100 | 1 | 18 | 507.8M |
| 9 | 6 | 100 | 12 | 16.3 | 241.3M | 7 | 100 | 12 | 14.7 | 384.6M |
| 10 | 3 | 100 | 10 | 24 | 388.9M | 4 | 100 | 11 | 20.5 | 401.4M |
| 11 | 8 | 100 | 1 | 8.1 | 179.4M | 3 | 100 | 11 | 15.3 | 454.9M |
| 12 | 2 | 100 | 12 | 23 | 920.5M | 1 | 100 | 12 | 43 | 1,272M |

**Table 4.** Forecasted streamflow drought characteristics derived from monthly streamflow data using the VTM, the FTM, and the SSI-1 approaches for the Rhine River initiated from 1[st] January 2003 to 1[st] December 2003 for 7 months ahead (215 days). See Table 3 for drought characteristic abbreviations. The unit for average drought duration is month

| Forecast initiation month | Drought characteristics | | | | | | | | | | | | | |
| | VTM | | | | | FTM | | | | | SSI-1 | | | |
| | N | Ne | T | D | DV | N | Ne | T | D | DV | N | Ne | T | D |
| 1 | 0 | 8 | 0 | 0 | 0 | 0 | 8 | 0 | 0 | 0 | 0 | 8 | 0 | 0 |
| 2 | 0 | 0 | 0 | 0 | 0 | 0 | 0 | 0 | 0 | 0 | 0 | 0 | 0 | 0 |
| 3 | 0 | 8 | 0 | 0 | 0 | 0 | 28 | 0 | 0 | 0 | 0 | 16 | 0 | 0 |
| 4 | 0 | 40 | 0 | 0 | 0 | 1 | 52 | 10 | 1 | 195.3M | 0 | 24 | 0 | 0 |
| 5 | 1 | 56 | 10 | 1 | 124.9M | 1 | 60 | 9 | 2 | 872.9M | 0 | 44 | 0 | 0 |
| 6 | 1 | 60 | 10 | 1 | 146.3M | 1 | 72 | 9 | 3 | 1,072M | 0 | 48 | 0 | 0 |
| 7 | 1 | 52 | 7 | 1 | 21.9M | 1 | 68 | 9 | 2 | 482.4M | 1 | 56 | 7 | 1 |
| 8 | 2 | 100 | 8 | 2 | 628.6M | 1 | 100 | 8 | 4 | 2,841M | 2 | 100 | 8 | 1.5 |
| 9 | 1 | 72 | 9 | 2 | 882.1M | 1 | 100 | 9 | 2 | 2,245M | 1 | 92 | 9 | 2 |
| 10 | 2 | 72 | 12 | 1 | 325.9M | 1 | 92 | 10 | 2 | 1,132M | 1 | 72 | 10 | 1 |
| 11 | 1 | 84 | 11 | 2 | 505.4M | 1 | 96 | 11 | 1 | 880.7M | 1 | 80 | 11 | 1 |
| 12 | 1 | 84 | 12 | 1 | 1,243M | 1 | 84 | 12 | 1 | 927.2M | 1 | 64 | 12 1 | |

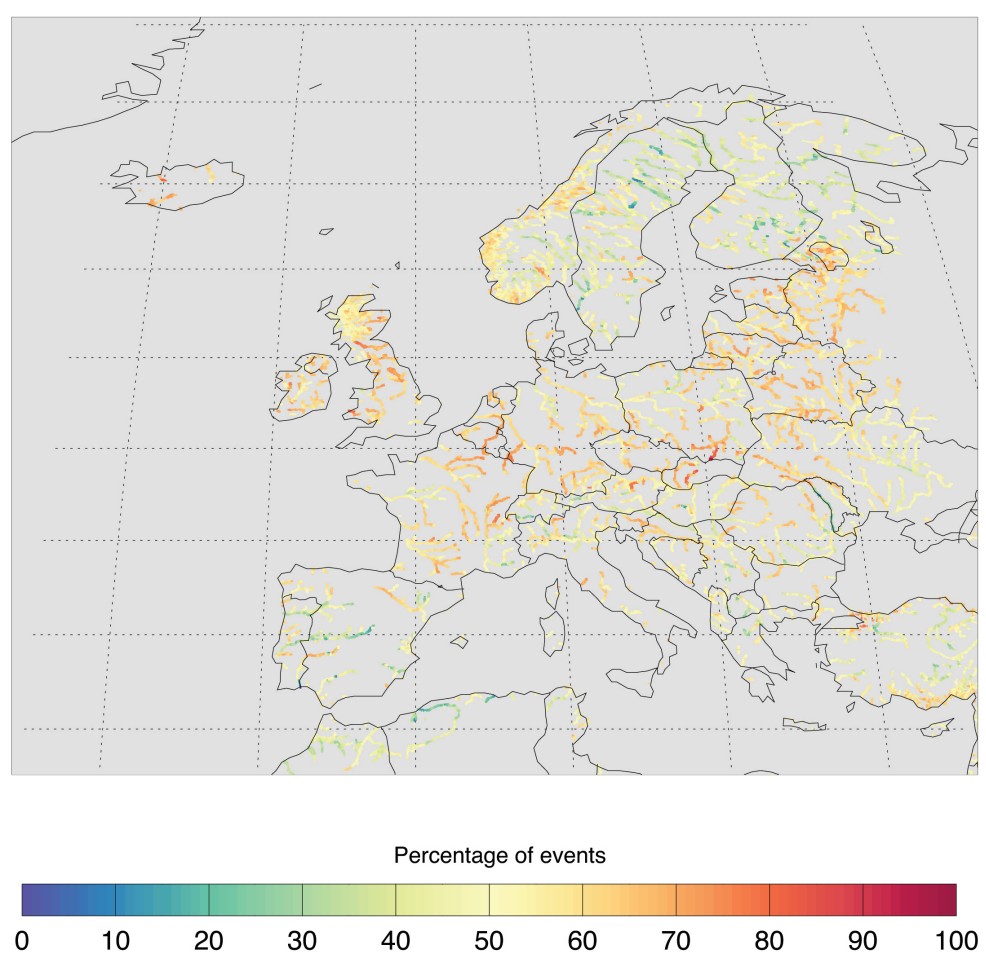

**Figure B1.** Minor drought occurrences (<30 days) in % of total number drought occurrences (Fig. 2) for the European rivers from October 1990 to September 2018 (28 years) identified using the variable threshold method with daily streamflow data (VTD drought).

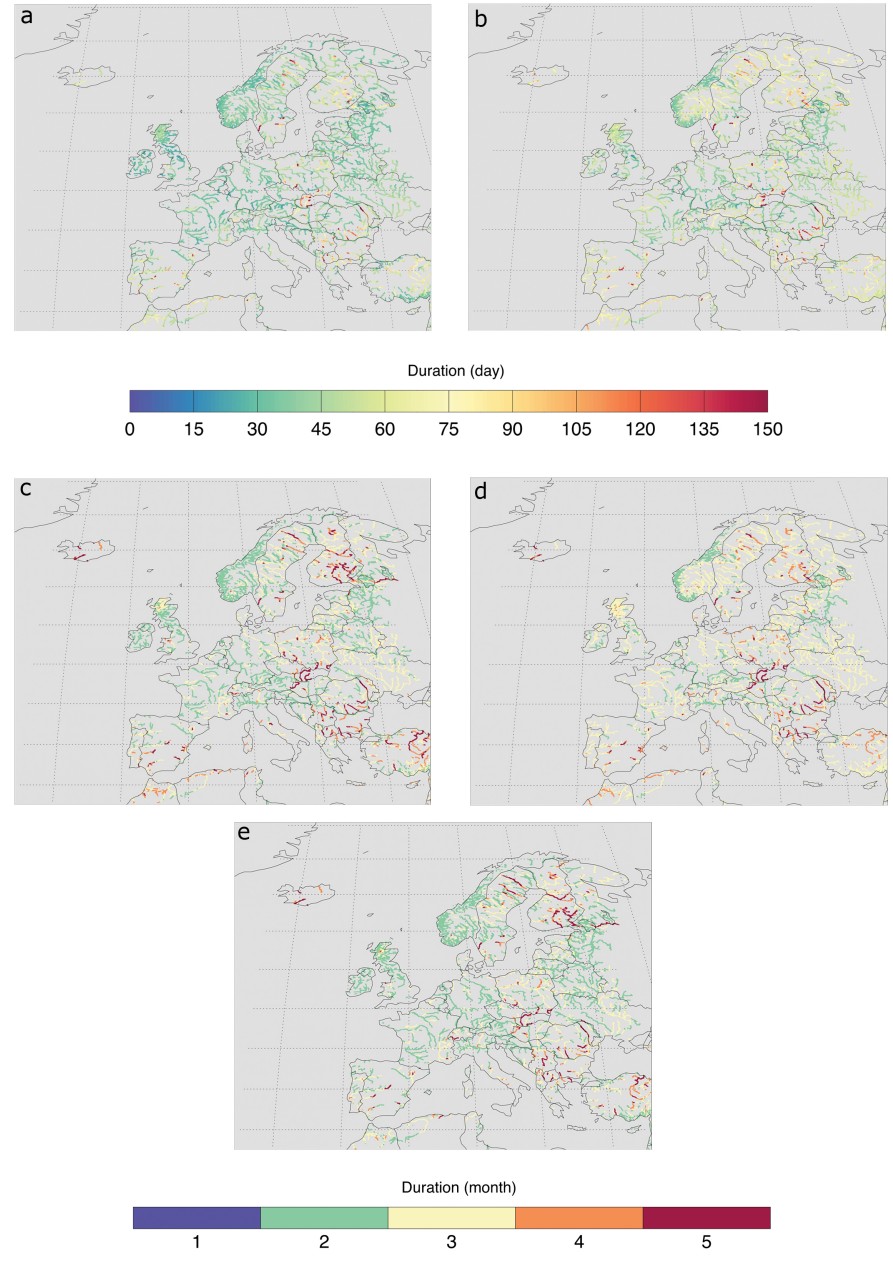

**Figure B2.** Average duration of drought events for the European rivers from October 1990 to September 2018 identified using different drought identification methods: a) the VTD, b) the FTD, c) the VTM, d) the FTM, and e) the SSI-1 approaches. For an explanation of the acronyms, see Fig. 2.

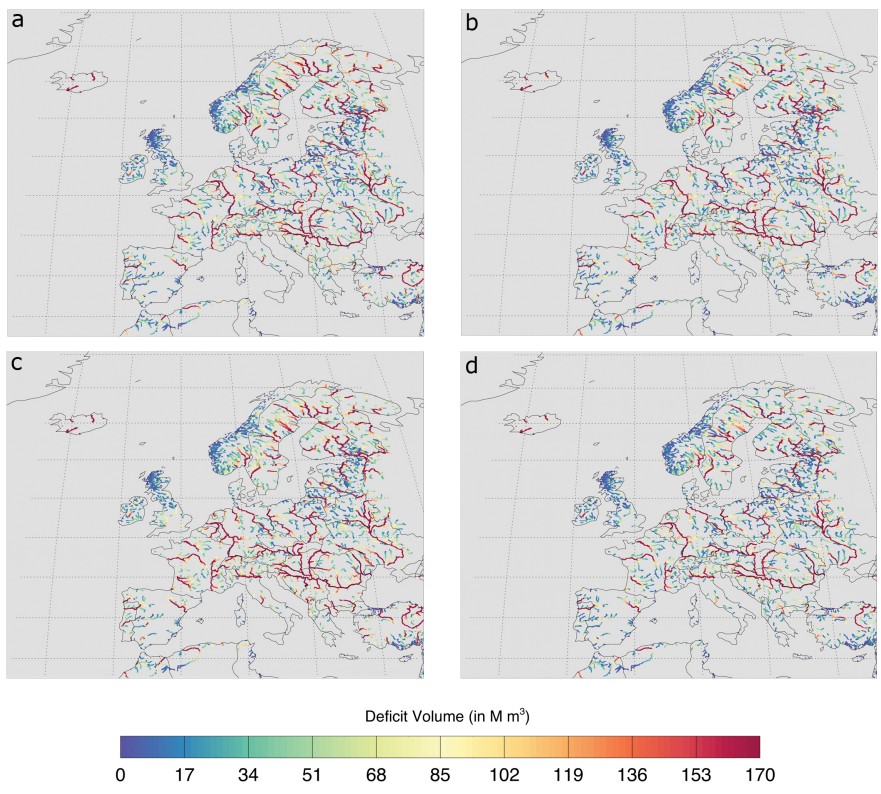

**Figure B3.** Average drought deficit volume for the European rivers from October 1990 to September 2018 identified using different drought identification methods: a) the VTD, b) the FTD, c) the VTM, and d) the FTM approaches. Note that deficit volume is not standardized.

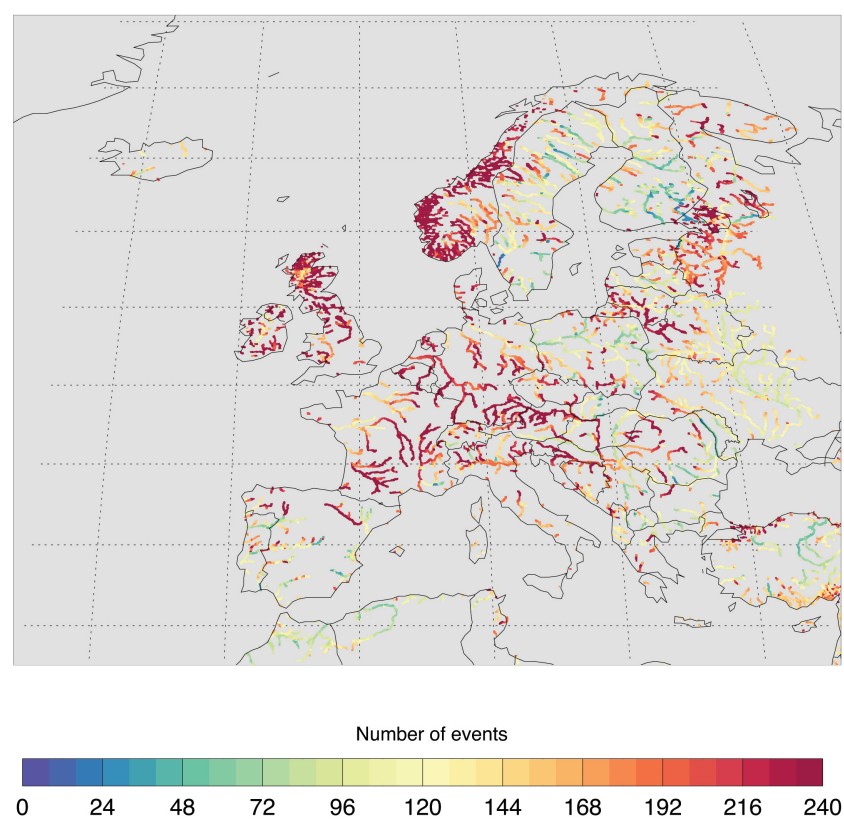

**Figure B4.** Drought occurrences in European rivers from October 1990 to September 2018 (28 years) identified using the VTD without the 30DMA application.

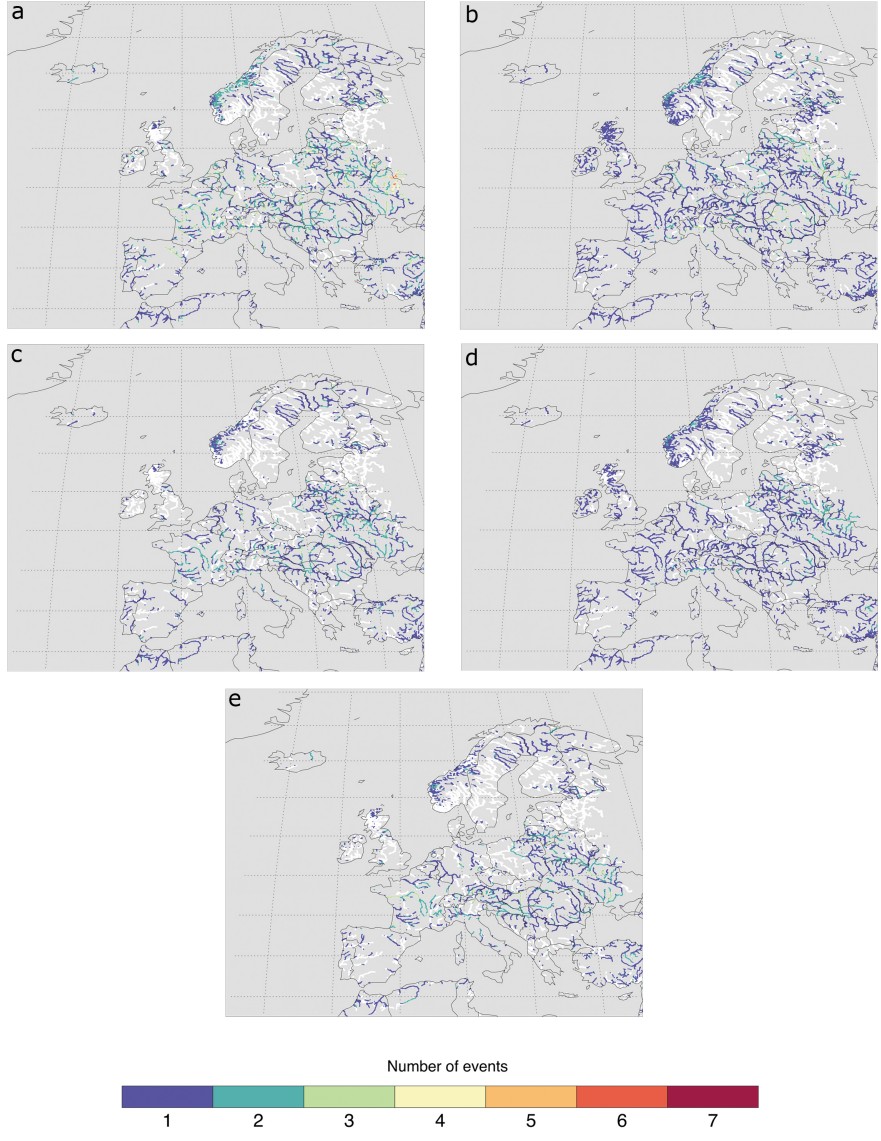

**Figure C1.** Forecasted drought occurrences (median of 25 ensemble members) in the European rivers using different drought identification methods and the forecast initiated on 1[th] July 2003 with a lead time 7-month for: a) the VTD, b) the FTD, c) the VTM, d) the FTM, and e) the SSI-1 approaches. White river color indicates that no drought was forecasted.

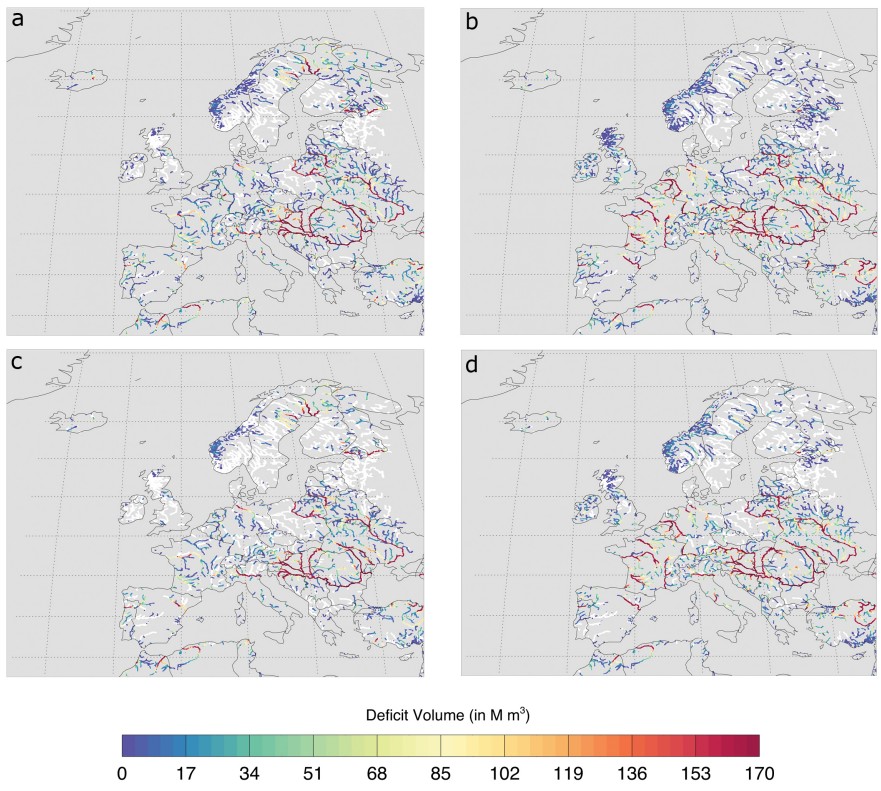

**Figure C2.** Forecasted average drought deficit volume (median of 25 ensemble members) in the European rivers using different drought identification methods and the forecast initiated on 1[th] July 2003 with a lead time 7-month for: a) the VTD, b) the FTD, c) the VTM, d) the FTM, and e) the SSI-1 approaches. White river color indicates that no drought was forecasted. Note that deficit volume is not standardized.