# Peer review of "Streamflow drought: implication of drought definitions and its application for drought forecasting"

_Hydrology and Earth System Sciences, 2020_

## Referee Comment (RC1) · Anonymous Referee #1 · 6 Nov 2020

**Review of "Streamflow drought: implication of drought definitions and its application for drought forecasting" by Sutanto and Van Lanen.**

The Study of Sutanto and Van Lanen compares different drought identification approaches: 1) the fixed threshold level method, 2) the variable threshold level method and 3) the threshold level method applied on SSI time series, for simulated river flow at the pan-European scale. They show that (average) drought event characteristics differ based on the used drought identification method. Consequently, they show that drought event forecasts differ, depending again on the used drought identification method. Overall, the main recommendation of the paper is strong and relevant, i.e., droughts differ depending on the used method and streamflow drought forecasters and stakeholders should agree which type of drought should be forecasted. In addition, I believe that Figure 6 provides an informative message for the users and developers of hydrological drought forecasting systems.

However, given that this paper focusses on the definitions of drought and methodology of drought identification, it sets an example which types of drought identification approaches can be used for drought forecasting applications (and how). Therefore, it should be extra "sharp" in its drought definition and identification approaches as well. At this stage, this is not the case and there are several methodological concerns that should be addresses carefully. In addition, the comparison of the results is far from straight forward. The used drought identification approaches do not only vary in overall method, but also in: 1) threshold (<10 percentile for the fixed and variable threshold approaches and around <50$^{th}$ percentile threshold for the SSI), 2) data accumulation period (1 month for the fixed and variable threshold based approaches vs. 6 months for the SSI), and 3) temporal resolution (daily vs. monthly). Finaly, the most novel part of this paper, which deals with the implications for drought forecasting, is rather limited and deserves more attention in my opinion.

**Major comments:**

**Methodology**

**SSI computation:**

Why SSI-6? For me, it makes sense to aggregate meteorological drought indices (SPI, SPEI) to differentiate between slow and fast responding (hydrological systems), e.g., catchment with small and large storage components. However, riverflow already encompasses the accumulation and delay of the meteorological signal caused by e.g. delayed groundwater flow. From a riverflow drought perspective, it is often important to know what is currently happening in the river (SSI-1) and not what happened in the past 6 months (SSI-6). Also, the SSI-6 is not at all comparable to the 30-Day moving window used for the FT and VT approaches. This makes the interpretation of the comparison between both approaches less straight forward. Finaly, the reasoning to choose the SSI-6 over the SSI-1 because the SSI-1 results in many minor drought events does not compensate for the advantages of the SSI-1.

Why an SSI threshold of zero to identify drought? I would not term something that happens 50% of time drought. Please note that the original SPI paper of Mckee (1993) uses a similar threshold, but has the additional requirement that the SPI should at least reach a value of -1 over the course of the drought event. In addition, an SSI threshold of zero is far from comparable to an FT or VT of Q90 used for the threshold level approaches.

Why the gamma distribution to derive the SSI? I agree that is hard to find a suitable distribution to fit to riverflow time series (line 150-151). However, that is not a good argument to simply use the Gamma distribution. There are likely to be better alternatives for your pan-European dataset (See e.g. Svensson et al., (2016), Tijdeman et al. 2020).

Why no goodness of fit testing? The studies above concludes on different suitable candidate distributions for the SSI (other than the gamma distribution) that might be applicable for the current study. However, that does not mean that they can be applied on your dataset of simulated streamflow series by default, as your dataset might exhibit different properties as compared to the observed riverflow timeseries. Careful evaluation which distribution is most suitable for your set of rivers is required.

Which distribution fitting method was use?

For the forecasted SSI: Did you use the parameters of the population distribution derived from historical monthly flow values to derive the SSI for forecasted values? Or did you replace the historical values with forecasted values and than recalculated the population distribution to derive the SSI? And why, e.g., what should a forecaster do?

**Threshold approach:**

Line 123-143: Many different smoothing procedures have been applied in combination with the threshold level method. This has been done for good reason, however, sometimes resulting in an (unwanted) increase/decrease in drought occurrence, especially for the VT method. For me, a 10th percentile implies that 10% of the time series is in drought and that drought occurrence is equally distributed over the year in case of the VT method. However, by first deriving the the threshold from daily streamflow data, and then smoothing both the threshold and riverflow timeseries seperately, this is not necessarily the case anymore. This might be solved relatively easily, i.e., first apply the moving average and then derive the threshold. Or you could use monthly data.

Line 366-367: You encourage using monthly streamflow data for drought forecasts but use daily streamflow in your own analyses. I would have find it logical to do this as well in this study, e.g., instead of the FT and VT approaches applied on daily data, it could be applied monthly averaged data. This also increases the comparability with the SSI. Further, is there really merit in forecasting streamflow drought duration and deficit at a daily resolution, especially for the longer lead-times? Is this being done somewhere? Can this be done with any skill? If not, wouldn't it be better to just stick to monthly data for which at least some skill might be achieved?

**Results and discussion:**

Section 3.2. The forecasting section, which is the most the novel part of this paper, would benefit from some more attention. Figure 6 provides a nice illustration, even though it might be a little obvious at this point in the papers that drought characteristics derived with different methods will vary, given that you apply a different threshold on the same forecast data. However:

- I disagree that the drought of 2003 in the river Rhine started in August 2003. According to the SSI-1, river levels dropped to below normal anomalies much earlier. I suggest to start earlier in the year.
- Why not add the observed hydrograph to the plot?
- Isn't the fact that the VT method does not forecast a drought a good thing? According to this method, there was also no drought in the observed hydrograph (Fig. 4a) – how could this method have "performed better" (line 340).
- Why not show the SSI-1 here?

Given the focus of the paper on river flow forecasts, I would expect more focus on the latter, and not only an examplary timeseries river flow forecasts for one river / event. It would be interesting to include.

- At least, an evaluation and discussion of the spread in streamflow forecast and especially in the spread in streamflow drought forecast, and (i.e., not only the evaluation of the median forecast). What are the ranges in drought characteristics derived from the forecast ensemble?
- Consequently an evaluation or discussion of the streamflow (drought) forecasts skill, i.e., can certain "types of droughts", e.g., FT vs. VT vs. SSI, be forecasted better?

The above evaluation would benefit the consideration of multiple rivers, drought events, or start months.

Again, I would avoid the SSI-6 here, due to the strong autocorrelation of this index, which makes it relatively easy to forecast on short lead times. For example, for a forecast with a lead-time of 1 month, 5 out of 6 months are already known. Rather, I would look at the SSI 1.

Finaly, some (non-committal) suggestions for Section 3.1 that could further improve the manuscript:

Section3.1.1 Next to showing the amount of streamflow droughts, you could consider showing other characteristics such as the average duration, deficit volume, or the number of minor drought events. This provides valuable insights in differences between methods, and further makes the notions in 3.3.1 about regions with more minor drought quantitative. In addition, you can derive a proxy for deficit volume from standardized time series. The units are meaningless and not comparable with the deficit volumes derived with FT and VT method. However, the relative difference over Europe should pop-up.

Section 3.1.2 In addition to discussing when most drought starts, it might be interesting to see when most drought occur in difference climates. This can be presented as a series of histograms for each climate, with the month on the x-axis and the fraction of drought months that occurred in that month on the y-axis.

**Minor comments:**

Line 2: "… the term streamflow drought forecasting, rather than streamflow forecasting …" You could briefly explain difference between the two here.

Line 5: "within" Correct?

Line 6: Be careful with terming these extreme events.  They are anomalies, but something that happens on average at least once every year, as is the case in your study, is not an extreme event.

Line 7, 8: "observed" might be "observations"

Line 7: "a LISFLOOD model"… are there more?

Line 10: add method to VT and FT, e.g. variable threshold level method.

Line 10: You also apply a threshold based approach on SSI time series. Mention this here.

Line 16: "Eliminate". Not true. You can still have 1-day droughts with these TL approaches.

Line 24: "IPCC" should be "The IPCC".

Line 34: This sentence slightly contradicts with Line 1, where you state that drought forecasting is a key element of DEWS. I would expect there to be some examples. Which contemporary "DEWS" include streamflow drought forecasting, using the approaches as described in the paper (FT, VT and SSI), not just streamflow forecasting)?

Line 41: "evaporation" should be potential evapotranspiration

Line 47: "used" should be "be used"

Line 49: Mention that you specifically focus on simulated streamflow drought.

Line 75: "There" should be "There is"

Line 85: "Proxy" should be "Proxies"

Line 89: "proxy observed streamflow" could just be "simulated streamflow"

Line 112: "reforcasted data 2003" should be "re-forecasted data of 2003"

Line 119: "in" should be "for"

Line 128: "were moving averaged" rephrase

Line 134: "For the threshold" …this refers to variable threshold approach I guess? In this section, make the clear distinction between FT and VT and seperately explain how both are derived.

Line 138-140: add here that MA introduces a significant amount of auto-correlation, which affects the skill of the river flow forecast for the first 30 days significantly.

Line 147: "median" should be "expected median".

Line 155-160: Add here that it is quite easy to forecast the SSI-6 for short lead times, given the strong autocorrelation of the timeseries. E.g., for 1-month lead-times, you already know five months and only have to forecast one.

Line 162-164: Please explain how you classify an event with varying SSI values into one category.

Line 162-177: Did you derive the climate classification yourself using the approach described in Peel et al (2007)? Or did you use their dataset?

Line 179: "definitions" … "drought identification approaches" might be better.

Line 188: "Lower than median streamflow" … Not nescecairely true. Technically, above median streamflow can still be a negative SSI and vice versa. Depends on the sample and (goodness of fit) population distribution to derive the SSI.

Line 189: Figure 3 does not show that streamflow droughts occur every year.

Line 200: This is comparing apples and pears, as the thresholds are completely different.

Line 203-206: Could this not be compensated by a higher number of drought in winter for the VT?

Line 221: "drought that has" should be "droughts that have"

Line 228. "(Coincides with hydrologic years in most of Europe)" remove: unneeded repetition

Line 264-266. Is the last part, i.e., about the lowest and n-day minimum flow, needed? Interrupts flow.

Line 266-267. Looking at Fig. 5a, I find the SSI-1 timeseries much more informative about drought in the river Rhine. Rhine drought reaches is maximum in summer 2003, and recovers in winter 2004. For me, this make much more sense than the SSI-6 timeseries. Was the drought in the river Rhine really a multiyear event? Were there impact directly related to Rhine river flows over the course of 2004?

Line 270. For me, this description of drought in the river Rhine makes much more sense. It would make even more sense if you would use a more appropriate drought threshold (maybe SSI-1 < -0.84, corresponding to the 20th percentile). I don't see the problem of having 2003 split up in different events and question why it is better to use an SSI-6 and thereby inflate the event to a multiyear drought.

Line 285: "C" should be century.

Line 295-302. Why limit yourself here to the four case study Rivers and the limited time window? You could directly compare the number of drought events & their deficit volumes over a longer time period and for all the catchments (starting by deriving the difference between Fig. 2a and b).

Line 312-329. According the definition of drought according to VT and the SSI, droughts are expected to occur for an equal amount of time over the year. Please provide an explanation for the distinct temporal differences in drought occurrences. Or is this still referring to the start month of the drought?

Line 309: "(except for the Rhine River)" this contradicts with the discussion in the paragraph above.

Line 337: Not only meteorological drought, also streamflow drought according to the SSI-1 (Fig. 5a).

Line 354. Which is good, because there was no drought according to the VT, or?

Line 382: "eliminate" … not correct as minor droughts can still occur.

Line 361: "rare extreme drought events" … extreme events are by definition rare. Rephrase.

Line 368: "the FT method produces higher drought deficit volumes and duration than VT" not shown for the pan-European dataset.

Line 375: "occurred" should be "started".

Line 377: "what being identified by" rephrase

Figure 1: Nice. What is the difference between light and dark grey in e.g., the Alps?

Figure 2: You could add the upper boundary, e.g. 30-xx instead of >30.

Figure 3: "The timing for drought was determined based on the first month of each drought event." This is the same as what is said in the beginning of the caption.

Figure 4: Some droughts are hardly visible (e.g. in Figure 4a). It might work to use a log-scale

Figure 4: Axis lables: $m^3 \, sec^{-1}$ or $m^3$ / sec instead of m3/sec

Figure 4: Are the grey vertical lines the hydrological years?

Figure 4. You might consider using a different color when VT and FT overlap.

Figure 5. Add grey vertical lines here as well.

Figure 6. Same comments as for Figure 4 and 5.

Table 1. Would be interesting to also compare average deficit volume and timing.

**References:**

Mckee, T. B., Doesken, N. J., and Kleist, J. (1993): "The Relationship of Drought Frequency and Duration to Time Scales." *AMS 8th Conference on Applied Climatology*

Svensson, C., Hannaford, J., and Prosdocimi, I. (2016): "Statistical Distributions for Monthly Aggregations of Precipitation and Streamflow in Drought Indicator Applications." *Water Resources Research*, https://doi.org/10.1002/2016WR019276

Tijdeman, E., Stahl, K., and Tallaksen, L. M. (2020) "Drought Characteristics Derived Based on the Standardized Streamflow Index – a Large Sample Comparison for Parametric and Nonparametric Methods." *Water Resources Research*, https://doi.org/10.1029/2019WR026315

---

## Referee Comment (RC2) · Anonymous Referee #2 · 9 Nov 2020

General Comments

The authors performed an intercomparison of three different streamflow drought indicators, with the goal to highlight the differences in the drought characteristics associated to each index and to detail the implication on drought forecast. I found the overall goal of the study meaningful, given the confusion that still arise among scientists and operational users on the topic, but I also found the paper and its structure generally out of focus. The key message of the paper "...developers of DEWS and end-users should clearly agree among themselves upon a sharp definition on which type of streamflow drought is required to be forecasted for a specific application." is in my opinion, even if

relevant, better suited for a short communication or letter paper rather then a research paper.

The research results that should support this conclusion as reported in this paper are somewhat lacking in both clarity and rigorousness.

The main drawback of the analysis is the fact that the authors uses three drought indicators that rely on quite different input data and basis hypotheses to conclude that they provide a different picture of drought. This result is quite obvious after an attentive read, given the background premises: - daily data for threshold methods vs. monthly data for SSI. - 90th percentile for threshold methods vs. median for SSI (SSI=0). - Event-based approach for threshold vs. single monthly value for SSI All these discrepancies in the drought definition make the intercomparison a mere exercise, and its outcomes are hard to translate into actual general considerations.

An additional drawback is the general lack of details on the implementation of the three approaches, which severely limits the possibility for the readers to extrapolate meaningful information from the research outputs.

Finally, the analysis on the implications on drought forecast, which should be the main focus of the paper according to the title, is very limited in scope, and it needs to be significantly expanded in order to keep it as the focus of the paper.

Specific Comments

Introduction

The authors should better highlight how different definitions of streamflow drought in DEWS exists also for two reasons: 1) different users have different needs that can be accommodate by different indicators (e.g. river navigation may be affected more by FT droughts that VT droughts), 2) different available input data lead to different definitions (e.g. threshold methods may not be suitable for monthly data, and daily data may not be available in near-real time).

Data and Methods

The description of the different drought indices need to be more explicit. How the drought events are defined for each index? How is the onset computed? Severity? Duration? Any event definition in the SSI? Etc... Also, more consistency on the adopted thresholds need to be enforced (why SSI=0 is used as threshold when 90th percentile is used for VT and FT?). It is also worth to mention that a VT method based on the same LISFLOOD data is currently operationally implemented as part of EDO (https://edo.jrc.ec.europa.eu/).

Results and discussion

There is a clear unbalance between the historical analysis and the forecast. Give the title of the paper, I would aspect much more emphasis on the latter.

---

## Author Comment (AC1) · 30 Nov 2020

**Reply to reviewer 1**

We would like to thank the reviewer for valuable suggestions and comments. In this document, **P** refers to the page number and **L** refers to the line number in the recent paper. For example, **P3L65-70**, refers to page 3, lines 65-70.

| No | Comment | Reply |
|----|---------|-------|
| **Reviewer 1** | | |
| 1 | The Study of Sutanto and Van Lanen compares different drought identification approaches: 1) the fixed threshold level method, 2) the variable threshold level method and 3) the threshold level method applied on SSI time series, for simulated river flow at the pan-European scale. They show that (average) drought event characteristics differ based on the used drought identification method. Consequently, they show that drought event forecasts differ, depending again on the used drought identification method. Overall, the main recommendation of the paper is strong and relevant, i.e., droughts differ depending on the used method and streamflow drought forecasters and stakeholders should agree which type of drought should be forecasted. In addition, I believe that Figure 6 provides an informative message for the users and developers of hydrological drought forecasting systems. | We would like to thank the reviewer for the comments, valuable suggestions, and acknowledgement of the message in our paper that drought forecasters and stakeholders should agree at front which type of hydrological drought should be forecasted. |
| 2a | However, given that this paper focusses on the definitions of drought and methodology of drought identification, it sets an example which types of drought identification approaches can be used for drought forecasting applications (and how). Therefore, it should be extra "sharp" in its drought definition and identification approaches as well. At this stage, this is not the case and there are several methodological concerns that should be addressed carefully. In addition, the comparison of the results is far from straight forward. The used drought identification approaches do not only vary in overall method, but also in: 1) threshold (<10 percentile for the fixed and variable threshold approaches and around <50th percentile threshold for the SSI), 2) data accumulation period (1 month for the fixed and variable threshold based approaches vs. 6 months for the SSI), and 3) temporal resolution (daily vs. monthly). | The referee is concerned about the methodology used in our paper, i.e. in three aspects: 1) the thresholds to identify drought, 2) the data accumulation period, and 3) the temporal resolution. Our answer to these three questions is as follows:
 i) Our paper uses the drought threshold based on common practice in the drought community. Using a threshold method either a Fixed Threshold (FT) or Variable Threshold (VT), drought is identified if the streamflow falls below the threshold, which is commonly in the range of 10-30th percentile of the flow duration curve (P70-90) (Hisdal et al. 2004; Van Loon, 2015). On the other hand, the standardized indices, e.g., the Standardized Streamflow Index (SSI) identifies drought if the SSI value falls below 0, which is 50th percentile (Vicente-Serrano et al., 2012). Our reason to use different thresholds (50th percentile for SSI and 10th percentile for the FT and VT) is that we would like to follow common practice for the different approaches. However, the reviewer has a point that the comparison between threshold methods (VT, FT) and SSI is not |

| | | |
|---|---|---|
| | | equal regarding to the use of different percentiles. Thus in the revised manuscript, we will change the thresholds from P90 into P80 for VT and FT, and SSI≤-0.84 (~P80) to have a fair comparison between different drought indices (Tijdeman et al., 2020). |
| | | ii) Our study also provides results obtained from SSI-1 (Fig. A1 and A2). The main reason we used the SSI-6 for comparison with the threshold method is that SSI-6 produces a similar number of drought events than the threshold method VT and FT (Figure 2 and Table 1). SSI-1 on the other hand produces many minor drought events (Fig. A1). This is due to the selected drought threshold (P50) we used, as mentioned in point 1 above. We realize that streamflow, as included SSI-1, comprises some catchment memory aspects (delayed flow from groundwater). Hence, in the revised manuscript, we will replace SSI-6 with SSI-1 in the main text. However, we need to realize that anomalies in the accumulated flow over a longer period (e.g. SSI-6) have relevance for some purposes, such as the management of surface water reservoirs. |
| | | iii) We do agree with the reviewer that our study used different temporal resolution to analyze drought, which are daily for threshold methods and monthly for SSI. Again, we followed common practice (see item i, above) to identify drought using these methods. Many studies used daily streamflow data to analyze drought using the threshold method and monthly streamflow data to analyze drought using the standardized indices. To the author's knowledge, only Tallaksen et al., 2009 used the monthly data to derive drought using the threshold method and only for a scientific purpose. In the revised manuscript, however, we will add to the common practice approach (daily resolution), an analysis of drought characteristics using monthly streamflow data in both FT and VT drought approaches. This allows an analysis of the VT and FT threshold approach and the SSI-1 using the same temporal resolution, i.e. monthly time scale. This implies that we will have two VT and FT threshold applications: daily resolution, as frequently used, and monthly resolution to allow comparison with SSI-1. |
| 2b | Finally, the most novel part of this paper, which deals with the implications for | We will extend the novel part of paper to illustrate that the outcome of the forecast |

| | | |
|---|---|---|
| | drought forecasting, is rather limited and deserves more attention in my opinion. | depends on the drought identification method. We will do this by describing: (i) pan-European maps showing forecasted drought timing and duration using different drought identification methods (FT and VT with daily and monthly resolution, and SSI-1) (number of drought occurrence/frequency and drought deficit volume will be provided in the Supplementary Material), and (ii) summary of forecasted drought characteristics identified using different approaches in the Rhine River using forecasts initiated from 1st January 2003 to 1st December 2003 with a lead time of 7-month. In addition we will also provide information on the percentage of ensemble members showing drought for each identification method. |
| 3a | **SSI computation:**
Why SSI-6? For me, it makes sense to aggregate meteorological drought indices (SPI, SPEI) to differentiate between slow and fast responding (hydrological systems), e.g., catchment with small and large storage components. However, riverflow already encompasses the accumulation and delay of the meteorological signal caused by e.g. delayed groundwater flow. From a riverflow drought perspective, it is often important to know what is currently happening in the river (SSI-1) and not what happened in the past 6 months (SSI-6). Also, the SSI-6 is not at all comparable to the 30-Day moving window used for the FT and VT approaches. This makes the interpretation of the comparison between both approaches less straight forward. Finaly, the reasoning to choose the SSI-6 over the SSI-1 because the SSI-1 results in many minor drought events does not compensate for the advantages of the SSI-1. | We do agree with the reviewer, and thus we will switch the SSI-6 results with SSI-1 (see our reply 2a, ii). |
| 3b | Why an SSI threshold of zero to identify drought? I would not term something that happens 50% of time drought. Please note that the original SPI paper of Mckee (1993) uses a similar threshold, but has the additional requirement that the SPI should at least reach a value of -1 over the course of the drought event. In addition, an SSI threshold of zero is far from comparable to an FT or VT of Q90 used for the threshold level approaches. | The reviewer has a reasonable point here. In the revised manuscript, we will change the threshold values into P80 for the threshold methods (VT, FT) and SSI≤-0.84 (~P80) in order to have a fair comparison (see our reply 2a, i). |
| 3c | Why the gamma distribution to derive the SSI? I agree that is hard to find a suitable distribution to fit to riverflow time series (line 150-151). However, that is not a good argument to simply use the Gamma distribution. There are likely to be better | We used the gamma distribution to derive the SSI because the gamma distribution has been used for hydrological forecasting of both high and low flows (Slater and Villarini, 2018, **P5L149-151**). The reviewer also recognized that it is hard to find a suitable |

| | | |
|---|---|---|
| | alternatives for your pan-European dataset (See e.g. Svensson et al., 2016, Tijdeman et al., 2020). Why no goodness of fit testing? The studies above conclude on different suitable candidate distributions for the SSI (other than the gamma distribution) that might be applicable for the current study. However, that does not mean that they can be applied on your dataset of simulated streamflow series by default, as your dataset might exhibit different properties as compared to the observed riverflow timeseries. Careful evaluation which distribution is most suitable for your set of rivers is required. Which distribution fitting method was use? | distribution to fit all streamflow regimes in Europe (see also Vicente-Serrano et al., 2012). Moreover, no single distribution fits well with all monthly streamflow data in all river grid cells (n=~10,106), e.g., sample properties of streamflow in January might differ from those in August in all places (Tijdeman, et al., 2020). Our study does not focus on the selection of the best distribution for drought forecasting. We do not believe that another distribution (or other distributions) that consider differences in streamflow regime across Europe will change the main message of the study, i.e. that the outcome of the hydrological drought forecast depends on the identification method. Thus we believe it is better to simply use the widely selected gamma distribution in our analysis. |
| 3d | For the forecasted SSI: Did you use the parameters of the population distribution derived from historical monthly flow values to derive the SSI for forecasted values? Or did you replace the historical values with forecasted values and than recalculated the population distribution to derive the SSI? And why, e.g., what should a forecaster do? | We used the distribution parameters derived from the observed (historic) datasets to identify the forecasted drought. Using this method, the gamma distributions were calculated from long time series of observed data, in our case 29 years, and then applied to the forecasted streamflow (Sutanto et al., 2020a, Figure A1). We did not calculate the distribution from the re-forecast datasets because the re-forecasted time series that we have are rather short (9 years) and obviously it is not the actual observed streamflow. We will add information in the revised manuscript. |
| 4a | **Threshold approach:** Line 123-143: Many different smoothing procedures have been applied in combination with the threshold level method. This has been done for good reason, however, sometimes resulting in an (unwanted) increase/decrease in drought occurrence, especially for the VT method. For me, a 10th percentile implies that 10% of the time series is in drought and that drought occurrence is equally distributed over the year in case of the VT method. However, by first deriving the threshold from daily streamflow data, and then smoothing both the threshold and riverflow timeseries seperately, this is not necessarily the case anymore. This might be solved relatively easily, i.e., first apply the moving average and then derive the threshold. Or you could use monthly data. | In our paper we used the moving average of the daily quantile approach (D_MA, Beyene et al., 2014) to obtain VT thresholds. In the revised manuscript, we will change the method on how we calculate the VT thresholds. We will use monthly streamflow data to derive the monthly threshold and then we assign the monthly threshold level to each day of the month. When confronting time series of daily data (observed data, 1990-2018, and re-forecasted data 2003) with monthly threshold levels (only relevant for the VT application using a daily resolution, see our reply 2a, iii), jumps between two consecutive months might result in unrealistic drought behavior that extends around the beginning and end of each month. Therefore, we apply a 30 days centered moving average to the discrete monthly thresholds, as done, for instance, by Beyene et al. (M_MA, 2014); Van Loon et al. (2012); Van Lanen et al. (2013); Van Huijgevoort et al. (2014); Heudorfer and Stahl (2017); Van Tiel et al. (2018). |
| 4b | Line 366-367: You encourage using monthly streamflow data for drought | We will add the monthly drought analysis derived from the FT and VT thresholds, as |

| | | |
|---|---|---|
| | forecasts but use daily streamflow in your own analyses. I would have find it logical to do this as well in this study, e.g., instead of the FT and VT approaches applied on daily data, it could be applied monthly averaged data. This also increases the comparability with the SSI. Further, is there really merit in forecasting streamflow drought duration and deficit at a daily resolution, especially for the longer lead-times? Is this being done somewhere? Can this be done with any skill? If not, wouldn't it be better to just stick to monthly data for which at least some skill might be achieved? | additional analysis to the daily resolution to enable comparison with the SSI-1 forecast. However, we will also keep the daily analysis in our revised manuscript because the daily streamflow data is commonly used in many studies using the threshold methods (see our reply 2a, iii), incl. hydrological drought projections (Prudhomme et al., 2014; Wanders and Van Lanen, 2015; Wanders et al., 2015). |
| 5a | **Results and discussion:**
Section 3.2. The forecasting section, which is the most the novel part of this paper, would benefit from some more attention. Figure 6 provides a nice illustration, even though it might be a little obvious at this point in the papers that drought characteristics derived with different methods will vary, given that you apply a different threshold on the same forecast data. However:
- I disagree that the drought of 2003 in the river Rhine started in August 2003. According to the SSI-1, river levels dropped to below normal anomalies much earlier. I suggest to start earlier in the year.
- Why not add the observed hydrograph to the plot?
- Isn't the fact that the VT method does not forecast a drought a good thing? According to this method, there was also no drought in the observed hydrograph (Fig. 4a) – how could this method have "performed better" (line 340).
- Why not show the SSI-1 here? | We would like to thank the reviewer for his/her valuable suggestions. We will extend the forecast results in Figure 6 with the series of 12 forecasts initiated from January to December 2003 with a lead time 7-month ahead (see our reply number 2b) including the observed streamflow. In the revised manuscript, instead of Figure 6, we will present the forecasted drought characteristics (occurrence, timing, duration, and deficit volume) using different identification approaches (daily FT and VT, monthly FT and VT, and SSI-1) for the Rhine River as a table. The VT method performs better than FT (Fig. 6a) since it is in a good agreement with the observed data (Fig, 4a). |
| 5b | Given the focus of the paper on river flow forecasts, I would expect more focus on the latter, and not only an exemplary timeseries river flow forecasts for one river / event. It would be interesting to include.
- At least, an evaluation and discussion of the spread in streamflow forecast and especially in the spread in streamflow drought forecast, and (i.e., not only the evaluation of the median forecast). What are the ranges in drought characteristics derived from the forecast ensemble?
- Consequently an evaluation or discussion of the streamflow (drought) forecasts skill, i.e., can certain "types of droughts", e.g., FT vs. VT vs. SSI, be forecasted better?
The above evaluation would benefit the consideration of multiple rivers, drought events, or start months. | We would like to thank the reviewer for the suggestions. We will extend the analysis by providing: (i) maps displaying forecasted drought timing and duration across Europe using forecast data issued in August 2003, and (ii) a table describing forecasted drought characteristics (occurrence, timing, duration, and deficit volume) using a series of 12 forecasts initiated from January 2003 to December 2003 with a lead time of 7-month (median ensemble) (see also our reply number 2b). An analysis of the forecast using different drought identification methods for several European rivers is beyond the scope of this paper. We believe that the map showing the pan-European pattern (see item i, above, point 5b) will make clear that the example of the Rhine River is sufficiently representative. In addition, we will also provide information on number of ensemble |

| | | members for which drought was forecasted (x ensembles out of 25). We would like to stress that the evaluation of forecast skill using SSI and threshold method (VT) is beyond the scope of this paper. This was published in previous papers (Van Hateren et al., 2019; Sutanto et al., 2020b). |
|---|---|---|
| 5c | Again, I would avoid the SSI-6 here, due to the strong autocorrelation of this index, which makes it relatively easy to forecast on short lead times. For example, for a forecast with a lead-time of 1 month, 5 out of 6 months are already known. Rather, I would look at the SSI 1. | As said above, we will replace the SSI-6 with SSI-1 in the main text (see our reply 2a, ii and our reply 3a). |
| 6a | Finaly, some (non-committal) suggestions for Section 3.1 that could further improve the manuscript:
• Section3.1.1 Next to showing the amount of streamflow droughts, you could consider showing other characteristics such as the average duration, deficit volume, or the number of minor drought events. This provides valuable insights in differences between methods, and further makes the notions in 3.3.1 about regions with more minor drought quantitative. In addition, you can derive a proxy for deficit volume from standardized time series. The units are meaningless and not comparable with the deficit volumes derived with FT and VT method. However, the relative difference over Europe should pop-up. | We thank the reviewer for the suggestions. We will add the drought duration derived from the FT, VT, and SSI approaches in the revised manuscript. However, the SSI drought deficit volume will not be added because it is impossible to derive the deficit volume using the SSI approach (major drawback of standardized approaches). |
| 6b | • Section 3.1.2 In addition to discussing when most drought starts, it might be interesting to see when most drought occur in difference climates. This can be presented as a series of histograms for each climate, with the month on the x-axis and the fraction of drought months that occurred in that month on the y-axis. | This is an interesting suggestion. In the revised manuscript, we will provide a summary of drought characteristics (number of drought occurrence/frequency, timing, duration, and deficit volume) for 5 Köppen Geiger climate regions identified using different approaches (daily FT and VT, monthly FT and VT, and SSI-1). |
| 7 | **Minor comments:**
Line 2: "... the term streamflow drought forecasting, rather than streamflow forecasting ..." You could briefly explain difference between the two here.
We will add one sentence to describe streamflow drought forecasting in the revised manuscript. | We will add one sentence to describe streamflow drought forecasting in the revised manuscript. |
| | Line 5: "within" Correct? | We will replace "within" with "of". |
| | Line 6: Be careful with terming these extreme events. They are anomalies, but something that happens on average at least once every year, as is the case in your study, is not an extreme event. | Naming of extreme events has always a sense of subjectivity. We suggest to stick to the definition extreme event because we identify a drought event if the streamflow falls below the P80. Droughts are like floods called extreme events. |
| | Line 7, 8: "observed" might be "observations" | We will change the word accordingly. |

| | | |
|---|---|---|
| | Line 7: "a LISFLOOD model"... are there more? | There is only one LISFLOOD model. We will change "a" in "the LISFLOOD model". |
| | Line 10: add method to VT and FT, e.g. variable threshold level method. | The word "method" will be added in the revised manuscript. |
| | Line 10: You also apply a threshold based approach on SSI time series. Mention this here. | An explanation about threshold to identify drought in SSI will be added. However, we will do this in the Methods section. Threshold-based drought indices (called deficit characteristics in Hisdal et al., 2004) are fundamentally different from the standardized -based drought indices (Van Loon, 2015). |
| | Line 16: "Eliminate". Not true. You can still have 1-day droughts with these TL approaches. | We will change the word "eliminate" into "minimize". |
| | Line 24: "IPCC" should be "The IPCC". | Thanks for the correction. |
| | Line 34: This sentence slightly contradicts with Line 1, where you state that drought forecasting is a key element of DEWS. I would expect there to be some examples. Which contemporary "DEWS" include streamflow drought forecasting, using the approaches as described in the paper (FT, VT and SSI), not just streamflow forecasting)? | We will revise L34 to avoid possible contradiction, i.e. "One of the elements to be included in a NDPP is a Drought Early Warning System that in addition to real-time monitoring contains ...". In the preceding sentence we will explain the abbreviation NDMP (National Drought Policy Plan). Furthermore, streamflow drought forecasting, using all the approaches as described in the paper (FT, VT and SSI) are developed in the EU H2020 ANYWHERE project (for background, see Sutanto et al., 2020a). |
| | Line 41: "evaporation" should be potential evapotranspiration
Line 47: "used" should be "be used"
Line 85: "Proxy" should be "Proxies" | We will revise the text accordingly. |
| | Line 49: Mention that you specifically focus on simulated streamflow drought. | We will change "hydrological drought forecasting" into "streamflow drought forecasting". |
| | Line 75: "There" should be "There is" | We will remove the word "There". Thus the sentence becomes: "...., which demonstrates that no one fits all.....". |
| | Line 89: "proxy observed streamflow" could just be "simulated streamflow" | We would like to keep the term "proxy observed streamflow" to indicate that in principal people would like to use observed data, but these spatio-temporal streamflow observed flow data do not exist. Hence, flow data obtained from a hydrological model driven by observed weather data are used as proxy for observed (same as EFAS-WB in Arnal et al., 2018 or offline simulation in Yuan et al., 2017). This is similar to reanalysis data that are a proxy for observed weather. In some cases these simulated data are just called observed, which we think should be avoided. |
| | Line 112: "re-forecasted data 2003" should be "re-forecasted data of 2003"
Line 119: "in" should be "for"
Line 147: "median" should be "expected | We will change the text accordingly. |

| | | |
|---|---|---|
| | median".
Line 179: "definitions" ... "drought identification approaches" might be better.
Line 221: "drought that has" should be "droughts that have" | |
| | Line 128: "were moving averaged" rephrase | The sentence will be corrected. |
| | Line 134: "For the threshold" ...this refers to variable threshold approach I guess? In this section, make the clear distinction between FT and VT and seperately explain how both are derived. | The threshold here refers to both FT and VT. We will revise the sentence. |
| | Line 138-140: add here that MA introduces a significant amount of auto-correlation, which affects the skill of the river flow forecast for the first 30 days significantly. | We will add an explanation about the effect of 30DMA on the forecast skill. |
| | Line 155-160: Add here that it is quite easy to forecast the SSI-6 for short lead times, given the strong autocorrelation of the timeseries. E.g., for 1-month lead-times, you already know five months and only have to forecast one. | We will replace the SSI-6 with SSI-1 in the main text, thus the explanation of preceding observed data is not necessary there. |
| | Line 162-164: Please explain how you classify an event with varying SSI values into one category. | In our study we only focused on the median ensemble and not the whole ensemble (25 members). Thus if the median value of SSI is in between -1 and -1.5, we classify the event as moderate drought. |
| | Line 162-177: Did you derive the climate classification yourself using the approach described in Peel et al (2007)? Or did you use their dataset? | We used their dataset. |
| | Line 188: "Lower than median streamflow" ... Not necessarily true. Technically, above median streamflow can still be a negative SSI and vice versa. Depends on the sample and (goodness of fit) population distribution to derive the SSI. | We will use the threshold SSI<-0.84 to identify SSI drought in the revise manuscript. |
| | Line 189: Figure 3 does not show that streamflow droughts occur every year. | Figure 3 shows the drought timing and not drought occurrences. The latter we show in Fig. 2. |
| | Line 200: This is comparing apples and pears, as the thresholds are completely different. | We will change the threshold values, i.e. special application of VT and FT thresholds, for better comparison in the revised manuscript (see our reply 2a, i). |
| | Line 203-206: Could this not be compensated by a higher number of drought in winter for the VT? | Sorry, we have to disagree. The VT method takes into account the seasonality. |
| | Line 228. "(Coincides with hydrologic years in most of Europe)" remove: unneeded repetition.
Line 264-266. Is the last part, i.e., about the lowest and n-day minimum flow, needed? Interrupts flow. | We will remove the sentence. |
| | Line 266-267. Looking at Fig. 5a, I find the SSI-1 timeseries much more informative about drought in the river Rhine. Rhine drought reaches is maximum in summer 2003, and recovers in winter 2004. For me, this make much more sense than the SSI-6 | In the revised manuscript, we will use only the SSI-1 forecasts rather than the SSI-6. Drought 2003 in Europe was one of the severe drought events and this was applied to the Rhine River as well. The impact of 2003 drought on the Rhine flow was |

| | | |
|---|---|---|
| | timeseries. Was the drought in the river Rhine really a multiyear event? Were there impact directly related to Rhine river flows over the course of 2004? | apparent but we are not aware if there was an impact in 2004 meaning that there would have been a multi year drought. |
| | Line 270. For me, this description of drought in the river Rhine makes much more sense. It would make even more sense if you would use a more appropriate drought threshold (maybe SSI-1 < -0.84, corresponding to the 20th percentile). I don't see the problem of having 2003 split up in different events and question why it is better to use an SSI-6 and thereby inflate the event to a multiyear drought. | We do agree with the reviewer and therefore we will revise the manuscript by using the drought threshold SSI<-0.84 and only using SSI-1 (see our reply 2a, i). |
| | Line 285: "C" should be century. Line 361: "rare extreme drought events" ... extreme events are by definition rare. Rephrase. | Typo will be corrected. |
| | Line 295-302. Why limit yourself here to the four case study Rivers and the limited time window? You could directly compare the number of drought events & their deficit volumes over a longer time period and for all the catchments (starting by deriving the difference between Fig. 2a and b). | We do agree with the reviewer and, as stated above, we will extend our analysis by providing drought duration using different approaches. The limited time window in Figure 4 was made to increase the readability. This was done by showing 2003 drought events in north, central, and east Europe and 2005-2006 droughts in south Europe. |
| | Line 312-329. According the definition of drought according to VT and the SSI, droughts are expected to occur for an equal amount of time over the year. Please provide an explanation for the distinct temporal differences in drought occurrences. Or is this still referring to the start month of the drought? | Yes, we refer to the drought timing, which is identified when drought mostly started. In the revised manuscript we will use in a specific application of the FT and VT methods monthly streamflow data, thus in that case there is no discrepancy in the temporal resolution between threshold methods and SSI. |
| | Line 309: "(except for the Rhine River)" this contradicts with the discussion in the paragraph above. | For the selected river basins (Section 3.1.4), we did the analysis only for the selected river grid cells. The discussion in the paragraph before this section was for the whole of Europe in general. |
| | Line 337: Not only meteorological drought, also streamflow drought according to the SSI-1 (Fig. 5a). | We do agree with the reviewer, we will revise the manuscript accordingly. |
| | Line 354. Which is good, because there was no drought according to the VT, or? | Here the VT method did not forecast drought in August 2003 using the 30DMA. The 30DMA, however, is very useful in reducing minor drought events and it is also recommended to increase the forecast skill (**P12L354-360**). |
| | Line 382: "eliminate" ... not correct as minor droughts can still occur. | We will change the word into "minimize". |
| | Line 372-373: "the FT method produces higher drought deficit volumes and duration than VT" not shown for the pan-European dataset. | We will add drought duration and deficit volume in the revised manuscript. |
| | Line 375: "occurred" should be "started". | We will change the word accordingly. |
| | Line 377: "what being identified by" rephrase | We will revise the sentence into: what is being identified. |

| | | |
|---|---|---|
| | Figure 1: Nice. What is the difference between light and dark grey in e.g., the Alps? | We only have grey color for ET region (Alps). |
| | Figure 2: You could add the upper boundary, e.g. 30-xx instead of >30. | We will revise the legend accordingly. |
| | Figure 3: "The timing for drought was determined based on the first month of each drought event." This is the same as what is said in the beginning of the caption. | We will remove the last sentence to avoid duplication. |
| | Figure 4: Some droughts are hardly visible (e.g. in Figure 4a). It might work to use a log-scale Figure 4: Axis lables: m3 sec-1 or m3 / sec instead of m3/sec | We will change the axis into $m^3 \, sec^{-1}$. |
| | Figure 4: Are the grey vertical lines the hydrological years? | We will add an explanation for the grey vertical lines. |
| | Figure 4. You might consider using a different color when VT and FT overlap. | We will revise the color as suggested. |
| | Figure 5. Add grey vertical lines here as well. | We will add the grey vertical lines. |
| | Figure 6. Same comments as for Figure 4 and 5. | We will revise the figure accordingly. |
| | Table 1. Would be interesting to also compare average deficit volume and timing. | We will add drought duration and deficit volume in Table 1. |

References:
1. Hisdal, H., Tallaksen, L. M., Clausen, B., Peters, E., and Gustard, A.: Hydrological Drought Characteristics. In: Tallaksen, L. M. & Van Lanen, H. A. J.(Eds.) Hydrological Drought, Processes and Estimation Methods for Streamflow and Groundwater. Development in Water Science 48, Elsevier Science B.V., pg. 139-198, 2004.
2. Van Loon, A. F.: Hydrological drought explained, WIREs Water, https://doi.org/10.1002/wat2.1085, 2015.
3. Vicente-Serrano, S. M., López-Moreno, J. I., Beguería, S., Lorenzo-Lacruz, J., Azorin-Molina, C., and Morán-Tejeda, E.: Accurate computation of a streamflow drought index, J. Hydrol. Eng., 17(2), 318-332, doi:10.1061/(ASCE)HE.1943-5584.0000433, 2012.
4. Tijdeman, E., Stahl, K., and Tallaksen, L. M.: Drought characteristics derived based on the Standardized Streamflow Index: A large sample comparison for parametric and nonparametric methods, Water Resources Research, 56, e2019WR026315, https://doi.org/ 10.1029/2019WR026315, 2020.
5. Tallaksen, L.M., Hisdal, H., and Van Lanen, H.A.J.: Space-time modeling of catchment scale drought characteristics, Journal of Hydrology, 375, 363–372, 2009.
6. Slater, L.J., and Villarini, G.: Enhancing the predictability of seasonal streamflow with a statistical-dynamical approach, Geophysical Research Letters, 45(13), pp.6504-6513, 2018.
7. Sutanto, S.J., Van Lanen, H.A.J., Wetterhall, F., Llort, X. (2020a). Potential of pan-European seasonal hydro-meteorological drought forecasts obtained from a Multi-Hazard Early Warning System. Bulletin of the American Meteorological Society. https://doi.org/10.1175/BAMS-D-18-0196.1.
8. Beyene, B.S., Van Loon, A.F., Van Lanen, H.A.J., and Torfs, P.J.J.: Investigation of variable threshold level approaches for hydrological drought identification, Hydrol. Earth Syst. Sci. Discuss., 11, 12765–12797, 2014

9.  Van Loon, A. F., Van Huijgevoort, M. H. J., and Van Lanen, H. A. J.: Evaluation of drought propagation in an ensemble mean of large-scale hydrological models, Hydrol. Earth Syst. Sci., 16, 4057-4078, doi:10.5194/hess-16-4057-2012, 2012.

10. Van Lanen, H. A. J., Wanders, N., Tallaksen, L. M., and Van Loon, A. F.: Hydrological drought across the world: impact of climate and physical catchment structure, Hydrol. Earth Syst. Sci., 17, 1715–1732, https://doi.org/10.5194/hess-17-1715-2013, 2013.

11. Van Huijgevoort, M. H. J., Van Lanen, H. A. J., Teuling, A. J., and Uijlenhoet, R.: Identification of changes in hydrological drought characteristics from a multi-GCM driven ensemble constrained by observed discharge, Journal of Hydrology, 512, 421-434, doi:10.1016/j.hydrol.2014.02.060, 2014.

12. B Heudorfer, K Stahl: Comparison of different threshold level methods for drought propagation analysis in Germany, Hydrology Research, 2017, 48 (5): 1311–1326. https://doi.org/10.2166/nh.2016.258.

13. Van Tiel, M., Teuling, A. J., Wanders, N., Vis, M. J. P., Stahl, K., and Van Loon, A. F.: The role of glacier changes and threshold definition in the characterisation of future streamflow droughts in glacierised catchments, Hydrol. Earth Syst. Sci., 22, 463–485, https://doi.org/10.5194/hess-22-463-2018, 2018.

14. Prudhomme, C., Giuntoli, I., Robinson, E. L., Clark, D. B., Arnell, N. W., Dankers, R., Fekete, B. M., Franssen, W., Gerten, D., Gosling, S. N., Hagemann, S., Hannah, D. M., Kim, H., Masaki, Y., Satoh, Y., Stacke, T., Wada, Y., and Wisser, D.: Hydrological droughts in the 21st century, hotspots and uncertainties from a global multimodel ensemble experiment, Proc. Natl. Acad. Sci., 111, 3262–3267, doi:10.1073/pnas.1222473110, 2014.

15. Wanders, N. and Van Lanen, H. A. J.: Future hydrological drought across climate regions around the world modelled with a synthetic hydrological modelling approach forced by three General Circulation Models, Nat. Hazards Earth Syst. Sci, 15, 487–504, doi:10.5194/nhess-15-487-2015.

16. Wanders, N., Wada, Y. and Van Lanen, H.A.J.: Global hydrological droughts in the 21st century under achanging hydrological regime, Earth Syst. Dynam., 6, 1–15, doi:10.5194/esd-6-1-2015, 2015.

17. Van Hateren, T., Sutanto, S. J., and Van Lanen, H. A. J.: Evaluating uncertainty and robustness of seasonal meteorological and hydrological drought forecasts at the catchment scale-case Catalonia (spain), Env. Int., 133, 105206, https://doi.org/10.1016/j.envint.2019.105206, 2019.

18. Sutanto, S.J., Wetterhall, F., Van Lanen, H.A.J. (2020b): Hydrological drought forecasts outperform meteorological drought forecasts. Environment Research Letters, 15: https://doi.org/10.1088/1748-9326/ab8b13.

19. Arnal, L., Cloke, H. L., Stephens, E., Wetterhall, F., Prudhomme, C., Neumann, J., Krzeminski, B., and Pappenberger, F.: Skilful seasonal forecasts of streamflow over europe?, Hydrol. Earth Syst. Sci., 22, 2057–2072, https://doi.org/10.5194/hess-22-2057-2018, 2018.

20. Yuan X., Zhang, M., Wang, L., and Zhou, T.: Understanding and seasonal forecasting of hydrological drought in the Anthropocene, Hydrol. Earth Syst. Sci., 21, 5477–5492, doi:10.5194/hess-21-5477-2017, 2017.

---

## Author Comment (AC2) · 30 Nov 2020

**Reply to reviewer 2**

We would like to thank the reviewer for valuable suggestions and comments. In this document, **P** refers to the page number and **L** refers to the line number in the recent paper. For example, **P3L65-70**, refers to page 3, lines 65-70.

| Reviewer 1 | | |
|---|---|---|
| **No** | **Comment** | **Reply** |
| 1 | The authors performed an intercomparison of three different streamflow drought indicators, with the goal to highlight the differences in the drought characteristics associated to each index and to detail the implication on drought forecast. I found the overall goal of the study meaningful, given the confusion that still arise among scientists and operational users on the topic, but I also found the paper and its structure generally out of focus. The key message of the paper "….developers of DEWS and end-users should clearly agree among themselves upon a sharp definition on which type of streamflow drought is required to be forecasted for a specific application." is in my opinion, even if relevant, better suited for a short communication or letter paper rather then a research paper. | We would like to thank the reviewer for the acknowledgement of the goal of our study. We appreciate the suggestion from the reviewer that our manuscript is better suited for a short communication paper rather than a research paper. However, a short communication paper would only be an option, if a systematic intercomparison of threshold and standardized streamflow drought indices across Europe obtained from commonly used identification approaches would exist. Such intercomparison, however, does not exist and consequently a technical paper is needed, which describes and discusses the use of different drought identification approaches to derive streamflow drought across Europe. This has to precede the section that deals with the implication on drought forecasting. Hence this paper should be a technical research paper instead of a short communication paper. |
| 2 | The research results that should support this conclusion as reported in this paper are somewhat lacking in both clarity and rigorousness. | We believe that the conclusions of our study support the results that different drought indices generate different number of drought occurrences/frequency and timing, which are strongly related to climate regions. We believe that we improved clarity and rigorousness of our results in the revised manuscript through making the drought identification methods more consistent in terms of: (i) thresholds, (ii) data accumulation period, and (iii) temporal resolution (see our reply number 3 below for more details). |
| 3 | The main drawback of the analysis is the fact that the authors uses three drought indicators that rely on quite different input data and basis hypotheses to conclude that they provide a different picture of drought. This result is quite obvious after an attentive read, given the background premises: - daily data for threshold methods vs. monthly data for SSI. - 90th percentile for threshold methods vs. median for SSI (SSI=0). - Event-based approach for threshold vs. single monthly value for SSI All these discrepancies in the drought definition make the intercomparison a mere exercise, and its outcomes are hard to translate into actual | We would like to thank the referee for the comments and valuable suggestions. Our paper uses the drought threshold based on common practice in the drought community. Using a threshold method either a Fixed Threshold (FT) or Variable Threshold (VT), drought is identified if the streamflow falls below the threshold, which is commonly in the range of 10-30th percentile of flow duration curve (P70-90)(Hisdal et al., 2004; Van Loon, 2015). On the other hand, the standardized indices e.g., the Standardized Streamflow Index (SSI) identifies drought if the SSI value falls below 0, which is 50th percentile (P50). Our reason that we use different thresholds (50th percentile for SSI |

| | | |
|---|---|---|
| | general considerations. | and 10th percentile for threshold for the VT and FT) is that we would like to follow common practice for the different approaches. However, the reviewer has a point that the comparison between threshold methods (VT and FT) and SSI is not equal regarding to the use of different percentiles. Thus, in the revised manuscript, we will change the threshold from P90 into P80 for VT and FT, and SSI≤-0.84 (~P80) to have a fair comparison between different drought indices (Tijdeman et al., 2020). We also agree with the reviewer that our study uses different temporal resolutions to analyze drought, which are daily for the threshold methods and monthly for SSI. Again, we followed the common practice to identify drought using these methods. Many studies used daily streamflow data to analyze drought using the threshold method and monthly streamflow data to analyze drought using the standardized indices. To the author's knowledge, only Tallaksen et al. (2009) used the monthly data to derive drought using the threshold method only for a scientific purpose. In the revised manuscript, we will add in a specific application an analysis of drought characteristics using monthly streamflow data in both FT and VT drought approaches. |
| 4 | An additional drawback is the general lack of details on the implementation of the three approaches, which severely limits the possibility for the readers to extrapolate meaningful information from the research outputs. | We will elaborate more the method section and will add more drought characteristics, such as drought duration and deficit volume using different methods in the revised manuscript. By adding more results especially in the forecasting section (Section 3.2) (see our reply 5 below and 6c), the reader hopefully will clearly see the differences in drought characteristics because of different drought identification methods. |
| 5 | Finally, the analysis on the implications on drought forecast, which should be the main focus of the paper according to the title, is very limited in scope, and it needs to be significantly expanded in order to keep it as the focus of the paper. | We thank the reviewer for the suggestion. We will expand the analysis using the series of 12 forecasts initiated from January 2003 to December 2003 with 7 months lead time for each initiation. We will do this by describing: (i) pan-European maps showing forecasted drought timing and duration (number of drought occurrence/frequency and drought deficit volume will be provided in the Supplementary Material), and (ii) summary of forecasted drought characteristics identified using different approaches (FT and VT with daily and monthly resolution, and SSI-1) in the Rhine River using the series of forecasts initiated from 1st January 2003 to 1st December 2003 with a lead time of 7-month. |
| 6a | **Specific Comments** | We will briefly describe in the Introduction |

| | | |
|---|---|---|
| | **Introduction**
The authors should better highlight how different definitions of streamflow drought in DEWS exists also for two reasons: 1) different users have different needs that can be accommodate by different indicators (e.g. river navigation may be affected more by FT droughts that VT droughts), 2) different available input data lead to different definitions (e.g. threshold methods may not be suitable for monthly data, and daily data may not be available in near-real time). | why different definitions of streamflow drought exists in DEWS. However, would like to leave the decision of using which drought identification approach to the users. We explained this in the Conclusions (**P12L383-P13L388**) where we stated, "*The use of monthly-aggregated forecasted flow data (e.g. SSI) is the best practice for seasonal drought forecasts. This method, however, cannot be used to calculate the drought deficit volume, which is a key component for water managers coping with hydrological drought. If deficit volumes are required for decision-making, then threshold approaches (VT or FT) should be applied on 30-day averaged flow data. The choice of the drought identification method when forecasting streamflow drought, in the end, lies to the end-users specific requirements and decisions and there is no one drought identification approach that fits all needs*". |
| 6b | **Data and Methods**
The description of the different drought indices need to be more explicit. How the drought events are defined for each index? How is the onset computed? Severity? Duration? Any event definition in the SSI? Etc. . . Also, more consistency on the adopted thresholds need to be enforced (why SSI=0 is used as threshold when 90th percentile is used for VT and FT?). It is also worth to mention that a VT method based on the same LISFLOOD data is currently operationally implemented as part of EDO (https://edo.jrc.ec.europa.eu/). | We will expand the method section in the revised manuscript, as suggested. We will add some information on drought characteristics in the method section, such as drought timing or onset (month when drought starts), number of drought occurrences/frequency, duration, and deficit volume. As mentioned above, we will change the drought threshold into P80 for FT and VT and SSI≤-0.84 (~P80) for the standardized index (our reply number 3). The suggested information about the VT method applied in EDO will be added. |
| 6c | **Results and discussion**
There is a clear unbalance between the historical analysis and the forecast. Give the title of the paper, I would aspect much more emphasis on the latter. | We will extend our forecast analysis by providing: 1) a map displaying forecasted drought timing and duration across Europe using forecast data issued in August 2003, and 2) a table describing forecasted drought characteristics (occurrence, timing, duration, and deficit volume) using the series of 12 forecasts initiated from January 2003 to December 2003 with a lead time of 7-month (median ensemble) (see our reply number 5). The latter, however, can only be performed only for one river. In addition, we will also provide information of number of ensemble members indicating drought in percent (x ensembles out of 25). |

References:

1. Hisdal, H., Tallaksen, L. M., Clausen, B., Peters, E., and Gustard, A.: Hydrological Drought Characteristics. In: Tallaksen, L. M. & Van Lanen, H. A. J.(Eds.) Hydrological Drought, Processes and Estimation Methods for Streamflow and Groundwater. Development in Water Science 48, Elsevier Science B.V., pg. 139-198, 2004.

2. Van Loon, A. F.: Hydrological drought explained, WIREs Water, https://doi.org/10.1002/wat2.1085, 2015.
3. Tijdeman, E., Stahl, K., and Tallaksen, L. M.: Drought characteristics derived based on the Standardized Streamflow Index: A large sample comparison for parametric and nonparametric methods, Water Resources Research, 56, e2019WR026315, https://doi.org/ 10.1029/2019WR026315, 2020.
4. Tallaksen, L.M., Hisdal, H., and Van Lanen, H.A.J.: Space-time modeling of catchment scale drought characteristics, Journal of Hydrology, 375, 363–372, 2009.

---

## Author Comment (AC3) · 7 Dec 2020

Attached, we provide a summary of the author's response to reviewer 1 and reviewer 2. In general, both reviewers are concerned about the methodology used in our manuscript and the unbalance in results between historic analyses and forecast. Here, we will provide a detailed author's response only for some important remarks that were raised during the review process, including some new results. We already provide some concrete results that we promised in the reply to reviewers. Please consult the previous author's response addressed to the individual reviewer for detailed information on the reply. Figures are also attached below for better readability.

[Figure]

Please also note the supplement to this comment:
https://hess.copernicus.org/preprints/hess-2020-458/hess-2020-458-AC3-supplement.pdf

Number of event

0    8    16   24   32   40   48   56   64   72   80

**Fig. 1.** Drought occurrences in European rivers

Duration (day)

0   20   40   60   80   100   120   140   160   180   200

Duration (month)

1        2        3        4        5        6        7

**Fig. 2.** Forecasted average duration of drought events

**Fig. 3.** Observed (SFO) and forecasted streamflow for 25 ensemble members and median streamflow in the Rhine River

**Supplement:**

**Streamflow drought: implication of drought definitions and its application for drought forecasting**

Samuel J. Sutanto[1,*)] and Henny A. J. Van Lanen[1)]

[1)]Hydrology and Quantitative Water Management Group, Environmental Sciences Department, Wageningen University and Research, Droevendaalsesteeg 3a, 6708PB, Wageningen, the Netherlands

**Executive Summary of author's response to reviewer 1 and 2**

In this file, we provide a summary of author's response to reviewer 1 and reviewer 2. In general, both reviewers are concerned about the methodology used in our manuscript and the unbalance in results between historic analyses and forecast. Here, we will provide detailed author's response only for some important remarks that were raised during the review process, including some new results. We already provide some concrete results that we promised in the reply to reviewers. Please consult previous author's response addressed to individual reviewer for detailed information on the reply. We summarize the main reviewer's concerns as follow:

1. The use of non-identical thresholds to identify drought in our paper, which are P90 for the variable and fixed threshold methods, and SSI<0 for the standardized approach. The use of the SSI threshold of zero to identify drought is not equal with P90 used in the threshold approaches, meaning that we compare droughts that occurs in 50% of the time (SSI) with the ones in 90% of the time (threshold method).

2. The use of SSI-6 instead of SSI-1 in the main text. The reviewers argued that river flow already encompasses the accumulation and delay of the meteorological signal caused by catchment properties, such as groundwater flow.

3. The use of different temporal resolution, namely daily data for the threshold approaches and monthly data for the SSI. The reviewers suggest to also use monthly streamflow data for the threshold methods to increase the comparability with the SSI, which uses monthly resolution.

4. The unbalance in results between historic analysis and forecasts. The reviewers suggest to include an evaluation and discussion of the spread in streamflow drought forecasts and to elaborate the forecast section (3.2) more.

5. Adding more drought characteristics, such as drought duration and severity (deficit volume). Moreover, the reviewers suggest to derive the drought deficit volume from the standardized time series.

6. The reviewers suggest to summarize the drought characteristic results for each climate regions.

We would like to thank both reviewers for the comments and the valuable suggestions to improve our manuscript. We do agree with the reviewer's main suggestions and therefore we will revise our paper as follow:

1. Our paper uses the drought thresholds based on common practice in the drought community. Using a threshold method either a Fixed Threshold (FT) or Variable Threshold (VT), drought is identified if the streamflow falls below the threshold, which is commonly in the range of 10-30th percentile of the flow duration curve (P70-90) (Hisdal et al. 2004; Van Loon, 2015). On the other hand, the standardized indices, e.g., the Standardized Streamflow Index (SSI) identifies drought if the SSI value falls below 0, which is the 50th percentile (Vicente-Serrano et al., 2012). Our reason to use different thresholds (50th percentile for SSI and 10th percentile for the FT and VT) is that we would like to align with common practice for the different approaches. However, the reviewer has a point that the comparison between the threshold methods (VT, FT) and SSI is not equal, because of the use of different percentiles. Thus in the revised manuscript, we will change the thresholds from P90 into P80 for VT and FT, and SSI≤-0.84 (~P80) to have a fair comparison between different drought indices (Tijdeman et al., 2020). Figure 1 shows drought occurrences (frequency) in European rivers identified using different approaches and derived using the new threshold levels, which are P80 for threshold methods and SSI<-0.84 for SSI.

2. We aware that streamflow, as included SSI 1, comprises some catchment memory aspects (delayed flow from groundwater). Hence, in the revised manuscript, we will replace SSI-6 with SSI-1 in the main text (See Fig. 1e for example). However, we need to realize that anomalies in the accumulated flow over a longer period (e.g. SSI-6) have relevance for some purposes, such as the management of surface water reservoirs.

3. We do agree with the reviewer that our study used different temporal resolution to analyze drought, which are daily for the threshold methods and monthly for SSI. Again, we followed common practice (see item 1, above) to identify drought using these methods. Many studies used daily streamflow data to analyze drought using the threshold methods and monthly streamflow data to analyze drought using the standardized indices. To the author's knowledge, only Tallaksen et al. (2009) used monthly data to derive drought using the threshold method and only for a scientific purpose. In the revised manuscript, however, we will add to the common practice approach (daily resolution), an analysis of drought characteristics using monthly streamflow data in both FT and VT drought approaches. This allows an analysis of the VT and FT threshold approach and the SSI-1 using the same temporal resolution, i.e. monthly time scale. This implies that we will have two VT and FT threshold applications: daily resolution (VTD and FTD, Fig. 1a and b, respectively), as frequently used, and monthly resolution (VTM and FTM, Fig. 1c and d, respectively) to allow comparison with SSI 1 (Fig. 1e).

[Figure]

**Fig. 1.** Drought occurrences in European rivers from October 1990 to September 2018 (28 years) identified using: a) the variable threshold method with daily streamflow data (VTD drought), b) using the fixed threshold method with daily streamflow data (FTD drought), c) using the variable threshold method with monthly streamflow data (VTM drought), d) using the fixed threshold method with monthly streamflow data (FTM drought), and e) using the Standardized Streamflow Index with accumulation time 1 month (SSI-1 drought).

4. We will extend the novel part of paper to pass the important message that the outcome of drought forecasts depends on the drought identification method, which frequently is overlooked by academics and end-users. We will do this by describing: (i) pan-European maps showing forecasted drought timing and duration using different drought identification methods (FT and VT with daily and monthly resolution, and SSI-1, see Fig. 2, for example, of forecasted drought duration from July 2003 to January 2004 using the forecast initiated on July 2003 for 7-month LT) (other drought characteristics, such as number of drought occurrence/frequency and drought deficit volume will be provided in the Supplementary Material), (ii) summary of forecasted drought characteristics identified using different approaches for the Rhine River using forecasts initiated from 1st January 2003 to 1st December 2003 with a lead time of 7-month (see Table 1 and 2), and (iii) ensemble spread of forecasted drought for the Rhine River using forecasts initiated in April and July 2003 for different approaches (VTD, FTD, VTM, FTM, and SSI) (Fig. 3). In addition we will also provide information on the percentage of ensemble members showing drought for each identification method (Ne in Table 1 and 2).

5. We will add drought duration and deficit volume derived from the FT, VT, and SSI approaches in the revised manuscript (see Fig. 2, for example, of forecasted drought duration). However, the SSI drought deficit volume will not be added because it is impossible to derive the deficit volume using the SSI approach (major drawback of standardized approaches).

6. In the revised manuscript, we will provide a summary of drought characteristics (number of drought occurrence/frequency, timing, duration, and deficit volume) for 5 Köppen Geiger climate regions identified using different approaches (FTD, VTD, FTM, VTM, and SSI-1; see Table 3 and 4).

In conclusion, we agree to elaborate all major suggestions raised by the reviewers in our revised paper. We will revise the paper accordingly and will submit the revised version after the online discussion is ended. Given that we already have all the new version of the results, we will manage to revise our paper within the given time by the editor. Some Figures and Tables presented in this executive summary are taken from the draft of our revised paper. We look forward for submitting the revised paper and hope that the reviewers will agree to review our revised paper.

[Figure]

**Fig. 2.** Forecasted average duration of drought events in the European rivers using the forecast initiated on 1st July 2003 with a lead time 7-month for: a) the VTD drought, b) the FTD drought, c) the VTM drought, d) the FTM drought, and e) the SSI-1 drought. White river color indicates that drought was not forecasted.

**Table 1.** Forecasted streamflow drought characteristics derived from daily streamflow data using the VTD and FTD approaches for the Rhine River initiated from 1st January 2003 to 1st December 2003 for 7 months ahead (215 days). Drought characteristics were derived using median of the ensemble. N stands for number of occurrence, Ne stands for maximum number of ensemble members falling below drought thresholds (%), T stands for timing (month), D stands for duration (day), and DV stands for deficit volume ($m^3$)

| Forecast initiation month | Drought characteristics | | | | | | | | | |
|---|---|---|---|---|---|---|---|---|---|---|
| | VTD | | | | | FTD | | | | |
| | N | Ne (%) | T (m) | D (d) | DV ($m^3$) | N | Ne (%) | T (m) | D (d) | DV ($m^3$) |
| 1 | 0 | 20 | 0 | 0 | 0 | 0 | 20 | 0 | 0 | 0 |
| 2 | 0 | 20 | 0 | 0 | 0 | 0 | 28 | 0 | 0 | 0 |
| 3 | 0 | 20 | 0 | 0 | 0 | 0 | 48 | 0 | 0 | 0 |
| 4 | 2 | 76 | 10 | 3 | 173 | 9 | 92 | 9 | 6.2 | 433 |
| 5 | 6 | 56 | 10 | 3.8 | 204 | 3 | 76 | 10 | 24 | 4244.7 |
| 6 | 7 | 64 | 10 | 5.6 | 534 | 4 | 80 | 8 | 28.7 | 5163.2 |
| 7 | 12 | 100 | 12 | 3 | 204 | 9 | 100 | 8 | 12.7 | 1467.5 |
| 8 | 11 | 100 | 11 | 12.1 | 1819 | 8 | 100 | 1 | 15.9 | 5172.7 |
| 9 | 6 | 100 | 12 | 16.3 | 2657 | 7 | 100 | 12 | 14.7 | 4478.9 |
| 10 | 3 | 100 | 10 | 23.7 | 4295 | 4 | 100 | 11 | 20.5 | 4685.2 |
| 11 | 6 | 100 | 1 | 10.2 | 2654 | 3 | 100 | 11 | 15.3 | 5295.9 |
| 12 | 2 | 100 | 12 | 22.5 | 10346 | 1 | 100 | 12 | 43 | 14803 |

**Table 2.** Forecasted streamflow drought characteristics derived from monthly streamflow data using the VTM, the FTM, and the SSI-1 approaches for the Rhine River initiated from 1st January 2003 to 1st December 2003 for 7 months ahead (215 days). See Table 5 for symbol descriptions.

| Forecast initiation month | Drought characteristics | | | | | | | | | | | | | |
|---|---|---|---|---|---|---|---|---|---|---|---|---|---|---|
| | VTM | | | | | FTM | | | | | SSI-1 | | | |
| | N | Ne (%) | T (m) | D (m) | DV ($m^3$) | N | Ne (%) | T (m) | D (m) | DV ($m^3$) | N | Ne (%) | T (m) | D (m) |
| 1 | 0 | 8 | 0 | 0 | 0 | 0 | 8 | 0 | 0 | 0 | 0 | 8 | 0 | 0 |
| 2 | 0 | 0 | 0 | 0 | 0 | 0 | 0 | 0 | 0 | 0 | 0 | 0 | 0 | 0 |
| 3 | 0 | 8 | 0 | 0 | 0 | 0 | 28 | 0 | 0 | 0 | 0 | 16 | 0 | 0 |
| 4 | 0 | 36 | 0 | 0 | 0 | 1 | 52 | 10 | 1 | 2314.2 | 0 | 24 | 0 | 0 |
| 5 | 1 | 56 | 10 | 1 | 1160 | 1 | 60 | 9 | 2 | 10210 | 0 | 44 | 0 | 0 |
| 6 | 1 | 60 | 10 | 1 | 1407 | 1 | 72 | 9 | 3 | 12569 | 0 | 48 | 0 | 0 |
| 7 | 0 | 48 | 0 | 0 | 0 | 1 | 68 | 9 | 2 | 5689.9 | 1 | 56 | 7 | 1 |
| 8 | 2 | 100 | 8 | 2 | 6649 | 1 | 100 | 8 | 4 | 33096 | 2 | 100 | 8 | 1.5 |
| 9 | 1 | 68 | 9 | 2 | 9843 | 1 | 100 | 9 | 2 | 26095 | 1 | 92 | 9 | 2 |
| 10 | 2 | 72 | 12 | 1 | 3508 | 1 | 92 | 10 | 2 | 13212 | 1 | 72 | 10 | 1 |
| 11 | 1 | 84 | 11 | 2 | 5423 | 1 | 96 | 11 | 1 | 10246 | 1 | 80 | 11 | 1 |
| 12 | 1 | 84 | 12 | 1 | 14150 | 1 | 84 | 12 | 1 | 10785 | 1 | 64 | 12 | 1 |

[Figure]

**Fig. 3.** Observed (SFO) and forecasted streamflow for 25 ensemble members and median streamflow in the Rhine River: a) daily streamflow drought (VTD and FTD) initiated on 1st April 2003 for 7 months ahead, c) monthly streamflow drought (VTM and FTM) initiated on 1st April 2003 for 7 months ahead, and e) forecasted SSI-1 drought initiated on 1st April 2003 for 7 months ahead. b), d), and f) same as a, c, and e but for forecasts initiated on 1st July 2003. Droughts are indicated by blue shaded area for VTD and VTM, red shaded area for FTD and FTM, and purple shaded area for SSI-1.

**Table 3.** Streamflow drought characteristics derived from daily streamflow data using the VTD and the FTD methods obtained from the hydrologic years 1991 to 2018 for the five climate regions. N stands for number of events, T stands for timing (month), D stands for duration (day), and DV stands for deficit volume ($m^3$). D, and DV are average drought characteristics and T is median drought timing for all river grid cells located in each climate region

| No | River | Drought characteristics | | | | | | | |
|---|---|---|---|---|---|---|---|---|---|
| | | VTD | | | | FTD | | | |
| | | N | T (m) | D (d) | DV ($m^3$) | N | T (m) | D (d) | DV ($m^3$) |
| 1 | ET | 55.4 | 4 | 44 | 571 | 51.5 | 8 | 80 | 1112.9 |
| 2 | DFB | 48.3 | 3 | 43.8 | 1113 | 47.9 | 7 | 57.9 | 1606.2 |
| 3 | DFC | 49.2 | 3 | 46.7 | 823 | 44.4 | 10 | 91.3 | 2136.1 |
| 4 | CFB | 57.8 | 10 | 36.4 | 886 | 55.6 | 7 | 59.5 | 1494.7 |
| 5 | Med | 41 | 10 | 56.3 | 455 | 38.6 | 7 | 96.8 | 997.1 |

**Table 4.** Streamflow drought characteristics derived from monthly streamflow data using the VTM, the FTM, and the SSI-1 drought identification method obtained from the hydrologic years 1991 to 2018 for the five climate regions. N stands for number of events, T stands for timing (month), D stands for duration (day), and DV stands for deficit volume $(m^3)$. D, and DV are average drought characteristics and T is median drought timing for all river grid cells located in each climate region

| No | River | Drought characteristics | | | | | | | | | | |
|----|-------|---------|-------|-------|----------|------|-------|-------|----------|------|-------|-------|
| | | VTM | | | | FTM | | | | SSI-1 | | |
| | | N | T (m) | D (m) | DV (m³) | N | T (m) | D (m) | DV (m³) | N | T (m) | D (m) |
| 1 | ET | 28.9 | 7 | 2.5 | 1344 | 39.3 | 9 | 3.4 | 1443.1 | 35.2 | 7 | 2.2 |
| 2 | DFB | 26.5 | 5 | 2.5 | 1727 | 33.8 | 8 | 2.6 | 2126.6 | 29.5 | 5 | 2.2 |
| 3 | DFC | 25.6 | 6 | 2.5 | 955 | 35.9 | 10 | 3.6 | 2406.8 | 30 | 5 | 2.4 |
| 4 | CFB | 30.7 | 5 | 1.9 | 1495 | 38.7 | 7 | 2.7 | 2106.4 | 34.8 | 7 | 1.9 |
| 5 | Med | 22.6 | 9 | 2.9 | 690 | 32.2 | 7 | 3.7 | 1194.8 | 25.5 | 8 | 2.4 |

**References**
1. Hisdal, H., Tallaksen, L. M., Clausen, B., Peters, E., and Gustard, A.: Hydrological Drought Characteristics. In: Tallaksen, L. M. & Van Lanen, H. A. J.(Eds.) Hydrological Drought, Processes and Estimation Methods for Streamflow and Groundwater. Development in Water Science 48, Elsevier Science B.V., pg. 139-198, 2004.
2. Van Loon, A. F.: Hydrological drought explained, WIREs Water, https://doi.org/10.1002/wat2.1085, 2015.
3. Vicente-Serrano, S. M., López-Moreno, J. I., Beguería, S., Lorenzo-Lacruz, J., Azorin-Molina, C., and Morán-Tejeda, E.: Accurate computation of a streamflow drought index, J. Hydrol. Eng., 17(2), 318-332, doi:10.1061/(ASCE)HE.1943-5584.0000433, 2012.
4. Tijdeman, E., Stahl, K., and Tallaksen, L. M.: Drought characteristics derived based on the Standardized Streamflow Index: A large sample comparison for parametric and nonparametric methods, Water Resources Research, 56, e2019WR026315, https://doi.org/ 10.1029/2019WR026315, 2020.
5. Tallaksen, L.M., Hisdal, H., and Van Lanen, H.A.J.: Space-time modeling of catchment scale drought characteristics, Journal of Hydrology, 375, 363–372, 2009.

---

## Author Response (AR1)

**Reply to reviewer 1**

We would like to thank the reviewer for valuable suggestions and comments. In this document, **P** refers to the page number and **L** refers to the line number in the recent paper. For example, **P3L65-70**, refers to page 3, lines 65-70.

| No | Comment | Reply |
|---|---|---|
| **Reviewer 1** | | |
| 1 | The Study of Sutanto and Van Lanen compares different drought identification approaches: 1) the fixed threshold level method, 2) the variable threshold level method and 3) the threshold level method applied on SSI time series, for simulated river flow at the pan-European scale. They show that (average) drought event characteristics differ based on the used drought identification method. Consequently, they show that drought event forecasts differ, depending again on the used drought identification method. Overall, the main recommendation of the paper is strong and relevant, i.e., droughts differ depending on the used method and streamflow drought forecasters and stakeholders should agree which type of drought should be forecasted. In addition, I believe that Figure 6 provides an informative message for the users and developers of hydrological drought forecasting systems. | We would like to thank the reviewer for the comments, valuable suggestions, and acknowledgement of the message in our paper that drought forecasters and stakeholders should agree at front which type of hydrological drought should be forecasted (**P18L568-570**). |
| 2a | However, given that this paper focusses on the definitions of drought and methodology of drought identification, it sets an example which types of drought identification approaches can be used for drought forecasting applications (and how). Therefore, it should be extra "sharp" in its drought definition and identification approaches as well. At this stage, this is not the case and there are several methodological concerns that should be addressed carefully. In addition, the comparison of the results is far from straight forward. The used drought identification approaches do not only vary in overall method, but also in: 1) threshold (<10 percentile for the fixed and variable threshold approaches and around <50th percentile threshold for the SSI), 2) data accumulation period (1 month for the fixed and variable threshold based approaches vs. 6 months for the SSI), and 3) temporal resolution (daily vs. monthly). | The referee is concerned about the methodology used in our paper, i.e. in three aspects: 1) the thresholds to identify drought, 2) the data accumulation period, and 3) the temporal resolution. Our answers to these three questions are as follows:
 i) Our paper used the drought thresholds based on common practice in the drought community, which are in the range of 10-30th percentile of the flow duration curve (P70-90) for a Fixed Threshold (FT) or Variable Threshold (VT) and SSI below 0 (~P50). Our reason to use different thresholds (50th percentile for SSI and 10th percentile for the FT and VT) was that we would like to follow common practice for the different approaches. However, the reviewer has a point that the comparison between threshold methods (VT, FT) and SSI is not equal regarding to the use of different percentiles. Thus in the revised manuscript, we changed the thresholds from P90 into P80 for VTs and FTs (**P5L145-146**), and SSI≤-0.84 (~P80) to have a fair comparison between different drought indices (Tijdeman et al., 2020) (**P7L193-195**). |

| | | | |
|---|---|---|---|
| | | | ii) We realize that streamflow, as included SSI, comprises some catchment memory aspects (delayed flow from groundwater). Hence, in the revised manuscript, we replaced SSI-6 with SSI-1. However, we need to realize that anomalies in the accumulated flow over a longer period (e.g. SSI-6) have relevance for some purposes, such as the management of surface water reservoirs (**P6L186-P7L190**).

iii) Again, we followed common practice (see item i, above) to identify drought using these methods. Many studies used daily streamflow data to analyze drought using the threshold method and monthly streamflow data to analyze drought using the standardized indices. To the author's knowledge, only Tallaksen et al. (2009) and Van Loon et al. (2019) used the monthly data to derive drought using the threshold method and these were done only for a scientific purpose (**P5L138-140**). In the revised manuscript, however, we added to the common practice approach (daily resolution), an analysis of drought characteristics using monthly streamflow data in both FT and VT drought approaches. This allows an analysis of the VT and FT threshold approach and the SSI-1 using the same temporal resolution, i.e. monthly time scale. This implies that we have two VT and FT threshold applications: daily resolution, as frequently used, and monthly resolution to allow comparison with SSI-1 (**P5L132-136**). |
| 2b | Finally, the most novel part of this paper, which deals with the implications for drought forecasting, is rather limited and deserves more attention in my opinion. | | We extended the novel part of paper to illustrate that the outcome of the forecast depends on the drought identification method. We do this by describing: (i) pan-European maps showing forecasted drought duration (Fig. 6) and timing (Fig. 7) using different drought identification methods (FT and VT with daily and monthly resolution, and SSI-1) (Section 3.2.1, **P13L405-P14L441**) (number of drought occurrence/frequency and drought deficit volume are provided in Appendix B), and (ii) a summary of forecasted drought characteristics identified using different approaches in the Rhine River using forecasts initiated from 1st January 2003 to 1st December 2003 with a lead time of 7-month. In addition we also provide information on the percentage of ensemble members showing drought for each identification method (Fig. 8, Table 3 and 4, Section 3.2.2, **P14L442-P16L519**). |

| | | |
|---|---|---|
| 3a | **SSI computation:**
Why SSI-6? For me, it makes sense to aggregate meteorological drought indices (SPI, SPEI) to differentiate between slow and fast responding (hydrological systems), e.g., catchment with small and large storage components. However, riverflow already encompasses the accumulation and delay of the meteorological signal caused by e.g. delayed groundwater flow. From a riverflow drought perspective, it is often important to know what is currently happening in the river (SSI-1) and not what happened in the past 6 months (SSI-6). Also, the SSI-6 is not at all comparable to the 30-Day moving window used for the FT and VT approaches. This makes the interpretation of the comparison between both approaches less straight forward. Finaly, the reasoning to choose the SSI-6 over the SSI-1 because the SSI-1 results in many minor drought events does not compensate for the advantages of the SSI-1. | We do agree with the reviewer, and thus we replaced the SSI-6 results with SSI-1 (see our reply 2a, ii). |
| 3b | Why an SSI threshold of zero to identify drought? I would not term something that happens 50% of time drought. Please note that the original SPI paper of Mckee (1993) uses a similar threshold, but has the additional requirement that the SPI should at least reach a value of -1 over the course of the drought event. In addition, an SSI threshold of zero is far from comparable to an FT or VT of Q90 used for the threshold level approaches. | The reviewer has a reasonable point here. In the revised manuscript, we changed the threshold values into P80 for the threshold methods (VT, FT) and SSI≤-0.84 (~P80) in order to have a fair comparison (see our reply 2a, i) (**P7L193-195**). |
| 3c | Why the gamma distribution to derive the SSI? I agree that is hard to find a suitable distribution to fit to riverflow time series (line 150-151). However, that is not a good argument to simply use the Gamma distribution. There are likely to be better alternatives for your pan-European dataset (See e.g. Svensson et al., 2016, Tijdeman et al., 2020). Why no goodness of fit testing? The studies above conclude on different suitable candidate distributions for the SSI (other than the gamma distribution) that might be applicable for the current study. However, that does not mean that they can be applied on your dataset of simulated streamflow series by default, as your dataset might exhibit different properties as compared to the observed riverflow timeseries. Careful evaluation which distribution is most suitable for your set of rivers is required. Which distribution fitting method was use? | We used the gamma distribution to derive the SSI because the gamma distribution has been used for hydrological forecasting of both high and low flows (Slater and Villarini, 2018). The reviewer also recognized that it is hard to find a suitable distribution to fit all streamflow regimes in Europe (see also Vicente-Serrano et al., 2012). Moreover, no single distribution fits well with all monthly streamflow data in all river grid cells (n=+29,000), e.g., sample properties of streamflow in January might differ from those in August in all places (Tijdeman, et al., 2020) (**P6L180-184**). Our study does not focus on the selection of the best distribution for drought forecasting. We do not believe that another distribution (or other distributions) that consider differences in streamflow regime across Europe will change the main message of the study, i.e. that the outcome of the hydrological drought forecast depends on the identification method. Thus we believe it is better to simply use the widely selected gamma distribution in our |

| | | analysis. |
|---|---|---|
| 3d | For the forecasted SSI: Did you use the parameters of the population distribution derived from historical monthly flow values to derive the SSI for forecasted values? Or did you replace the historical values with forecasted values and than recalculated the population distribution to derive the SSI? And why, e.g., what should a forecaster do? | We used the distribution parameters derived from the observed (historic) datasets to identify the forecasted drought. Using this method, the gamma distributions were calculated from long time series of observed data, in our case 29 years, and then applied to the forecasted streamflow (Sutanto et al., 2020a, Figure A1). We did not calculate the distribution from the re-forecast datasets because the re-forecasted time series that we have are rather short (9 years) and obviously it is not the actual observed streamflow. We added this information in the revised manuscript (**P7L198-203**). |
| 4a | **Threshold approach:**
Line 123-143: Many different smoothing procedures have been applied in combination with the threshold level method. This has been done for good reason, however, sometimes resulting in an (unwanted) increase/decrease in drought occurrence, especially for the VT method. For me, a 10th percentile implies that 10% of the time series is in drought and that drought occurrence is equally distributed over the year in case of the VT method. However, by first deriving the threshold from daily streamflow data, and then smoothing both the threshold and riverflow timeseries seperately, this is not necessarily the case anymore. This might be solved relatively easily, i.e., first apply the moving average and then derive the threshold. Or you could use monthly data. | In our paper we used the moving average of the daily quantile approach (D_MA, Beyene et al., 2014) to obtain the VT thresholds. In the revised manuscript, we changed the method on how we calculate the VT thresholds. We now use monthly streamflow data to derive the monthly threshold and then we assign the monthly threshold level to each day of the month. When confronting time series of daily data (observed data, 1990-2018, and re-forecasted data 2003) with monthly threshold levels (only relevant for the VT application using a daily resolution, see our reply 2a, iii), jumps between two consecutive months might result in unrealistic drought behavior that extends around the beginning and end of each month. Therefore, we apply a 30 days centered moving average to the discrete monthly thresholds, as done, for instance, by Beyene et al. (M_MA, 2014); Van Loon et al. (2012); Van Lanen et al. (2013); Van Huijgevoort et al. (2014); Heudorfer and Stahl (2017); Van Tiel et al. (2018) (**P5L149-P6L156**). |
| 4b | Line 366-367: You encourage using monthly streamflow data for drought forecasts but use daily streamflow in your own analyses. I would have find it logical to do this as well in this study, e.g., instead of the FT and VT approaches applied on daily data, it could be applied monthly averaged data. This also increases the comparability with the SSI. Further, is there really merit in forecasting streamflow drought duration and deficit at a daily resolution, especially for the longer lead-times? Is this being done somewhere? Can this be done with any skill? If not, wouldn't it be better to just stick to monthly data for which at least some skill might be achieved? | We added the monthly drought analysis derived from the FT and VT thresholds, as additional analysis to the daily resolution to enable comparison with the SSI-1 forecast. However, we also keep the daily analysis in our revised manuscript because the daily streamflow data is commonly used in many studies using the threshold methods (see our reply 2a, iii), incl. hydrological drought projections (Prudhomme et al., 2014; Wanders and Van Lanen, 2015; Wanders et al., 2015) (**P5L132-138**). |
| 5a | **Results and discussion:**
Section 3.2. The forecasting section, which is the most the novel part of this paper, would benefit from some more attention. | We would like to thank the reviewer for his/her valuable suggestions. We extended the forecast results with the series of 12 forecasts initiated each month (from January |

| | | |
|---|---|---|
| | Figure 6 provides a nice illustration, even though it might be a little obvious at this point in the papers that drought characteristics derived with different methods will vary, given that you apply a different threshold on the same forecast data. However:
- I disagree that the drought of 2003 in the river Rhine started in August 2003. According to the SSI-1, river levels dropped to below normal anomalies much earlier. I suggest to start earlier in the year.
- Why not add the observed hydrograph to the plot?
- Isn't the fact that the VT method does not forecast a drought a good thing? According to this method, there was also no drought in the observed hydrograph (Fig. 4a) – how could this method have "performed better" (line 340).
- Why not show the SSI-1 here? | to December 2003) with a lead time of 7-month (see our reply number 2b) including the observed streamflow (Table 3 and 4). In the revised manuscript, we present the forecasted drought characteristics (occurrence, timing, duration, and deficit volume) using different identification approaches (daily FT and VT, monthly FT and VT, and SSI-1) for the pan-European river network (Section 3.2.1) and for the Rhine River in Table 3 and 4 (Section 3.2.2) (**P13-P16**). |
| 5b | Given the focus of the paper on river flow forecasts, I would expect more focus on the latter, and not only an exemplary timeseries river flow forecasts for one river / event. It would be interesting to include.
- At least, an evaluation and discussion of the spread in streamflow forecast and especially in the spread in streamflow drought forecast, and (i.e., not only the evaluation of the median forecast). What are the ranges in drought characteristics derived from the forecast ensemble?
- Consequently an evaluation or discussion of the streamflow (drought) forecasts skill, i.e., can certain "types of droughts", e.g., FT vs. VT vs. SSI, be forecasted better?
The above evaluation would benefit the consideration of multiple rivers, drought events, or start months. | We would like to thank the reviewer for the suggestions. We extended the analysis by providing: (i) maps displaying forecasted drought timing and duration across Europe using forecast data issued in July 2003 (Fig. 6, 7, B1 and B2), and (ii) tables describing forecasted drought characteristics (occurrence, timing, duration, and deficit volume) for the Rhine River using a series of 12 forecasts initiated from January 2003 to December 2003 with a lead time of 7-month (median ensemble) (see also our reply number 2b) (Table 3 and 4). An analysis of the forecast using different drought identification methods for several European rivers is beyond the scope of this paper. We believe that the map showing the pan-European pattern (see item i, , point 5b) clarifies that the example of the Rhine River is sufficiently representative. In addition, we also provide information on number of ensemble members for which drought was forecasted (x ensembles out of 25) (See Table 3 and 4). We would like to stress that the evaluation of forecast skill using SSI and threshold methods (VTs and FTs) is beyond the scope of this paper. This was published in previous papers (Van Hateren et al., 2019; Sutanto et al., 2020b). |
| 5c | Again, I would avoid the SSI-6 here, due to the strong autocorrelation of this index, which makes it relatively easy to forecast on short lead times. For example, for a forecast with a lead-time of 1 month, 5 out of 6 months are already known. Rather, I would look at the SSI 1. | As said above, we replaced the SSI-6 with SSI-1 in the main text (see our reply 2a, ii and our reply 3a). |
| 6a | Finaly, some (non-committal) suggestions for Section 3.1 that could further improve | We thank the reviewer for the suggestions. We added the drought duration and deficit |

| | | volume derived from the FTs and VTs approaches, and only drought duration for SSI in the revised manuscript (see Fig. A2 and A3). The SSI drought deficit volume is not presented because it is impossible to derive the deficit volume using the SSI approach (major drawback of standardized approaches) (**P7L210-212**). In addition, we also added a European map showing the number of minor drought events derived using the VTD (Fig. A1). |
|---|---|---|
| | the manuscript:
 • Section3.1.1 Next to showing the amount of streamflow droughts, you could consider showing other characteristics such as the average duration, deficit volume, or the number of minor drought events. This provides valuable insights in differences between methods, and further makes the notions in 3.3.1 about regions with more minor drought quantitative. In addition, you can derive a proxy for deficit volume from standardized time series. The units are meaningless and not comparable with the deficit volumes derived with FT and VT method. However, the relative difference over Europe should pop-up. | |
| 6b | • Section 3.1.2 In addition to discussing when most drought starts, it might be interesting to see when most drought occur in difference climates. This can be presented as a series of histograms for each climate, with the month on the x-axis and the fraction of drought months that occurred in that month on the y-axis. | This is an interesting suggestion. In the revised manuscript, we provided a summary of drought characteristics (number of drought occurrence/frequency, timing, duration, and deficit volume) for 5 Köppen Geiger climate regions identified using different approaches (daily FT and VT, monthly FT and VT, and SSI-1) (Table 1 and 2). |
| 7 | **Minor comments:**
 Line 2: "… the term streamflow drought forecasting, rather than streamflow forecasting …" You could briefly explain difference between the two here. | We added text to describe streamflow drought forecasting in the revised manuscript (**P1L3-4**). |
| | Line 5: "within" Correct? | We replaced "within" with "of" (**P1L6**). |
| | Line 6: Be careful with terming these extreme events. They are anomalies, but something that happens on average at least once every year, as is the case in your study, is not an extreme event. | Naming of extreme events has always a sense of subjectivity. We removed the words (**P1L7**). |
| | Line 7, 8: "observed" might be "observations" | We changed the word accordingly (**P1L8**). |
| | Line 7: "a LISFLOOD model"… are there more? | There is only one LISFLOOD model. We changed "a" in "the LISFLOOD model" (**P1L8**). |
| | Line 10: add method to VT and FT, e.g. variable threshold level method. | The word "method" was added in the revised manuscript (**P1L14**). |
| | Line 10: You also apply a threshold based approach on SSI time series. Mention this here. | An explanation about threshold to identify drought in SSI was added. However, we do this in the Methods section (**P7L193-194**). Threshold-based drought indices (called deficit characteristics in Hisdal et al., 2004) are fundamentally different from the standardized -based drought indices (Van Loon, 2015). |
| | Line 16: "Eliminate". Not true. You can still have 1-day droughts with these TL approaches. | We removed the sentence in the revised manuscript. |
| | Line 24: "IPCC" should be "The IPCC". | Thanks for the correction (**P2L29**). |
| | Line 34: This sentence slightly contradicts | We revised L34 to avoid possible |

| | | |
|---|---|---|
| | with Line 1, where you state that drought forecasting is a key element of DEWS. I would expect there to be some examples. Which contemporary "DEWS" include streamflow drought forecasting, using the approaches as described in the paper (FT, VT and SSI), not just streamflow forecasting)? | contradiction, i.e. "One of the elements to be included in a NDPP is a Drought Early Warning System that in addition to real-time monitoring contains …" (**P2L37-39**). In the preceding sentence we explain the abbreviation NDPP (National Drought Policy Plan) (**P2L36**). Furthermore, streamflow drought forecasting, using all the approaches as described in the paper (FT, VT and SSI) are developed in the EU H2020 ANYWHERE project (for background, see Sutanto et al., 2020a) (**P18L570-572**). |
| | Line 41: "evaporation" should be potential evapotranspiration
Line 47: "used" should be "be used"
Line 85: "Proxy" should be "Proxies" | We revised the text accordingly (**P2L46, L53, P4L98**). |
| | Line 49: Mention that you specifically focus on simulated streamflow drought. | We changed "hydrological drought forecasting" into "streamflow drought forecasting" (**P2L54-55**). |
| | Line 75: "There" should be "There is" | We removed the word "There". Thus the sentence became: "…., which demonstrates that none of the hydrological drought forecast approaches fit all needs" (**P3L83**). |
| | Line 89: "proxy observed streamflow" could just be "simulated streamflow" | We would like to keep the term "proxy observed streamflow" to indicate that in principal people would like to use observed data, but these spatio-temporal streamflow observed flow data do not exist. Hence, flow data obtained from a hydrological model driven by observed weather data are used as proxy for observed (same as EFAS-WB in Arnal et al., 2018 or offline simulation in Yuan et al., 2017). This is similar to reanalysis data that are a proxy for observed weather. In some cases these simulated data are just called observed, which we think should be avoided. |
| | Line 112: "re-forecasted data 2003" should be "re-forecasted data of 2003"
Line 119: "in" should be "for"
Line 147: "median" should be "expected median".
Line 179: "definitions" … "drought identification approaches" might be better.
Line 221: "drought that has" should be "droughts that have" | We changed the text accordingly (**P5L128, L141, P6L179, P8L250, P16L512**). |
| | Line 128: "were moving averaged" rephrase | The sentence was corrected (**P6L158-159**). |
| | Line 134: "For the threshold" …this refers to variable threshold approach I guess? In this section, make the clear distinction between FT and VT and seperately explain how both are derived. | The threshold here refers to both FT and VT. We revised the sentence (**P5L149-P6L156**). |
| | Line 138-140: add here that MA introduces a significant amount of auto-correlation, which affects the skill of the river flow forecast for the first 30 days significantly. | We added an explanation about the effect of 30DMA on the forecast skill (**P6L170-172**). |
| | Line 155-160: Add here that it is quite easy | We replaced the SSI-6 with SSI-1 in the main |

| | | |
|---|---|---|
| | to forecast the SSI-6 for short lead times, given the strong autocorrelation of the timeseries. E.g., for 1-month lead-times, you already know five months and only have to forecast one. | text, thus the explanation of preceding observed data is not necessary there. |
| | Line 162-164: Please explain how you classify an event with varying SSI values into one category. | In our study we only focused on the median ensemble and not the whole ensemble (25 members). Thus if the median value of SSI is in between -1 and -1.5, we classify the event as moderate drought. |
| | Line 162-177: Did you derive the climate classification yourself using the approach described in Peel et al (2007)? Or did you use their dataset? | We used their dataset (**P8L235-236**). |
| | Line 188: "Lower than median streamflow" ... Not necessarily true. Technically, above median streamflow can still be a negative SSI and vice versa. Depends on the sample and (goodness of fit) population distribution to derive the SSI. | We use the threshold SSI<-0.84 to identify SSI drought in the revised manuscript (**P7L193-195**). |
| | Line 189: Figure 3 does not show that streamflow droughts occur every year. | Figure 3 shows the drought timing i.e. the month in which commonly start, and not drought occurrences. The latter we show in Fig. 2. |
| | Line 200: This is comparing apples and pears, as the thresholds are completely different. | We changed the threshold values, i.e. special application of VT and FT thresholds, for better comparison in the revised manuscript (see our reply 2a, i). |
| | Line 203-206: Could this not be compensated by a higher number of drought in winter for the VT? | Sorry, we have to disagree. The VT method takes into account the seasonality. |
| | Line 228. "(Coincides with hydrologic years in most of Europe)" remove: unneeded repetition. Line 264-266. Is the last part, i.e., about the lowest and n-day minimum flow, needed? Interrupts flow. | We removed the sentences. |
| | Line 266-267. Looking at Fig. 5a, I find the SSI-1 timeseries much more informative about drought in the river Rhine. Rhine drought reaches is maximum in summer 2003, and recovers in winter 2004. For me, this make much more sense than the SSI-6 timeseries. Was the drought in the river Rhine really a multiyear event? Were there impact directly related to Rhine river flows over the course of 2004? | In the revised manuscript, we use only the SSI-1 forecasts rather than the SSI-6. Drought 2003 in Europe was one of the severe drought events and this was applied to the Rhine River as well. The impact of 2003 drought on the Rhine flow was apparent but we are not aware if there was an impact in 2004 meaning that there would have been a multi year drought. |
| | Line 270. For me, this description of drought in the river Rhine makes much more sense. It would make even more sense if you would use a more appropriate drought threshold (maybe SSI-1 < -0.84, corresponding to the 20th percentile). I don't see the problem of having 2003 split up in different events and question why it is better to use an SSI-6 and thereby inflate the event to a multiyear drought. | We do agree with the reviewer and therefore we revised the manuscript by using the drought threshold SSI<-0.84 and only using SSI-1 (see our reply 2a, i) (**P7L193-195**). |
| | Line 285: "C" should be century. Line 361: "rare extreme drought events" ... | Typo was corrected (**P12L365**) and we removed sentence (L361). |

| | | |
|---|---|---|
| | extreme events are by definition rare. Rephrase. | |
| | Line 295-302. Why limit yourself here to the four case study Rivers and the limited time window? You could directly compare the number of drought events & their deficit volumes over a longer time period and for all the catchments (starting by deriving the difference between Fig. 2a and b). | We do agree with the reviewer and, as stated above, we extended our analysis by providing drought duration using different approaches (Fig. A2, see also Table 1 and 2). The limited time window in Figure 4 was made to increase the readability. This was done by showing 2003 drought events in north, central, and east Europe and 2005-2006 droughts in south Europe. In the Supplementary Material we have given the drought characteristics for all four selected rivers (Tabel S1 and S2) derived from data of the entire period (1990-2018). This allows a comparison with Table 1 and 2, but please note that drought characteristics obtained for individual rivers over the period 1990-2018 may deviate from the general pattern, as reported in Table 1 and 2 (Section 3.1.1), because the drought analysis of a specific river only involves streamflow generation upstream of the river grid cell that has been selected to represent the river. |
| | Line 312-329. According the definition of drought according to VT and the SSI, droughts are expected to occur for an equal amount of time over the year. Please provide an explanation for the distinct temporal differences in drought occurrences. Or is this still referring to the start month of the drought? | Yes, we refer to the drought timing, which is identified as the month when drought mostly started. In the revised manuscript we use in a specific application of the FT and VT methods monthly streamflow data, thus in that case there is no discrepancy in the temporal resolution between threshold methods and SSI. |
| | Line 309: "(except for the Rhine River)" this contradicts with the discussion in the paragraph above. | For the selected river basins (Section 3.1.4), we did the analysis only for the selected river grid cells. The discussion in the paragraph before this section was for the whole of Europe in general. We moved the detailed analysis of four selected river basins to the Supplementary Material. |
| | Line 337: Not only meteorological drought, also streamflow drought according to the SSI-1 (Fig. 5a). | We do agree with the reviewer. However, this sentence has been deleted in the revised manuscript. |
| | Line 354. Which is good, because there was no drought according to the VT, or? | Here the VT method did not forecast drought in August 2003 using the 30DMA. The 30DMA, however, is very useful in reducing minor drought events and it is also recommended to increase the forecast skill (previous version: **P12L354-360**). In the revised manuscript, we changed the forecast initiation months to April and July. |
| | Line 382: "eliminate" ... not correct as minor droughts can still occur. | We removed the paragraph since our study does not discuss the forecast skill. |
| | Line 372-373: "the FT method produces higher drought deficit volumes and duration than VT" not shown for the pan-European dataset. | We added drought duration and deficit volume in the revised manuscript (see Figure A2, A3, Table 1, 2, Figure 6, B2, Table 3, and 4). |
| | Line 375: "occurred" should be "started". | We changed the word accordingly (e.g. **P17L542**). |
| | Line 377: "what being identified by" | We revised the whole paragraph. |

| | rephrase | |
|---|---|---|
| | Figure 1: Nice. What is the difference between light and dark grey in e.g., the Alps? | We only have grey color for ET region (Alps). |
| | Figure 2: You could add the upper boundary, e.g. 30-xx instead of >30. | We changed the whole figure in the revised manuscript. |
| | Figure 3: "The timing for drought was determined based on the first month of each drought event." This is the same as what is said in the beginning of the caption. | We removed the last sentence to avoid duplication. |
| | Figure 4: Some droughts are hardly visible (e.g. in Figure 4a). It might work to use a log-scale Figure 4: Axis lables: m3 sec-1 or m3 / sec instead of m3/sec | We changed the axis into $m^3\ sec^{-1}$. |
| | Figure 4: Are the grey vertical lines the hydrological years? | We added an explanation for the grey vertical lines. |
| | Figure 4. You might consider using a different color when VT and FT overlap. | We revised the color as suggested (orange color). |
| | Figure 5. Add grey vertical lines here as well. | We added the grey vertical lines. |
| | Figure 6. Same comments as for Figure 4 and 5. | We revised the figure accordingly. |
| | Table 1. Would be interesting to also compare average deficit volume and timing. | We added drought duration and deficit volume in Table 1, 2, 3, and 4. |

[revised manuscript text omitted]

17. Van Tiel, M., Teuling, A. J., Wanders, N., Vis, M. J. P., Stahl, K., and Van Loon, A. F.: The role of glacier changes and threshold definition in the characterisation of future streamflow droughts in glacierised catchments, Hydrol. Earth Syst. Sci., 22, 463–485, https://doi.org/10.5194/hess-22-463-2018, 2018.

18. Vicente-Serrano, S. M., López-Moreno, J. I., Beguería, S., Lorenzo-Lacruz, J., Azorin-Molina, C., and Morán-Tejeda, E.: Accurate computation of a streamflow drought index, J. Hydrol. Eng., 17(2), 318-332, doi:10.1061/(ASCE)HE.1943-5584.0000433, 2012.

19. Wanders, N. and Van Lanen, H. A. J.: Future hydrological drought across climate regions around the world modelled with a synthetic hydrological modelling approach forced by three General Circulation Models, Nat. Hazards Earth Syst. Sci, 15, 487–504, doi:10.5194/nhess-15-487-2015.

20. Wanders, N., Wada, Y. and Van Lanen, H.A.J.: Global hydrological droughts in the 21st century under achanging hydrological regime, Earth Syst. Dynam., 6, 1–15, doi:10.5194/esd-6-1-2015, 2015.
21. Yuan X., Zhang, M., Wang, L., and Zhou, T.: Understanding and seasonal forecasting of hydrological drought in the Anthropocene, Hydrol. Earth Syst. Sci., 21, 5477–5492, doi:10.5194/hess-21-5477-2017, 2017.

**Reply to reviewer 2**

We would like to thank the reviewer for valuable suggestions and comments. In this document, **P** refers to the page number and **L** refers to the line number in the recent paper. For example, **P3L65-70**, refers to page 3, lines 65-70.

| Reviewer 1 | | |
|---|---|---|
| **No** | **Comment** | **Reply** |
| 1 | The authors performed an intercomparison of three different streamflow drought indicators, with the goal to highlight the differences in the drought characteristics associated to each index and to detail the implication on drought forecast. I found the overall goal of the study meaningful, given the confusion that still arise among scientists and operational users on the topic, but I also found the paper and its structure generally out of focus. The key message of the paper "….developers of DEWS and end-users should clearly agree among themselves upon a sharp definition on which type of streamflow drought is required to be forecasted for a specific application." is in my opinion, even if relevant, better suited for a short communication or letter paper rather then a research paper. | We would like to thank the reviewer for the acknowledgement of the goal of our study. We appreciate the suggestion from the reviewer that our manuscript is better suited for a short communication paper rather than a research paper. However, a short communication paper would only be an option, if a systematic intercomparison of threshold and standardized streamflow drought indices across Europe obtained from commonly used identification approaches would exist. Such intercomparison, however, does not exist and consequently a technical paper is needed, which describes and discusses the use of different drought identification approaches (in this paper five approaches) to derive streamflow drought across Europe (Section 3.1). This has to precede the section that deals with the implication on drought forecasting, which was extended to show the differences in a more comprehensive way (Section 5.2). Hence this paper should be a technical research paper instead of a short communication paper. |
| 2 | The research results that should support this conclusion as reported in this paper are somewhat lacking in both clarity and rigorousness. | We believe that the conclusions of our study support the results that different drought indices generate different number of drought occurrences/frequency and timing, which are also related to climate regions (**P17L537**). We believe that we improved clarity and rigorousness of our results in the revised manuscript through making the drought identification methods more consistent in terms of: (i) thresholds, (ii) data accumulation period, and (iii) temporal resolution (see our reply number 3 below for more details). |
| 3 | The main drawback of the analysis is the fact that the authors uses three drought indicators that rely on quite different input data and basis hypotheses to conclude that they provide a different picture of drought. This result is quite obvious after an attentive read, given the background premises: - daily data for threshold methods vs. monthly data for SSI. - 90th percentile for threshold methods vs. median for SSI (SSI=0). - Event-based approach for threshold vs. single monthly value for SSI All these discrepancies in the | We would like to thank the referee for the comments and valuable suggestions. Our paper used the drought threshold based on common practice in the drought community, which are in the range of 10-30th percentile of the flow duration curve (P70-90) for a Fixed Threshold (FT) or Variable Threshold (VT) and SSI below 0 (~P50). Our reason that we used different thresholds (50th percentile for SSI and 10th percentile for threshold for the VT and FT) was that we would like to follow common practice for the different approaches. However, the reviewer has a |

| | | drought definition make the intercomparison a mere exercise, and its outcomes are hard to translate into actual general considerations. | point that the comparison between threshold methods (VT and FT) and SSI is not equal regarding to the use of different percentiles. Thus, in the revised manuscript, we changed the threshold from P90 into P80 for VT and FT, and SSI≤-0.84 (~P80) to have a fair comparison between different drought indices (Tijdeman et al., 2020) (**P7L193-195**). We also agree with the reviewer that our study used different temporal resolutions to analyze drought, which were daily for the threshold methods and monthly for SSI. Again, we followed the common practice to identify drought using these methods. Many studies used daily streamflow data to analyze drought using the threshold method and monthly streamflow data to analyze drought using the standardized indices. To the author's knowledge, only Tallaksen et al. (2009) and Van Loon et al. (2019) used the monthly data to derive drought using the threshold method and these were done only for a scientific purpose (**P5L136-140**). In the revised manuscript, we added analyses of drought characteristics using monthly streamflow data in both FT and VT drought approaches. |
|---|---|---|
| 4 | An additional drawback is the general lack of details on the implementation of the three approaches, which severely limits the possibility for the readers to extrapolate meaningful information from the research outputs. | We elaborated more the method section and added more drought characteristics, such as drought duration and deficit volume using different methods in the revised manuscript (Section 2.3, **P7-8**). By adding more results especially in the forecasting section (Section 3.2) (see our reply 5 below and 6c), the reader hopefully will clearly see the differences in drought characteristics because of different drought identification methods (Fig. 6, 7, 8, B1, B2, Table 3 and 4). |
| 5 | Finally, the analysis on the implications on drought forecast, which should be the main focus of the paper according to the title, is very limited in scope, and it needs to be significantly expanded in order to keep it as the focus of the paper. | We thank the reviewer for the suggestion. We expanded the analysis using the series of 12 forecasts initiated in each month (from January 2003 to December 2003) with 7 months lead time for each initiation. We do this by describing: (i) pan-European maps showing forecasted drought duration (Fig. 6) and timing (Fig. 7) using different drought identification methods (FT and VT with daily and monthly resolution, and SSI-1) (Section 3.2.1, **P13L405-P14L441**), (number of drought occurrence/frequency and drought deficit volume are provided in the Appendix B), and (ii) summary of forecasted drought characteristics identified using different approaches (FT and VT with daily and monthly resolution, and SSI-1) in the Rhine River using the series of forecasts initiated from 1st January 2003 to 1st December 2003 with a lead time of 7-month (Fig. 8, Table 3 and 4, Section 3.2.2, **P14L442-P16L519**). |

| | | |
|---|---|---|
| 6a | **Specific Comments**
**Introduction**
The authors should better highlight how different definitions of streamflow drought in DEWS exists also for two reasons: 1) different users have different needs that can be accommodate by different indicators (e.g. river navigation may be affected more by FT droughts that VT droughts), 2) different available input data lead to different definitions (e.g. threshold methods may not be suitable for monthly data, and daily data may not be available in near-real time). | We describe the reason why different definitions of streamflow drought exist in DEWS in the Conclusions (**P18L559-562**). However, we would like to leave the decision of using which drought identification approach to the users (**P18L568-573**). |
| 6b | **Data and Methods**
The description of the different drought indices need to be more explicit. How the drought events are defined for each index? How is the onset computed? Severity? Duration? Any event definition in the SSI? Etc. . . Also, more consistency on the adopted thresholds need to be enforced (why SSI=0 is used as threshold when 90th percentile is used for VT and FT?). It is also worth to mention that a VT method based on the same LISFLOOD data is currently operationally implemented as part of EDO (https://edo.jrc.ec.europa.eu/). | We expanded the method section in the revised manuscript, as suggested (Sections 2.2.1 and 2.2.2, **P5-7**). We added some information on drought characteristics in the method section, such as drought timing or onset (month when drought starts), number of drought occurrences/frequency, duration, and deficit volume (Section 2.3, **P7-8**). As mentioned above, we changed the drought threshold into P80 for FT and VT and SSI≤-0.84 (~P80) for the standardized index (our reply number 3) (**P7L193-195**). The suggested information about the VT method applied in EDO was added (**P18L560-562**). |
| 6c | **Results and discussion**
There is a clear unbalance between the historical analysis and the forecast. Give the title of the paper, I would aspect much more emphasis on the latter. | We extended our forecast analysis by providing: 1) maps displaying forecasted drought timing and duration across Europe using forecast data issued in July 2003 (Fig. 6, 7, B1 and B2), and 2) tables describing forecasted drought characteristics (occurrence, timing, duration, and deficit volume) using the series of 12 forecasts initiated from January 2003 to December 2003 with a lead time of 7-month (median ensemble) (see our reply number 5) (Table 3 and 4). The latter, however, can only be performed only for one river. In addition, we also provide information of number of ensemble members indicating drought in percent (x ensembles out of 25) (See Table 3 and 4). |

---

## Referee Report (RR1)

**Review of "Streamflow drought: implication of drought definitions and its application for drought forecasting" by Sutanto and Van Lanen.**

I thank the authors for their detailed reply to my comments. The authors greatly improved the manuscript. The comparison between daily and monthly resolution is a nice addition. The figures, especially the maps, are nice! However, few points remain.

Below you find a more detailed list of comments. In summary, the drought identification methods should be better explained and discussed. With the scope of your, you are setting an example for the drought forecasting community. Further, I think the discussion can focus a bit more on the implications of the results. You find many differences among methods and conclude that, based on these differences, end-users should agree upon a sharp drought definition. What would be nice is to have some more discussion on this. For example: What are the (dis-)advantages of the different methods? Which end-user would benefit from a fixed vs. variable approach. Who might be interested in the SSI over the VTM? And who would prefer daily instead of monthly data. Finally, I think that the readability of the manuscript can be increased by being more consistent in the used terminology. Use the same wording when describing the same thing (e.g. the wording you use to describe drought properties N, T, D etc., or the wording used to describe the different approaches).

**Comments and suggestions:**

- Line 6: "the differences of streamflow droughts using different identification approaches" → unclear, rephrase.
- Line 12: "the Standardized Streamflow Index". I do not think this is the accurate description of the approach. I would refer to it as the threshold level method applied on SSI time series
- Line 13,14: why define acronyms VTs and FTs in the abstract.
- Line 18: "Overall, the characteristics of SSI-1 drought are more or less similar to what is being identified by the monthly threshold approaches (FTM and VTM)."I am a bit surprised that SSI and FTM are the same. Especially because SSI and VTM should be very similar, and FTM and VTM show differences.
- Line 21: "To the end" should be "in the end"
- Line 39: Could remove brackets here
- Line 46: Write out to what "these" refers.
- Line 49: "Should be not" → Should not be sounds more natural to my non-native ears.
- Line 55: "which is defined as … below normal" → add "the forecasting of " on the place of the dots.
- Line 59-60: "which measures monthly normalized anomalies in streamflow and" → would at a bit more detail here, e.g., the SSI is a probabilistic index.
- Line 74: "drought indices" → drought indices is confusing here (refers to SPI, SPEI etc.)
- Line 78: "data" → streamflow data?
- Line 82: "its" → refers to nothing.
- Line 88: "results" → results and discussion sections.
- Line 98: "daily proxies for observed streamflow" → From your reply, I get why you use this terminology. However, either use it consistently throughout the manuscript, or do something like: daily proxies for observed streamflow (hereafter referred to as just streamflow for brevity reasons) ….
- Line 107: "river streamflow" → just streamflow
- Line 124: "threshold drought approach" → threshold level method. Please use this (or similar) terminology consistently throughout the manuscript.
- Line 125: "Standardized drought approach" → threshold level method applied on SSI timeseries. Please use this (or similar) terminology consistently throughout the manuscript.
- Line 128: "the water deficit in different domains of the water cycle, in our case, it is the" → redundant. Could delete.
- Line 130: ref to the original work of Zelenhasić & Salvai (1987) would fit here well.

- Line 144: How was the data aggregated: Sum or mean?
- Line 146-147: "The Q80 was considered as the drought threshold because most of the rivers across Europe are classified as perennial rivers." → I would remove this sentence. Why would one use a different threshold for Intermittent Rivers? And not a different drought identification approach (e.g. Van Huijgevoort et al. 2012)? The next sentence provides enough justification of why Q80.
- Line 149: "fewer drought events" → Nitpicking here, but not necessarily true: Q70 could also mean that few minor Q80 droughts are pooled together in one larger Q70 event.
- Line 150: "be straightforwardly be" → remove one be.
- Section 2.1. It is still not clearly described how the daily threshold is arrived.
- "whereas for the VTD method, the calculated monthly thresholds were firstly assigned as the threshold levels for each day of the respective months" → Is this correct? Isn't the VTD usually derived from daily data of the flow duration curve within a certain month?
  - If it is correct (I guess so after reading 4c). please discuss that a threshold derived from monthly data might be different from a threshold derived from daily data.
  - If it is not correct: Please clarify.
  - Didn't the cited study of Beyene et al. (2014) find that other threshold smoothing procedures were more suitable for e.g., highly seasonal (snow) regimes? Please discuss.
- Line 178: How did you estimate the parameters of the gamma distribution? L-moments, Maximum likelihood estimation, or a combination of the both. Please add.
- Section 2.2.2. You study sets an example for a broad community. This is obviously a good thing! However, I feel certain topics should be more carefully explained and discussed.
  - Please provide a bit more background about the SSI, e.g. it is a probability index, it has certain assumptions, it has uncertainties etc.
  - Ok – you use the gamma distribution, fine. However, what I would highly encourage is to include one more map to the supplementary material that shows the suitability of this distribution across all rivers. For example, you can derive a goodness of fit metric (Shapiro-Wilk, KS or something else) and show for each river how many months pass this this goodness of fit metric. Also, please discuss that other distributions might be more suitable for streamflow. Testing goodness of fit is a regularly ignored, but essential step, before using any standardized drought index.
- Line 225-226: "Obviously, the average deficit volume in a river grid cell, which we use in the historic analysis, equals the total deficit divided by the number of droughts." → Suggest to delete this sentence as it is indeed obvious.
- Line261-262: "This happens when the streamflow falls below the threshold, which is Q80 (VTs and FTs) or equal to SSI<-0.84 in our study." → consider deleting.
- Line 305. Also negatively correlated for the threshold level method applied on the SSI.
- Line 346: "This precipitation had a more marked effect on the SSI-1 drought than on the VTM and FTM droughts." → Visually, yes. But you are kind of comparing apples and pears, i.e., changes in absolute flow versus changes in SSI (standard normal distribution). Would remove or rephrase.
- Line 348: "the SSI-1 …" → and VTM
- Line 360-361: "(Tallaksen and Van Lanen, 2004)" what is this reference doing here? They give me a definition of multi-year drought? Might remove.
- Line 365-375; You might have a look for the work of Vicente-Serrano, as he did a lot of research to both the Ebro Basin and SSI.
- "The SSI-1 droughts follow the pattern of VTM droughts" this comparison is done a few times. You might at somewhere that these results are expected, given that the SSI and VTM are very similar metric (only difference is the probability distribution fitting step).
- Line 396-399: Nested sentence – difficult to follow.
- Section 3.1: general – What I miss a bit in this section is the interpretation of the results. For example: there are earlier / longer / more minor droughts with this method as compared to the other method. So what? Why and for who is this important? And which drought monitoring and early warning application benefits more from a VT as compared to a FT method and vice versa.
- Line 538: "fixed" → variable

- Line 543: "The start of SSI-1 droughts is closest to VTM droughts" as expected (see above).
- Line 555: "The differences in drought frequency, average duration, timing, and deficit volumes
- Between VT droughts (incl. SSI-1) and FT droughts highlight the importance of whether end-users of drought forecasts should take seasonality into account or not." Good point! But what would strengthen it are some examples of end-users that might benefit from a fixed versus variable drought definition.
- "forecast both standardized-based and threshold-based drought indices." → Nice! Do they also forecast using fixed and variable thresholds?
- "based upon the provided description of the identification method and product." → Nice point again – but could pick-up on this point a bit more in the discussion. I think it is even more crucial to provide accurate guidance with interpretation than a bunch of different products.

**Figures:**
- General: very nice maps!
- Fig 1. Mention the four basins in the caption.
- Fig 2. (caption) "Drought occurrences" → number of drought occurrence (consistent with 2.3)
- Fig 3. "Months when drought mostly started" → drought initiation time
- Fig 4-5. Please ignore if it does not make sense – but just wanted to note that I found red a more logic color for overlapping deficits (not orange). Or maybe you can do purple instead of orange for overlap? (red + blue = purple).
- Fig 7: "Section 2.3 explains how the drought timing is determined using forecast data." → Not needed.
- Fig 7. Why not connect all ensemble members to last observed month?
- Fig 7. Add "threshold" to the legend of Figure e-f.
- Table 1: explain why no timing (T) in table 1 for Europe.
- Figure S1: color scales are not matching, i.e., red color 90 days and 5 months

---

## Referee Report (RR2)

Dear authors,

Thanks a lot for your detailed response. Over two rounds of revisions, the methods have become a lot clearer, the results got much more complete, and the discussion improved (I much like the added paragraphs at the end). I have one comment and only few minor suggestions remaining, which the authors could consider implementing.

**Comment:**

Deficit volume is expressed in $m^3$ but derived from average daily or monthly flow in $m^3$ $sec^{-1}$. Please provide the actual deficits volumes in $m^3$ or change the unit and explain how someone (e.g. a water manager) can calculate the actual volume of water missed. Another solution would be to transfer flow to mm / day and derive deficit volume from these time series.

**Minor suggestions:**

- L16: „Earlier drought" → could state earlier in the year as earlier could also refer to the considered period.
- L45: "The standardized drought indices" → could replace with "These standardized drought indices" as there are others (not mentioned ones).
- Line 152-154: From this sentence, it is still not completely clear how you calculated the 12 monthly thresholds.
- Line 182: Suggest removing "widely selected".
- Line 189: "was" → "were"
- Line 285: "somewhat lower" → I would not call such a large decrease "somewhat lower"
- Line 310: "60% shorter" → I think 40% shorter (60% of the original)
- Line 320-329: Here, I would specifically mention differences in average river basin size among climates, as this might be a large contributor to differences in deficit volume.
- Line 525: Could start a new section here.
- Line 532: "cause impacts" → replace with "might cause impacts" as this is particularly questionable in the high flow season.
- Line 534: "or if observation record is short" → do not agree. Why are monthly methods more suitable for short records compared to daily methods?

---

## Author Response (AR2)

**Reply to reviewer 1**

We would like to thank the reviewer for valuable suggestions and comments. In this document, **P** refers to the page number and **L** refers to the line number in the recent paper. For example, **P3L65-70**, refers to page 3, lines 65-70.

| Reviewer 1 | | |
|---|---|---|
| No | Comment | Reply |
| 1 | I thank the authors for their detailed reply to my comments. The authors greatly improved the manuscript. The comparison between daily and monthly resolution is a nice addition. The figures, especially the maps, are nice! | We thank the reviewer for complements to our revised paper. This could be done because of the valuable suggestions from the reviewers. |
| 2 | In summary, the drought identification methods should be better explained and discussed. With the scope of your, you are setting an example for the drought forecasting community. Further, I think the discussion can focus a bit more on the implications of the results. You find many differences among methods and conclude that, based on these differences, end-users should agree upon a sharp drought definition. What would be nice is to have some more discussion on this. For example: What are the (dis-)advantages of the different methods? Which end-user would benefit from a fixed vs. variable approach. Who might be interested in the SSI over the VTM? And who would prefer daily instead of monthly data. | We thank the reviewer for these valuable suggestions. We added text to better explain the drought identification methods (see point AA, AB below). Furthermore, we added discussion about the advantages and disadvantages of the different drought identification methods in the revised manuscript (**P16L525-P17L549**). We also included thoughts about which end user could benefit from each of these methods (VTD, FTD, VTM, FTM, and SSI-1) (**P17L550-P18L566**). |
| 3 | Finally, I think that the readability of the manuscript can be increased by being more consistent in the used terminology. Use the same wording when describing the same thing (e.g. the wording you use to describe drought properties N, T, D etc., or the wording used to describe the different approaches) | The reviewer has a point here. We changed the inconsistency in terminology used in our manuscript. We believe that the revised manuscript now has a consistent terminology throughout the text. |
| 4 | Line by line comments | |
| A | Line 6: "the differences of streamflow droughts using different identification approaches" -> unclear, rephrase. | We revised the sentence into "…..overview of the differences between different drought identification approaches to identify droughts in the European rivers,…… (**P1L6-7**)" |
| B | Line 12: "the Standardized Streamflow Index". I do not think this is the accurate description of the approach. I would refer to it as the threshold level method applied on SSI time series | We are sorry that we confused the reviewer by using a misleading term, i.e. threshold, for a standardized approach, i.e. the SSI (whether the river is in drought or not according the SSI-1 time series).  By using the suggested phrasing, we believe that the reader will be confused about the distinction between the threshold methods and the standardized methods, as introduced in the literature (Van Loon, 2015). Keeping in mind that we would like to have a clear distinction between the two methods. We will not use "threshold level" in the context of the |

| | | | standardized methods. Hence, we replaced at relevant places the "threshold level" by "limit value" (**P7L196**). So rivers are in drought according to the SSI-1 when the limit value is below -0.84. |
|---|---|---|---|
| C | Line 13,14: why define acronyms VTs and FTs in the abstract. | | We believe the reviewer means L16-17. We removed the definition of the acronyms VTs and FTs in the revised manuscript (**P1L16-17).** These have been already defined earlier in the Abstract. |
| D | Line 18: "Overall, the characteristics of SSI-1 drought are more or less similar to what is being identified by the monthly threshold approaches (FTM and VTM)."I am a bit surprised that SSI and FTM are the same. Especially because SSI and VTM should be very similar, and FTM and VTM show differences. | | We said here "Overall". If we refer to Figure 2 (drought occurrences) and Figure B2 (drought duration), then it is hard to distinguish the difference between VTM, FTM, and SSI-1. We can see a clear difference between SSI and FTM for drought timing (Figure 3). We revised the sentence into "Overall, the characteristics of SSI-1 drought are close to what is being identified by the VTM" (**P1L18-19**). |
| E | Line 21: "To the end" should be "in the end" | | We changed the word accordingly (**P1L20**). |
| F | Line 39: Could remove brackets here | | We removed the brackets (**P2L38**). |
| G | Line 46: Write out to what "these" refers. | | We changed the word "these" into "the standardized drought indices" (**P2L45**). |
| H | Line 49: "Should be not" -> Should not be sounds more natural to my non-native ears. | | We swapped the words (**P2L48**). |
| J | Line 55: "which is defined as ... below normal" -> add "the forecasting of " on the place of the dots. | | We added the words accordingly (**P2L54**). |
| K | Line 59-60: "which measures monthly normalized anomalies in streamflow and" -> would at a bit more detail here, e.g., the SSI is a probabilistic index. | | We think that adding such details in the Introduction would disturb the text flow. We added "The SSI expresses the streamflow as a non-exceedance probability and ..." in the method section (**P6L173**). |
| L | Line 74: "drought indices" -> drought indices is confusing here (refers to SPI, SPEI etc.) | | We revised the sentence and specified which indices are meant (**P3L75**). |
| M | Line 78: "data" -> streamflow data? | | Correct, we added the word "streamflow" in the revised manuscript (**P3L79**). |
| N | Line 82: "its" -> refers to nothing. | | We revised the sentence and specified what the implications are (**P3L83**). |
| O | Line 88: "results" -> results and discussion sections. | | We revised the sentence accordingly (**P3L89-90**). |
| P | Line 98: "daily proxies for observed streamflow" -> From your reply, I get why you use this terminology. However, either use it consistently throughout the manuscript, or do something like: daily proxies for observed streamflow (hereafter referred to as just streamflow for brevity reasons). | | We added the explanation that the proxy observed streamflow hereafter is referred to as observed streamflow (**P4L100-101**) and used it throughout the rest of the manuscript. |
| Q | Line 107: "river streamflow" -> just streamflow | | We removed the word river in the revised manuscript (**P4L109**). |
| R | Line 124: "threshold drought approach" -> threshold level method. Please use this (or similar) terminology consistently throughout the manuscript. | | Thanks for reminding us to use consistent terminology. We use the term threshold drought approach and not the threshold level method throughout the manuscript. |
| S | Line 125: "Standardized drought approach" | | We believe that it will confuse readers to |

| | | | |
|---|---|---|---|
| | | -> threshold level method applied on SSI timeseries. Please use this (or similar) terminology consistently throughout the manuscript. | change the SSI into the threshold level method applied on SSI. Please see our reason in point B above, and we revised text to respond to the comment made by the reviewer (**P7L196**). |
| T | | Line 128: "the water deficit in different domains of the water cycle, in our case, it is the" -> redundant. Could delete. | We revised the sentence into "....to calculate the water deficit in streamflow" (**P5L130**). |
| U | | Line 130: ref to the original work of Zelenhasić & Salvai (1987) would fit here well. | Reference (Zelenhasić and Salvai, 1987) was added (**P5L131**). |
| V | | Line 144: How was the data aggregated: Sum or mean? | The data was averaged. We added this information in the revised manuscript for clarity (**P5L146**). |
| W | | Line 146-147: "The Q80 was considered as the drought threshold because most of the rivers across Europe are classified as perennial rivers." -> I would remove this sentence. Why would one use a different threshold for Intermittent Rivers? And not a different drought identification approach (e.g. Van Huijgevoort et al. 2012)? The next sentence provides enough justification of why Q80. | We removed the sentence as suggested. |
| X | | Line 149: "fewer drought events" -> Nitpicking here, but not necessarily true: Q70 could also mean that few minor Q80 droughts are pooled together in one larger Q70 event. | We agree with the reviewer the statement we made about Q80 and Q70 is not necessarily true. We decided to remove the sentence in the revised manuscript. |
| Y | | Line 150: "be straightforwardly be" -> remove one be. | We thank the reviewer for pointing out typo. We removed the first "be" (**P5L152**). |
| Z | | Section 2.2.1. It is still not clearly described how the daily threshold is arrived. | Below (point AA) we clarify the calculation of the thresholds. |
| AA | | "whereas for the VTD method, the calculated monthly thresholds were firstly assigned as the threshold levels for each day of the respective months" -> Is this correct? Isn't the VTD usually derived from daily data of the flow duration curve within a certain month?
• If it is correct (I guess so after reading 4c). please discuss that a threshold derived from monthly data might be different from a threshold derived from daily data.
• If it is not correct: Please clarify.
• Didn't the cited study of Beyene et al. (2014) find that other threshold smoothing procedures were more suitable for e.g., highly seasonal (snow) regimes? Please discuss. | The reviewer is correct. First, we averaged daily data into monthly data (**P5L146**). Second, we calculated the threshold level for each month. Third, we assigned the monthly threshold to each day of the respective months to obtain a first estimate of the daily thresholds (**P5L152-3154**). Lastly, we applied the 30DMA to these daily threshold to obtain the final daily thresholds. The smoothing is done to avoid jumps in the threshold (**P5L155-157**). The adopted approach in this study has been widely used in the scientific literature, e.g. Van Loon et al., 2012; Van Lanen et al., 2013; and Van Huijgevoort et al., 2014; Beyene et al., 2014. This method is called M_MA in Beyene et al. (2014). The reviewer is correct that the VTD can also be derived from daily data of the flow duration curve (called D_MA in Beyene et al., 2014). Our study analyzes the streamflow drought across Europe and not only for a specific region e.g. a mountainous region (snow region) or a semi-arid region. We decided to use the M_MA instead of D_MA or other methods because this method has been widely applied in many drought studies. |

| | | We added a discussion about the use of M_MA instead of D_MA or others in the revised manuscript (**P6L160-167**). |
|---|---|---|
| AB | Line 178: How did you estimate the parameters of the gamma distribution? L-moments, Maximum likelihood estimation, or a combination of the both. Please add. | The alpha and beta parameters of the gamma probability density function are estimated for each grid cell and for each month of the year. We calculated the alpha and beta by using the method of moments. We added this information in the revised version (**P6L177-178**). |
| AC | Section 2.2.2. You study sets an example for a broad community. This is obviously a good thing! However, I feel certain topics should be more carefully explained and discussed.
• Please provide a bit more background about the SSI, e.g. it is a probability index, it has certain assumptions, it has uncertainties etc.
• Ok – you use the gamma distribution, fine. However, what I would highly encourage is to include one more map to the supplementary material that shows the suitability of this distribution across all rivers. For example, you can derive a goodness of fit metric (Shapiro-Wilk, KS or something else) and show for each river how many months pass this goodness of fit metric. Also, please discuss that other distributions might be more suitable for streamflow. Testing goodness of fit is a regularly ignored, but essential step, before using any standardized drought index. | We expanded the background of SSI in the revised manuscript (**P6L173-174**).

We believe that providing results on the testing of the suitability of the gamma distribution to derive the SSI for all river grid cells in Europe and each month (in total > 348,000 parameter sets) is beyond the scope of this study. The main message in our study is that the different approaches produce different drought characteristics, which we think is not substantially impacted by the choice of another probability distribution. We added text on the use of different probability distributions that might be more suitable for streamflow drought in some cases than the gamma distribution including references for studies that performed this analysis (**P6L178-182**). We also added a remark on the choice of the probability distribution at the end of Section 3.2 Implication of different drought identification approaches to forecast streamflow (**P17L546-549**). |
| AD | Line 225-226: "Obviously, the average deficit volume in a river grid cell, which we use in the historic analysis, equals the total deficit divided by the number of droughts." -> Suggest to delete this sentence as it is indeed obvious. | We revised the sentence (**P8L226-228**). |
| AE | Line261-262: "This happens when the streamflow falls below the threshold, which is Q80 (VTs and FTs) or equal to SSI<-0.84 in our study." -> consider deleting. | The sentence was deleted. |
| AF | Line 305. Also negatively correlated for the threshold level method applied on the SSI. | We removed part of the sentence, because the negative correlation applies to all drought identification approaches (**P10L306**). |
| AG | Line 346: "This precipitation had a more marked effect on the SSI-1 drought than on the VTM and FTM droughts." -> Visually, yes. But you are kind of comparing apples and pears, i.e., changes in absolute flow versus changes in SSI (standard normal distribution). Would remove or rephrase. | We agree with the reviewer and removed the sentence accordingly. |
| AH | Line 348: "the SSI-1 ..." -> and VTM | We revised the sentence accordingly (**P11L350**). |
| AI | Line 360-361: "(Tallaksen and Van Lanen, | Reference was removed. |

| | | |
|---|---|---|
| | 2004)" what is this reference doing here? They give me a definition of multi-year drought? Might remove. | |
| AJ | Line 365-375; You might have a look for the work of Vicente-Serrano, as he did a lot of research to both the Ebro Basin and SSI. | We thank for the suggestion. We added the reference (**P12L377**). |
| AK | "The SSI-1 droughts follow the pattern of VTM droughts" this comparison is done a few times. You might at somewhere that these results are expected, given that the SSI and VTM are very similar metric (only difference is the probability distribution fitting step). | We thank the reviewer for the suggestion. We revised the sentence into "As expected, the SSI-1 droughts follow the pattern of VTM droughts because both metrics consider seasonality" (**P12L374-375**). |
| AL | Line 396-399: Nested sentence – difficult to follow. | We divided the sentence into two: "Our generic finding that the streamflow drought characteristics (frequency, duration, timing) derived using different identification methods differ is in line with the observations made by Vidal et al. (2010). Their study in France also concluded that different identification methods (only standardized-based indices at multiple time scales) generate different drought characteristics" (**P13L400-403**). |
| AM | Section 3.1: general – What I miss a bit in this section is the interpretation of the results. For example: there are earlier / longer / more minor droughts with this method as compared to the other method. So what? Why and for who is this important? And which drought monitoring and early warning application benefits more from a VT as compared to a FT method and vice versa. | The discussion about why drought identification approaches differ, as well as advantages and disadvantages of different approaches was added in the revised manuscript (**P16L525-P17L549**). Moreover, which end user would benefit from each of these methods (VTD, FTD, VTM, FTM, and SSI-1) are also discussed (**P17L550-P18L566**). |
| AN | Line 538: "fixed" -> variable | We thank the reviewer. The typo was corrected (**P18L585**). |
| AO | Line 543: "The start of SSI-1 droughts is closest to VTM droughts" as expected (see above). | We revised the sentence into "The start of SSI-1 droughts is closest to VTM droughts because both methods use a monthly resolution and consider seasonality" (**P18L590-591**). |
| AP | Line 555: "The differences in drought frequency, average duration, timing, and deficit volumes between VT droughts (incl. SSI-1) and FT droughts highlight the importance of whether end-users of drought forecasts should take seasonality into account or not." Good point! But what would strengthen it are some examples of end-users that might benefit from a fixed versus variable drought definition. | We added some examples of end-users that might benefit from different drought identification methods in the revised manuscript (**P17L550-P18L566**). |
| AQ | "forecast both standardized-based and threshold-based drought indices." -> Nice! Do they also forecast using fixed and variable thresholds? | Thanks for asking this question. The ADEWS forecasts droughts in streamflow, groundwater, runoff, and precipitation using the VTD approach (Sutanto et al., 2020) while the EDO only forecasts a combined indicator consisted of the SPI, SPEI, a Soil Moisture Index (SMI, soil moisture anomaly), the fAPAR (vegetation) anomaly, and low flow |

| | | index (Cammalleri et al., 2020; 2021). The low flow index also uses VTD. The fixed threshold methods (FTD and FTM) are not used in any of these DEWS. We added discussion about the use of drought identification approaches in both DEWSs in the revised manuscript (**P19L609-616**). |
|---|---|---|
| AR | "based upon the provided description of the identification method and product." -> Nice point again – but could pick-up on this point a bit more in the discussion. I think it is even more crucial to provide accurate guidance with interpretation than a bunch of different products. | We thank the reviewer for his/her suggestion.  We added a discussion about advantages and disadvantages of different drought identification methods in the end of Section 3.2. (**P16L525-P17L549**), as well as a discussion about the end user that would possibly benefit from each of these methods (VTD, FTD, VTM, FTM, and SSI-1) (**P17L550-P18L566**). See, also point AM. |
| 5 | Figures | |
| A | Fig 1. Mention the four basins in the caption. | The name of four river basins was added in the figure's caption (**P29**) |
| B | Fig 2. (caption) "Drought occurrences" -> number of drought occurrence (consistent with 2.3) | We revised the caption accordingly (**P30**). |
| C | Fig 3. "Months when drought mostly started" -> drought initiation time | We revised the caption into "Drought timing (onset)…" (**P31**) |
| D | Fig 4-5. Please ignore if it does not make sense – but just wanted to note that I found red a more logic color for overlapping deficits (not orange). Or maybe you can do purple instead of orange for overlap? (red + blue = purple). | Purple color is used to identify SSI drought. We believe it is better to stick to the color choice as it is. |
| E | Fig 7: "Section 2.3 explains how the drought timing is determined using forecast data." -> Not needed. | We removed the sentence accordingly. |
| F | Fig 8. Why not connect all ensemble members to last observed month? | When we designed the figure, we thought that the figure would read best, when the x-axis would cover the same period Jan 2003 – Jan 2004 for both initiation dates (April, July). The observed streamflow covers the whole period, whereas the forecast ensemble covers 7 months. Hence, we decided not to revise the figure. |
| G | Fig 8. Add "threshold" to the legend of Figure e-f. | The legends for Figure 8e and 8f were updated. In Fig. 8e and 8f "limit value -0.84" is added (**P36**) |
| H | Table 1: explain why no timing (T) in table 1 for Europe. | Timing (onset) from each region differs and it is not a quantitative measure. We could calculate the median timing or average but we believe this does not make sense. For example, if the timing in one climate region is in spring and in another climate region in autumn, then the median or average timing will be in summer, which is not correct. We explained this in the revised manuscript (**P37**). |
| I | Figure A2: color scales are not matching, i.e., red color 90 days and 5 months | We changed the colors for monthly threshold in the revised manuscript (Now Figure B2, **P42**). |

**References**

Van Loon, A. F. Hydrological drought explained. WIREs Water, doi:10.1002/wat2.1085 (2015).

Zelenhasic, E. & Salvai, A. A mthod of streamflow drought analysis. Water Resources Research, 23(1), 156-168, https://doi.org/10.1029/WR023i001p00156 (1987).

**Reply to reviewer 2**

We would like to thank the reviewer for valuable suggestions and comments. In this document, **P** refers to the page number and **L** refers to the line number in the recent paper. For example, **P3L65-70**, refers to page 3, lines 65-70.

| Reviewer 2 | | |
|---|---|---|
| **No** | **Comment** | **Reply** |
| 1 | I would like to thank the authors for carefully considering my suggestions and comments. Overall, I found the revised version much improved, with the additional analyses adding very interesting insight on the topic. | We thank the reviewer for complements to our revised paper. This could be done because of the suggestions from the reviewers. |
| 2 | I still have only one major concern with the presented paper. The authors stress the difference between monthly and daily methods in term of number of event, average duration and deficit, while also highlighting how many of the reported events (especially in the VTD) are basically only minor events. Under such circumstances, average statistics may be not good proxy of the performance of the index, and in my opinion this is an issue that needs to be better addressed, with specific analyses. As an example, if the VTD reports 3 events, 1 very big (2 months) plus 2 minor (of 1 day each), whereas the VTM reports only the major event (2 months), are the two versions so different at the end (3 events vs. 1)? I think that a more "fair" comparison would be in terms of "total" quantities rather than average. Like: total deficit and total number of day under drought. I suggest to add to the mix those metrics as well, in order to better explain the differences between the different indices. | We thank the reviewer for his/her concern about the minor drought events when using the VTD. As you noted, the number of drought occurrences is a total quantity. However, we do not believe that it is good idea to change the average statistic of the other drought characteristics into total quantities. For example, total drought duration will end up in 20% of length of the time series because we used the Q80 threshold, i.e. 28 years of observations will result in 67 months drought duration (28 x 12 months x 20%). Thus, we believe it is fair to use average statistics of drought characteristics instead of total quantities. We addressed the topic of minor droughts in Section 3.1.1 (**P9L279-287**). As described there in a vast area (e.g. Cfb and Dfb climates) more than half of the drought events are shorter than 30 days. This implies that the number of VTD droughts longer than 1 month in these regions is somewhat lower than VTM droughts. We added some text to let the reader realize this aspect of minor drought (**P9L284-285**). Furthermore, we would like to add that we implemented the VTD approach as commonly done in literature, where specific methods are used to exclude minor droughts. The purpose of the paper is to compare the outcome of drought identification approach as commonly implemented. In our study, we applied the 30DMA to avoid minor drought events e.g. drought that has a duration of 1 day or 2 days. This 30DMA method, in general, removes the short drought event but it cannot completely remove the minor drought event, e.g. drought that has duration in between a few days to a month. Additional methods such as the inter-event time method (IT-method), the moving average procedure (used in this study), and the sequent peak algorithm (SPA) are also used to exclude minor droughts when using daily data, as we discuss in our manuscript |

| | | | |
|---|---|---|---|
| | | | (**P16L507-514**). |
| | Similarly, when FTD and VTD are compared, you need to find a way to distinguish between the cases when VTD detects more actual events (i.e. events in different seasons) vs. the cases when FTD detects a single event while VTD "splits" the same in multiple smaller events (but close in time). Also in this case, total quantities may alleviate the problem. | | The FTD and VTD are conceptually different; the VTD considers seasonality whereas the FTD does not, as described in the manuscript e.g. **P5L143-145**). This means that the major difference is that the VTD may detect droughts both in the low flow season and in the high flow season when the flow is below normal. The FTD only detects event in the low flow season. This is the major reason that the VTD approach identifies more drought events than the FTD. Hence, we think, in general, we should not focus too much on the low flow season where some events may be split in more subevents according to one of the methods as a main reason for the differences in number of drought occurrences.

We believe that the detailed comparison of different drought identification approaches performed for 4 selected river basins is sufficient to illustrate also the difference between VTD and FTD (Figure 4 and 5). For example, Figure 4a clearly shows minor VTD drought in spring 2003 and minor FTD drought in autumn 2004. |
| 3 | As a final comment, I found the revised text a little unpolished. I report some minor issues and comments regarding the first pages of the manuscript, but I suggest a careful revision of the full text. | | We thank for the reviewer's feedback on the first pages. We read carefully the rest of the manuscript and we revised it at several places. |
| 4 | Line by line comments | | |
| A | P1 L5. "The way, how…" is repetitive. | | We corrected the text (**P1L5**). |
| B | P1 L14. Here it should read "more than…" | | We revised the word accordingly (**P1L14-15**). |
| C | P1 L21. "To the end…" sounds out of place. | | We revised the sentence into "In the end" (**P1L20**). |
| D | P2 L29. "from among others drought…" is not clear. please rephrase. | | We revised the sentence into "….that impacts of drought on society….." (**P2L28**). |
| E | P2 L31. Given the focus of the paper on streamflow drought over Europe, it may be worth to mention this recent study (HESS 24, 5919-5945). | | Suggested literature was added (**P2L30**). |
| F | P3 L82. Please spell our incl. | | The word was fully written (**P3L83**). |
| G | P3 L82. "Europa" should read "Europe". | | We corrected the typo (**P3L85**). |
| H | P4 L95. Looking at the maps, I'm assuming that only cells with a minimum contributing area are considered, especially because the threshold method may not work as intended in rivers with streamflow close to zero. This is a good place to mention that. | | For the map, we only plotted the major European rivers. This indicated that small rivers are excluded from our plot. We thank the reviewer for mentioning that the threshold method used in our study does not work for rivers with flow close to zero. Indeed, the reviewer is correct. We added this information in the Section 2.2.1 paragraph 2 where we discuss the use of threshold Q80 (**P5L149-151**). |
| I | P4 L100. "Center" should read "Centre". | | We thank the reviewer for spotting the typo |

| | | | |
|---|---|---|---|
| | | | because of the US English auto correction. The word was revised (**P4L102**). |
| J | P5 149-174. I suggest to summarize this discussion a bit, since is taking most of the methodology section even if this is not the key point on the analysis. | | We moved the details about how the 30DMA has been implemented to avoid minor droughts in the historic and forecasted daily streamflow data to a new Appendix A (**P20L631-649**). The other appendices have been renamed. |
| K | P7 L193. "… falls below -0.84…"Looking at the next section, it seems that also for SSI an event-based approach is adopted. Does an event start when SSI falls below the threshold and ends when it returns above (as for the threshold methods) or is each monthly value treated separately? Please clarify here. | | We derived drought characteristics from the drought events when the streamflow falls below the threshold level (Q80) and the SSI-1 values fall below the limit value of -0.84 for both historical analysis and reforecasts. We clarified this in the revised manuscript (**P5L133-134** for threshold and **P7L195-196** for SSI) |
| L | P8 L252. Again, the criteria adopted to selected those 29,000 cells (i.e. minimum contributing area) need to be highlighted in the methodology. I'm sure that Europe is covered by much more than 29,000 5x5 km2 cells. | | We thank the reviewer for raising this topic. In this study, we only selected major European rivers, indicated by river cells that have average discharge above 10 m³/sec (n=~29,000). We added information about selecting river grid cells in the data section (Section 2.1.) (**P4L105-106**). |
| M | Authors: The suggested information about the VT method applied in EDO was added (P18L560-562). The added reference is to the CDI index, and not to the streamflow drought index (Hydrol. Sci. J. 62(3), 346-358). In my opinion, a better place to refer to this operational index would be in the Lisflood section (since is based on these data). | | We added the suggested reference (**P19L610**) and moved the Sepulcre-Canto et al., 2012 reference to the method section (**P4L115**). |

---

## Author Response (AR3)

**Reply to reviewer**

We would like to thank the reviewer for valuable suggestions and comments. In this document, **P** refers to the page number and **L** refers to the line number in the recent paper. For example, **P3L65-70**, refers to page 3, lines 65-70.

| No | Comment | Reply |
|---|---|---|
| **Reviewer** | | |
| 1 | Deficit volume is expressed in $m^3$ but derived from average daily or monthly flow in $m^3$ $sec^{-1}$. Please provide the actual deficits volumes in $m^3$ or change the unit and explain how someone (e.g. a water manager) can calculate the actual volume of water missed. Another solution would be to transfer flow to mm / day and derive deficit volume from these time series. | We thank the reviewer for his/her remark. Indeed the reviewer is correct that we derive the deficit volume from average daily streamflow in $m^3$ $s^{-1}$. We changed the streamflow unit into $m^3$ $d^{-1}$ for deficit volume calculation in the revised manuscript (table caption and text) (e.g. **P10L323, P37** Table 1) and explained how to calculate the actual deficit volume in $m^3$ (**P7L225-P8L227**). |
| 2 | L16: „Earlier drought" -> could state earlier in the year as earlier could also refer to the considered period. | We revised the sentence into "….drought occurrences earlier than….." (**P1L16-17**). |
| 3 | L45: "The standardized drought indices" -> could replace with "These standardized drought indices" as there are others (not mentioned ones). | We revised the text accordingly (**P2L45**). |
| 4 | Line 152-154: From this sentence, it is still not completely clear how you calculated the 12 monthly thresholds. | To make it clear, we divided the sentence into two parts. First part is dedicated for VTM (**P5L151-152**) and the second part is for VTD (**P5L152-153**). We also added explanation that we followed the M_MA method in Beyene et al. (2014) (**P5L157**). |
| 5 | Line 182: Suggest removing "widely selected". | The words were deleted (**P6L182**). |
| 6 | Line 189: "was" -> "were" | We revised the word accordingly (**P6L189**). |
| 7 | Line 285: "somewhat lower"-> I would not call such a large decrease "somewhat lower" | The word "somewhat" was deleted (**P9L286**). |
| 8 | Line 310: "60% shorter" -> I think 40% shorter (60% of the original) | We thank the reviewer for careful reading and we changed the number into 40% (**P10L311**). |
| 9 | Line 320-329: Here, I would specifically mention differences in average river basin size among climates, as this might be a large contributor to differences in deficit volume. | The sentence was added in the revised manuscript (**P11L330-331**). |
| 10 | Line 525: Could start a new section here. | We decided to leave it as it is. |
| 11 | Line 532: "cause impacts" à replace with "might cause impacts" as this is particularly questionable in the high flow season. | We revised the sentence accordingly (**P17L534**). |
| 12 | Line 534: "or if observation record is short" -> do not agree. Why are monthly methods more suitable for short records compared to daily methods? | We revised the sentence and removed "or if observation record is short" (**P17L535-536**). |

**Reference**

Beyene, B. S., Van Loon, A. F., Van Lanen, H. A. J. & Torfs, P. J. J. F. Investigation of variable threshold level approaches for hydrological drought identification. Hydrol. Earth Syst. Sci. Discuss., 11, 12765–12797, doi:10.5194/hessd-11-12765-2014, 2014.